# Determining structures of RNA conformers using AFM and deep neural networks

Maximilia F. S. Degenhardt[1], Hermann F. Degenhardt[1], Yuba R. Bhandari[1], Yun-Tzai Lee[1], Jienyu Ding[1], Ping Yu[1], William F. Heinz[2], Jason R. Stagno[1], Charles D. Schwieters[3], Norman R. Watts[4], Paul T. Wingfield[4], Alan Rein[5], Jinwei Zhang[6] & Yun-Xing Wang[1✉]

Much of the human genome is transcribed into RNAs[1], many of which contain structural elements that are important for their function. Such RNA molecules—including those that are structured and well-folded[2]—are conformationally heterogeneous and flexible, which is a prerequisite for function[3,4], but this limits the applicability of methods such as NMR, crystallography and cryo-electron microscopy for structure elucidation. Moreover, owing to the lack of a large RNA structure database, and no clear correlation between sequence and structure, approaches such as AlphaFold[5] for protein structure prediction do not apply to RNA. Therefore, determining the structures of heterogeneous RNAs remains an unmet challenge. Here we report holistic RNA structure determination method using atomic force microscopy, unsupervised machine learning and deep neural networks (HORNET), a novel method for determining three-dimensional topological structures of RNA using atomic force microscopy images of individual molecules in solution. Owing to the high signal-to-noise ratio of atomic force microscopy, this method is ideal for capturing structures of large RNA molecules in distinct conformations. In addition to six benchmark cases, we demonstrate the utility of HORNET by determining multiple heterogeneous structures of RNase P RNA and the HIV-1 Rev response element (RRE) RNA. Thus, our method addresses one of the major challenges in determining heterogeneous structures of large and flexible RNA molecules, and contributes to the fundamental understanding of RNA structural biology.

Knowledge about RNA structure and dynamics is important for understanding functions[3,4,6,7], designing novel RNA devices[8] and developing RNA-targeting compounds[9]. Since the first 3D structures of tRNA were determined nearly half a century ago[10,11], the structures of many stable RNA structures have been determined by NMR, X-ray crystallography, and most recently, cryo-electron microscopy[12]. As these methods rely on signal averaging over relatively homogeneous samples, they are not particularly suited to studying highly heterogeneous RNA molecules, which are functionally dynamic and do not exist in a single stable conformation under physiological conditions. Nevertheless, a number of techniques have been applied to studying RNA conformational diversity and dynamics of either ensemble behaviours[4,13–18] or single molecules using sparse distances[19–21], but none of these techniques provides direct visualization of heterogeneous structures.

A recent study under physiologically relevant conditions illustrated the conformational heterogeneity of a 210-nucleotide (nt) RNA[2], showing that RNA can remain structured but in multiple heterogeneous conformations. The conformational heterogeneity of RNA is conceptually different from unfolded or intrinsically disordered proteins. It is a hallmark feature of functional RNAs and a prerequisite for their ability to interact with various ligands in the cellular environment[4,22]. Thus, a single snapshot structure of an RNA falls short of accurately describing the conformational landscape associated with its function, as clearly demonstrated recently in a cellular context[4]. Given the rapid progress in RNA research and the widespread applications of RNA in the biomedical and public health sectors, developing a method for studying the highly heterogeneous conformational space of RNA is all the more important and urgent.

Atomic force microscopy (AFM) topographic images provide direct global structural information with a high signal-to-noise ratio, enabling visualization of individual molecules at a resolution that is sufficient to discern duplex helical grooves without distortion[2,23–27]. The usefulness of global structural information in restraining RNA structures for obtaining more accurate structure prediction was previously implicated[28,29] and the use of AFM to visualize heterogeneous RNA conformations under a physiological solution condition has been demonstrated[2]. However, the quantitative correlation between a topographic AFM image and the underlying atomistic topological structure, its use for

[1]Protein–Nucleic Acid Interaction Section, Center for Structural Biology, Center for Cancer Research, National Cancer Institute, Frederick, MD, USA. [2]Optical Microscopy and Analysis Laboratory, Cancer Research Technology Program, Frederick National Laboratory for Cancer Research, Frederick, MD, USA. [3]Computational Biomolecular Magnetic Resonance Core, National Institute of Diabetes and Digestive and Kidney Diseases, National Institutes of Health, Bethesda, MD, USA. [4]Protein Expression Laboratory, National Institute of Arthritis and Musculoskeletal and Skin Diseases, National Institutes of Health, Bethesda, MD, USA. [5]Retrovirus Assembly Section, HIV Dynamics and Replication Program, National Cancer Institute, Frederick, MD, USA. [6]Structural Biology of Noncoding RNAs and Ribonucleoproteins Section, Laboratory of Molecular Biology, National Institute of Diabetes and Digestive and Kidney Diseases, National Institutes of Health, Bethesda, MD, USA. ✉e-mail: wangyunx@mail.nih.gov

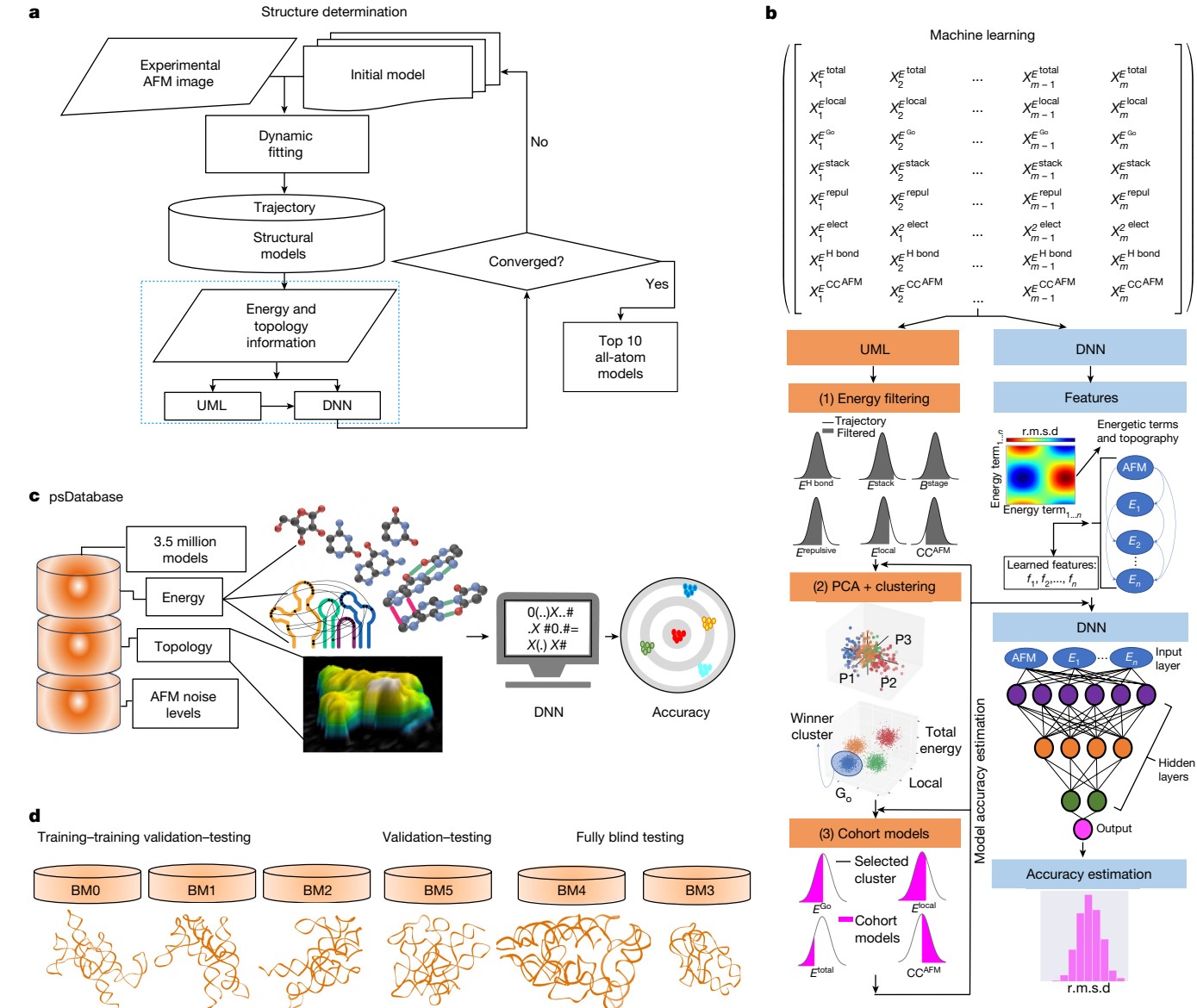

**Fig. 1 | Holistic RNA structure determination using HORNET. a**, Overall workflow for HORNET. The input comprises AFM topography data ($x$, $y$ and $z$ dimensions) from an experimental AFM image and an initial model. Dynamic fitting is driven by the AFM and structure-based potentials (Methods). Coarse-grained models are generated from the trajectory of dynamic fitting in the form of energy and topology information containing the complete list of all energy values and the overall fit to AFM topography (CC$^{AFM}$) associated with each trajectory model. This information is passed to the unsupervised machine learning (UML) or/and DNN for clusterization and estimation of the accuracy of each model in terms of r.m.s.d. relative to the ground-truth structure. Convergence is defined as a distribution of estimated accuracy with a population of models with r.m.s.d. below 7 Å. If the DNN process has converged, the top 10 models are converted to all-atom coordinates. If convergence is not reached, the dynamic fitting is performed a second time with a different initial model or a longer time for the dynamic fitting. **b**, Schematic of the main steps of UML and DNN (Methods). PCA, principal components analysis. **c**, Representation of the psDatabase composed of 3.5 million continuous trajectory structures of the RPR catalytic domain used to train and optimize the machine learning algorithm. The trained DNN architecture has the capability to estimate the accuracy of the structural model underneath the AFM topography. **d**, The approximately 56 million trajectory models used for HORNET benchmarking, grouped according to usage: training–training validation–testing (BM0, BM1 and BM2), validation–testing (BM5) and fully blind testing (BM3 and BM4).

recapitulation of individual RNA conformers with accuracy estimation, and the software to carry out the calculation—all of which are essential for establishing a robust and reliable method for 3D structure determination—have not yet been disseminated and demonstrated.

Here we present HORNET[30], a novel method for determining the individual 3D topological structures of heterogeneous RNA conformers (Fig. 1). Our method drives the conformational trajectory of models from dynamic fitting towards a convergence that satisfies both the weighted AFM pseudo-potentials and classical Gibbs free-energy descriptions[31,32] (Fig. 1a and Methods). The trajectory structures are then clustered and evaluated by both unsupervised and supervised deep learning using a holistic consideration of all energetic and topographic information (Fig. 1b). We applied a novel deep neural network (DNN) architecture trained using our pseudo-structure database (psDatabase) and tested and validated extensively using six benchmark cases (BM0–BM5) to provide an accuracy estimation of top structures in terms of root-mean-square deviation (r.m.s.d.) (Fig. 1c,d). A total of around 56 million trajectory models used for benchmarking were generated using three different RNAs more than 200 nt in size: RNase P RNA (RPR), cobalamin riboswitch and group II intron, and various

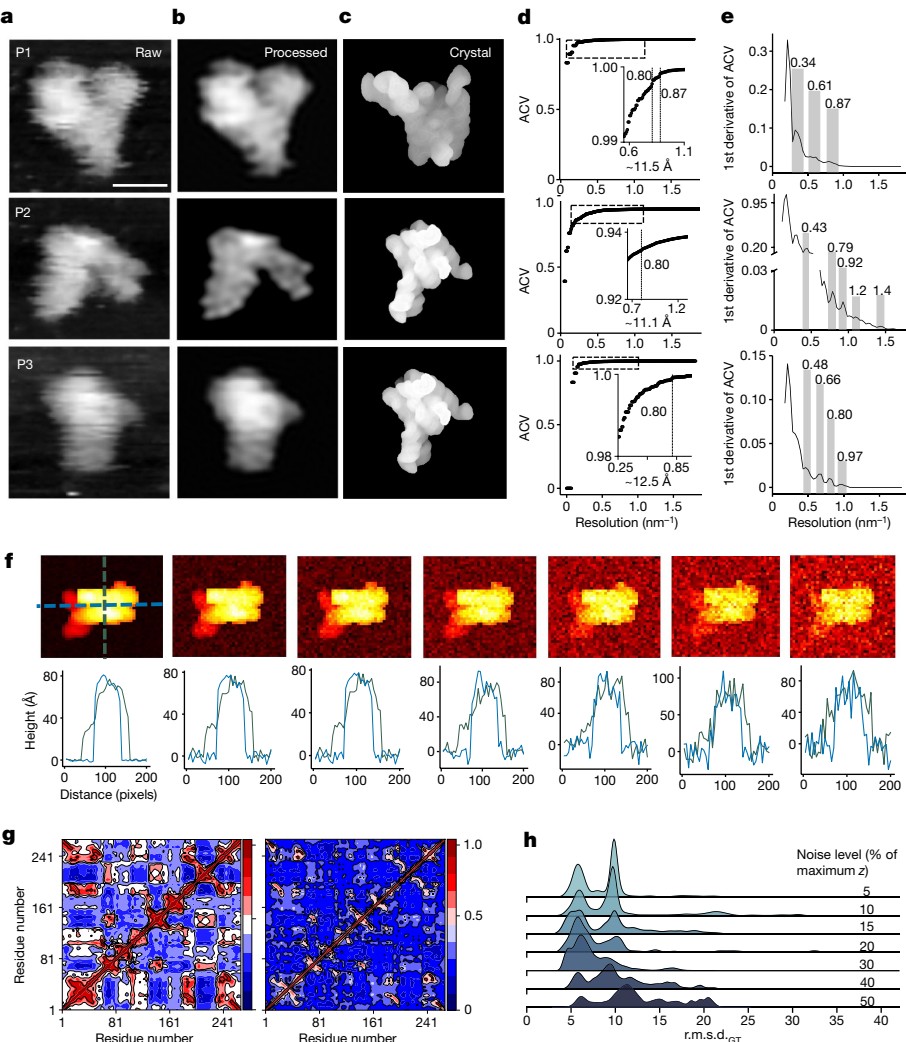

**Fig. 2 | RNA structures from AFM images. a–c**, Raw (**a**) and processed (**b**) AFM particle images and crystal structure (PDB: 2A64) rendered as a molecular surface (**c**) of the full-length RPR. The raw single-molecule images were taken once—that is, without signal averaging. For particles P1, P2 and P3, the crystal structure were orientated to best match the AFM images. Scale bar, 10 nm. **d**, Resolution estimation using the auto-correlation value[34] (ACV). Inset graphs show similar plots for the spatial regimes with an abrupt decrease in ACV, where the discontinuities in the ACV profiles indicate structural features present in the image at a particular spatial resolution, observed at approximately 0.87 nm[−1] (11.5 Å), 0.90 nm[−1] (11.1 Å) and 0.80 nm[−1] (12.5 Å) in P1, P2 and P3, respectively. **e**, The first derivative of ACV showing more detailed profile variation. Grey bars indicate ACV discontinuity regimes, with a maximum spatial threshold at approximately 0.34 nm[−1] (29 Å) and minimum spatial thresholds of approximately 0.87 nm[−1] (11.5 Å), 1.4 nm[−1] (7.2 Å) and 0.97 nm[−1] (10.3 Å) for P1, P2 and P3, respectively. **f**, AFM topographical images of the RPR catalytic domain (PDB: 3DHS) simulated at different noise levels. The $z$ height signal profiles in the $x$–$y$ plane are shown below each image. Applied noise levels (left to right): 5, 10, 15, 20, 30, 40 and 50% of the maximum $z$ height. **g**, DCCMs of free (left) and AFM-restrained (right) molecular dynamics trajectories. **h**, Stacked r.m.s.d. relative to the ground-truth structures (r.m.s.d.$_{GT}$) distribution plots of the dynamic fitting trajectory in Å at each noise level.

initial structural models (Fig. 1d and Supplementary Table 1). We then applied HORNET to solve three novel structures of the full-length RPR and five novel structures of the HIV-1 RRE conformers.

## From particle AFM image to 3D structure

We use RPR from *Bacillus stearothermophilus* as an example to show the images of individual RNA molecules recorded by AFM (Fig. 2a,b and Extended Data Fig. 1). These 3 topographic images capture 3 individual RNA molecules (P1, P2 and P3) in 3 different conformational states, none of which is identical to the crystal structure[33], as evidenced by their cross-correlation (CC[AFM]) scores of 0.77, 0.80 and 0.87 for P1, P2 and P3, respectively (Fig. 2c). The background noise of the particle topography is between 1% and 5% of the maximum $z$ height (Extended Data Fig. 2). By applying a low-pass Fourier filter[34] (Extended Data Fig. 2), the resolutions of the P1, P2 and P3 images were determined to be

0.87 nm[−1] (11.5 Å), 0.90 nm[−1] (11.1 Å) and 0.80 nm[−1] (12.5 Å), respectively (Fig. 2d,e). Although an image resolution of around 12 Å would seem to limit the use of AFM for structure determination, the characteristics of RNA structure make AFM an ideal technique for investigating the structures and dynamics of heterogeneous RNAs. First, RNA folding is hierarchical[35] and modular, making topographic spatial information easily discernable. Second, the majority of RNA structures comprises A-form duplexes, whose dimensions of major and minor grooves are within the resolution of the AFM image. Third, the backbones of those highly conserved A-form duplexes, which account for more than 70% of the mass in RNA structures in structural databases, vary within approximately 1.5 Å in terms of r.m.s.d.[36]. Thus, in principle, given an initial structural model constrained by covalent bond linkages and secondary structural information as prior knowledge, 3D topological structures can be recapitulated from AFM molecular surfaces with an uncertainty significantly lower than the inherent resolution limits of

the AFM data itself, an approach similar to that used in low-resolution electron density maps[37]. Of note, no structures of biomacromolecules have been determined using information solely from an individual macromolecule, as all reported structures were determined using signal-averaging methods. Moreover, the paucity of RNA structures does not sufficiently cover the broad conformational landscapes that RNA can sample. In fact, there are only 4 classes of naked RNAs larger than 210 nt with resolutions better than 3.5 Å: adenosylcobalamin riboswitch, group I and group II introns, and RPR. RPR is a multi-turnover ribozyme that processes the 5′-ends of pre-tRNA and other RNAs, and exhibits diverse structural features with known conformational flexibility[16,38,39]. For establishing the method, we used the catalytic core domain of RPR[40] (Protein Data Bank (PDB) ID: 3DHS) (Methods) to generate initial simulated data, called BM0 (Extended Data Fig. 6a–c), by dynamic fitting at 7 different levels of imposed Gaussian noise (5, 10, 15, 20, 30, 40 and 50% of the maximum $z$ height of the AFM topographic image) (Fig. 2f). The dynamic fitting was performed using classical Langevin coarse-grained molecular dynamics[2,31] for efficient sampling over broad conformational space restrained by AFM pseudo-potential. The effect of AFM topographic restraints is illustrated in the residue dynamic cross-correlation maps (DCCM), where the AFM-restrained DCCM (Fig. 2g, right) is only a subset of the free (unrestrained) DCCM (Fig. 2g, left), and populations of the best structures (lowest r.m.s.d. relative to 3DHS) decrease with increased noise levels, with the lowest r.m.s.d. ranging from 2.97 (5% noise) to 6.04 Å (50% noise) (Fig. 2h).

## Top structures from UML

Analysis of the r.m.s.d. values of the trajectory models from BM0 clearly indicates that a large number of those models are close to the ground-truth structure (3DHS) but their energies are similar to those of other conformers even though the structures are markedly different from the ground-truth structure. Therefore, when the ground-truth structure is unknown, simple conventional statistics approaches based solely on energetics are insufficient for identifying the structures that are closest to the ground-truth structures underneath the AFM image. Similarly, $CC^{AFM}$ between a molecular surface and a structure[32] alone is neither sufficient nor designed to identify top structure models because a near-perfect $CC^{AFM}$ could also be achieved at the expense of structural integrity and the hierarchical folding principle (overfitting). We used holistic UML by considering a combination of three types of information as input (Fig. 1b, left): (1) energies associated with the primary chemical, secondary and tertiary structures; (2) $CC^{AFM}$ scores; and (3) energy costs associated with AFM topographical restraints (AFM biasing potential). Notably, all of this information is inherent in the structure models and AFM topographic images, and none of it is presumptive.

First, an initial energy filtering is used to remove outliers from the full trajectory (Extended Data Fig. 3 and Methods), followed by two steps of UML: a PCA of all energy terms (Fig. 3a), and a successive clusterizing algorithm that identifies the cluster of models with the lowest energetic distribution (Fig. 3b) according to the native contact (*Go*) and total energies[31] (Methods). The *Go* potential contains information about how well a given model is folded, based on many factors[31]. We stress that the use of the *Go* potential in dynamic fitting does not prevent the AFM biasing potential from sampling structures that are substantially different from the initial structure because the weighing factor for *Go* is set to the lowest value[2]. From the top UML cluster, the cohort of models that exhibit both the lowest energies ($E^{Go}$, $E^{local}$ and $E^{total}$) and highest $CC^{AFM}$ is selected (Fig. 3c and Methods). The UML pipeline iteratively selects the sub-population of models with the lowest r.m.s.d. (Fig. 3d). For BM0, at all tested noise levels, almost all the top models fell within the final UML cohort. Of note, even at the highest noise levels, the average r.m.s.d. of the top 10 selected models was around 5 Å (Extended Data Fig. 4 and Supplementary Table 2), illustrating the usefulness of

topographic restraints over global conformation. As with any structure fitting method, the efficacy of dynamic fitting to AFM topography depends heavily on the initial model. Our benchmarks (BM0–BM5) cover not only different RNAs but also different starting conformations derived using various methods (structure prediction, trajectory model and small-angle X-ray scattering (SAXS) data) (Fig. 1d and Supplementary Table 1). The results show that, for all benchmarks (BM0–BM5), the top 10 lowest-total-energy models from the UML cohort had an average r.m.s.d. of 5 Å relative to their respective ground-truth structures, with a lowest r.m.s.d. of approximately 3.5 Å (Fig. 3e).

## Estimating model accuracy using a DNN

Although UML is capable of selecting top cohorts, it does not provide an estimate of the accuracy of each model relative to the ground-truth structure beneath the AFM topography, which is critical for the determination of unknown structures. Since the accuracy of a recapitulated structure is embedded within the model (energy terms) and the AFM topography ($CC^{AFM}$), in principle, a well-trained DNN is capable of providing accuracy estimation in cases where the ground-truth structure is unknown. Using DNN to estimate the accuracy (confidence level) has been demonstrated in the latest success in protein structure prediction[5], which leverages the abundant structural information available in databases and the sequence–3D structure correlation. As both structural and sequence–structure correlation information is lacking for RNA[41], structure prediction for RNA is far more challenging, especially as RNA molecules become larger and more conformationally heterogeneous because of the geometric and energetic equivalence among conformers and the lack of global restraints[28]. As one RNA sequence may fold into very different conformations, experimental single-molecule data are essential for structure determination of individual conformers.

To overcome the problem of an insufficient pool of experimentally determined RNA structures available, we created a psDatabase containing more than 3.5 million structure models of the RPR catalytic domain: approximately 1.5 million trajectory models from BM0, and approximately 1 million each from BM1 and BM2 (Extended Data Fig. 5 and Methods). These trajectory models cover a continuous conformational space, which is advantageous over the discrete fragment-based structural database[42]. In particular, the difference among the structure models in the psDatabase is as large as 37 Å in terms of r.m.s.d., indicating the broadness of the conformational space. We partitioned 80% of the psDatabase for training the DNN model (training set) and 20% for initial validation and testing (training-validation set) for underfitting and overfitting in regularization processes (Fig. 1d, Extended Data Fig. 5 and Supplementary Table 3). To assess whether the learning by the DNN was generalizable not only to different RNA trajectories but also to different ground-truth structures[43–45], we used 5% of BM5 as our further validation dataset. Such a generalization scheme has been shown to provide a more realistic and robust assessment of the performance than one that includes fractions of data from all benchmark datasets[44].

For the training-validation set, the DNN showed a high correlation between the estimated and true r.m.s.d. values, with a Pearson coefficient of 0.95 (Fig. 4a). The decreasing loss function versus epoch for both the training and training-validation sets shows increased learning up to around 50 epochs (Fig. 4b, blue and red lines), the overall loss profile up to 300 epochs indicates that there is no underfitting or overfitting by the trained model. The epoch exhibiting the smallest loss in the validation set (Fig. 4b, black line) was then used to determine where to stop the training. To evaluate the performance of HORNET for different trajectories and RNAs with different shapes, sizes and sequences, we then tested the full trajectories of BM1–BM5 to illustrate that our holistic DNN is learning generalizable structural 'metrics' of accuracy, not merely memorizing specific structural features (Fig. 4c–g). Critically, the data used for the training and training-validation sets

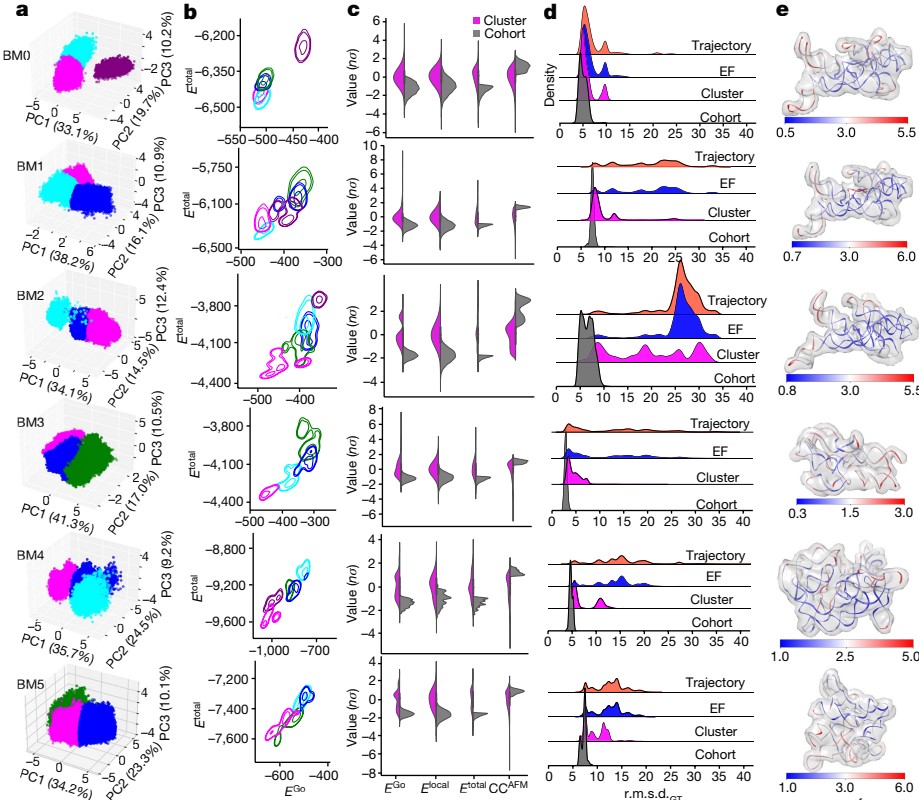

**Fig. 3 | Benchmarks of UML. a**, 3D plots of the first 3 principal components (PC1–PC3) from PCA analysis of BM0–BM5, the three UML clusters from each benchmark with the largest eigenvalue fluctuation. **b**, Contour plots for total ($E^{total}$) and *Go* ($E^{Go}$) energies in kcal/mol for all cluster populations, determined after PCA analysis. The UML cluster with the lowest *Go* energy (magenta) is selected. **c**, Violin plots of energies ($E^{Go}$, $E^{local}$ and $E^{total}$) and $CC^{AFM}$ of the models from the selected cluster (magenta). After applying energy and topology thresholds, the top cohort of models (grey) is selected. **d**, Post-UML analysis of the progression of the selection process in terms of r.m.s.d. in Å from the ground-truth structures in each benchmark case, from the total trajectory population (orange), through the energy filter (blue) and clustering (magenta) to the final cohorts (grey). The vertical axis indicates the density of populations. **e**, Structures from each benchmark with the lowest $E^{total}$, rendered in colour in terms of r.m.s.f. in Å compared with the ground-truth structures.

were omitted in this evaluation, and thus only data that the holistic DNN had never seen were used. In particular, BM3 and BM4 are the two blind tests with different RNAs that our training model has never seen (Fig. 4e,f) and show that the results from our DNN architecture corroborates and cross-validates the results from the UML (Fig. 4h and Supplementary Table 4) and estimate the accuracy of top cohort models (Fig. 4i).

## Validation with different initial models

Having established our DNN procedure, we next tested whether HOR-NET could determine a structure using the same AFM topography but with two different starting conformations. The initial structures of RPR catalytic domain used for BM1(S142) and BM2(S1076) were generated using FARFAR2[46] (Rosetta's fragment assembly of RNA with full-atom refinement) with ARES[28] (atomic rotationally equivariant score) scores of 9.23 and 9.04, respectively (Extended Data Fig. 6e,f). These two initial structures were selected based on the FARFAR2 models pool for having the combination of best ARES and FARFAR2 score among models with r.m.s.d. values relative to the crystal structure (PDB: 2A64)[33], with cut-offs of 10 Å and 20 Å, respectively. The r.m.s.d. between the crystal structure and BM1 and BM2 are 13.5 Å and 22.3 Å, respectively, and the r.m.s.d. between BM1 and BM2 is approximately 18.7 Å, indicating that they have markedly different topologies from each other and from the ground-truth structure (Extended Data Fig. 6c). In each case, both UML and DNN were capable of identifying the top cohorts (Figs. 3d and 4c,d,h). The estimated accuracy of all models

from DNN showed a high correlation with the actual r.m.s.d. values relative to the crystal structure, with Pearson correlations of 0.92 and 0.80 for BM1 and BM2, respectively (Fig. 4c,d). HORNET detects non-convergence by poor r.m.s.d. values (greater than 7 Å for the data presented here) for the top cohort of models. As an example, initial structure S257, the FARFAR2-predicted model that had the best ARES score of 7.72 among 10,000 models generated (Extended Data Fig. 6d), did not converge through dynamic fitting within practical computing time, reflected in low HORNET estimated accuracies (best: 12.4 Å, mean: 27.0 Å) for the entire trajectory of 15.6 million models (Extended Data Fig. 7a,b). S257 is likely to be a structure trapped in a local minimum. We thus performed an unconstrained molecular dynamics simulation to drive the structure out of the local minimum, and then applied UML without AFM-potential terms, after which five models were selected from the top cohort on the basis of their lowest $E^{total}$ (Extended Data Fig. 7c,d) and were used as initial structural models for dynamic fitting; all five trajectories showed convergence toward the ground-truth structure, with the best model from the DNN-selected cohort having an r.m.s.d. of 3.6 Å from the crystal structure (Extended Data Fig. 7e,f).

## Validation using data from a smaller RNA

The 210-nt cobalamin-sensing riboswitch (rCbl (BM3)) is capable of folding into heterogeneous conformations[2]. This RNA has a completely different sequence, shape, size and fold from the benchmarks used for training and validation. Thus, BM3 serves as a blind test for

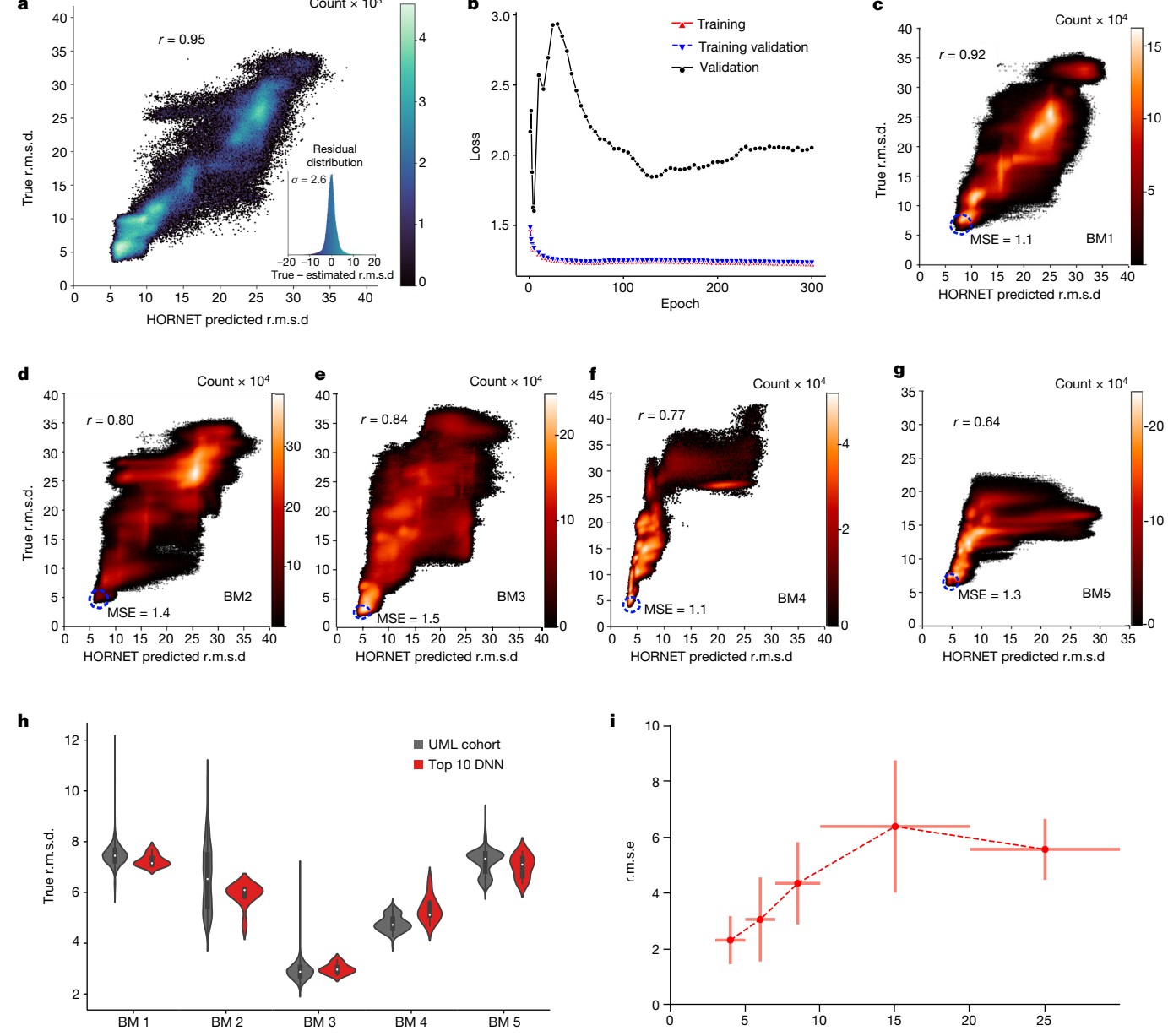

**Fig. 4 | Supervised DNN. a,** True versus estimated r.m.s.d. (*r* = 0.95) for the training-validation set. Inset, histogram of the residuals (true r.m.s.d. − predicted r.m.s.d.) with *σ* = 2.6 Å. **b,** Loss function versus training epoch for the training, training-validation and validation sets. The loss was evaluated after the end of each training epoch. **c**–**g,** Tests using the full trajectories from BM1 (**c**), BM2 (**d**), BM3 (**e**), BM4 (**f**) and BM5 (**g**) (excluding training and training-validation data), the mean square error (MSE) for the cohort of best models are highlighted in a blue dashed circle in the lower left corners. Approximate number of trajectory models: BM1, 10 million; BM2, 11 million; BM3, 7 million; BM4, 2 million; BM5, 13 million. **h,** Violin plots comparing the true r.m.s.d. values of the models from the UML-selected cohort with *n* = 7,450, 11,927,

10,867, 93 and 12,771 structure models, and the top 10 models from DNN (*n* = 10) for each benchmark. Maximum, mean and minimum values of the plots for UML and DNN, respectively: BM1 (12.0, 7.4, 5.7) and (7.8, 7.3, 6.6); BM2 (10.8, 6.5, 4.1) and (6.1, 5.5, 4.1); BM3 (7.2, 2.9, 1.8) and (3.2, 2.7, 2.3); BM4 (5.5, 4.8, 3.9) and (5.3, 4.8, 4.4); BM5 (9.3, 7.2, 5.8) and (7.4, 6.5, 6.2). **i,** Uncertainty of estimated accuracy, provided as root-mean-square error (r.m.s.e.), across random samples of all tests set of benchmarks. Dots indicate the average r.m.s.e. and vertical lines are s.d. for five bins (range indicated by horizontal lines) of estimated accuracy: 3–5 Å, 5–7 Å, 7–10 Å, 10–20 Å and 20–30 Å, *n* = 5.4 million total independent structures.

further demonstrating the capability of HORNET. The initial model for BM3 is from a coarse-grained molecular dynamics trajectory and has an r.m.s.d. of 10.2 Å from the ground-truth structure (PDB: 4GMA)[47] (Extended Data Fig. 8a). After applying HORNET, the UML and DNN cohorts of selected models showed similar true r.m.s.d. values (average approximately 3 Å) from the ground-truth structure (Fig. 4h), and the DNN model showed good performance, with a Pearson score of 0.84 for the whole trajectory (Fig. 4e).

## Validation using SAXS-derived models

BM4 and BM5 were selected to further demonstrate HORNET using initial models generated from low-resolution experimental techniques. For this purpose, we used RS3D-derived models from simulated SAXS data[48] of group II intron (BM4, 395 nt) and RPR (BM5, 298 nt), which are 16.1 Å and 14.0 Å r.m.s.d., respectively, from their ground-truth crystal structures[33,49] (Extended Data Fig. 8b,c). The selected cohorts of models

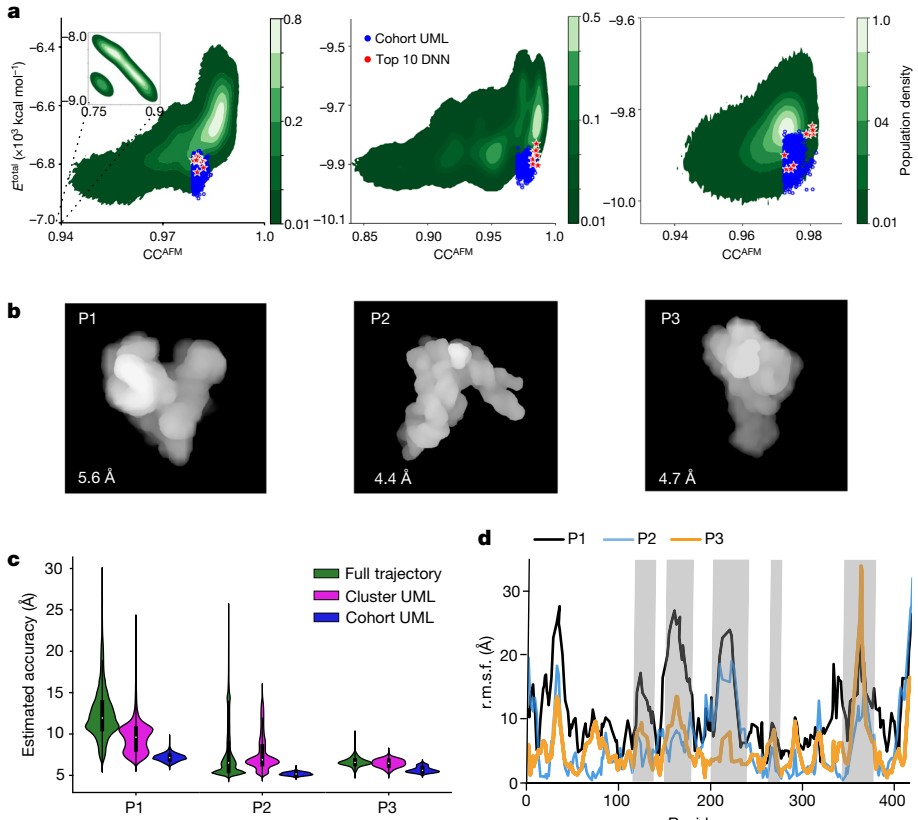

**Fig. 5 | Recapitulating three topological structures from experimental AFM images of three RPR particles. a**, Contour maps of the total energy and CC^AFM. The green colour gradient indicates the population density and symbols indicate the cohort models selected by UML (blue) and the top 10 models scored by DNN (red) for P1 (left, 4 million models), P2 (middle, 4 million models) and P3 (right, 2 million models). **b**, The top selected models rendered in molecular surface mode. **c**, Accuracy (r.m.s.d.) estimated by DNN prediction for the full trajectory (green) with the sample size described in **a**, and evaluation for the selected cluster and cohort from UML for P1, P2 and P3. **d**, r.m.s.f. profiles between the crystal model and the recapitulated models of P1, P2 and P3. Residues with missing electron density in the crystal structure are highlighted in grey.

from HORNET UML had minimum r.m.s.d. values of 3.8 Å for BM4 and 5.4 Å for BM5 (Fig. 3d), with Pearson coefficients of 0.77 and 0.64 for BM4 and BM5, respectively (Fig. 4f,g).

Considering the results from UML and DNN, which were performed independently for all benchmarks, we conclude that the DNN model alone (without UML) is sufficient for determining the structural model and reliably estimating the accuracy of that structure in terms of r.m.s.d. However, the estimated accuracy becomes less certain beyond about 7.5 Å r.m.s.d. (Fig. 4i), which may be explained by energetic, topographic and geometric equivalency among conformers, the under-representative psDatabase in this r.m.s.d. range, or both.

## Structures of heterogeneous conformers

Studies using chemical probing and SAXS show that large conformational changes in RPR, which can be as large[16,21,50,51] as 30 Å, occur via the motions of individual helical structural elements without disruption of base-pairing interactions[16,50]. The conformational flexibility may account for the substrate promiscuity of RNase P[16]. Only a partial structure (PDB: 2A64) comprising 298 residues of the 417-nt full-length RPR has been determined using crystallography[33].

HORNET was applied to the AFM particles (P1, P2 and P3) of the full-length RPR (Fig. 2a,b), which showed three conformations that are highly distinct from the crystal structure. The dynamic fitting trajectories for P1, P2 and P3 showed significantly different conformational landscapes (Fig. 5a and Extended Data Fig. 9). P1 samples a vast range of atomic displacement, whereas P3 shows an intermediate sampling

and P2 shows the most restricted displacements, probably owing to the presence of more short-distance information in the P2 AFM image (Fig. 2e) relative to the others. After HORNET UML and DNN, the best recapitulated models for P1, P2 and P3 have estimated accuracies of 4–6 Å r.m.s.d. (Fig. 5b, Supplementary Tables 5 and 6). Of note, the UML- and DNN-selected models exhibited similar ranges in energies and CC^AFM (Fig. 5c and Supplementary Table 6). The estimated accuracy for P1 showed a very wide range in r.m.s.d. from 5.6–31 Å, whereas P2 and P3 showed lower and much narrower distributions of 4.4–27 Å and 4.7–12 Å, respectively (Fig. 5c). This shows that the trained model is capable of identifying and scoring different ranges of atomic displacements. The root-mean-square fluctuation (r.m.s.f.) per residue for the trajectories of P1, P2 and P3 show that the largest fluctuations occur in regions where the crystal structure could not be modelled owing to insufficient electron density (Fig. 5d).

## A few conformers of the HIV-1 RRE RNA

Many functional RNAs are highly dynamic and do not adopt single stable structures under physiological conditions. The HIV-1 RRE RNA is one such RNA. Nuclear export of unspliced and singly spliced viral transcripts is a critical step in the HIV-1 viral replication cycle, and the RRE RNA is key for the virus to distinguish and select its own RNA from the more abundant host RNAs for export. Although an averaged molecular envelope of RRE has been derived on the basis of SAXS data[52], no high-resolution structure of RRE in any conformational state has been determined. Thus, the structure of RRE and the binding mode between

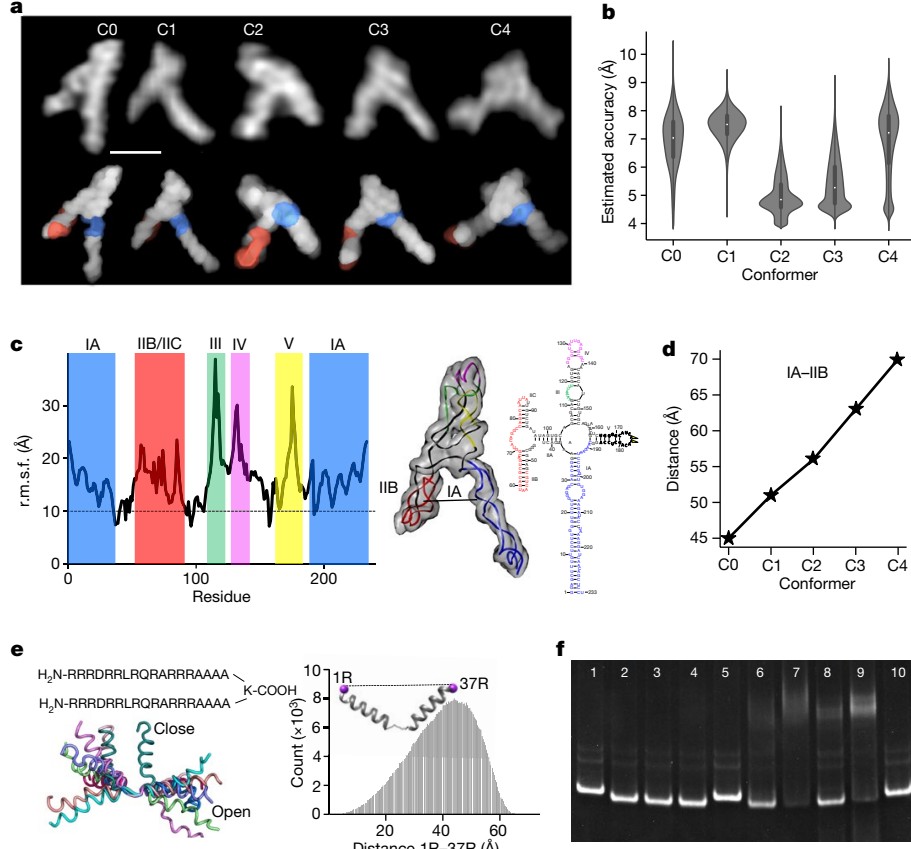

**Fig. 6 | Topological structures of the RRE RNA conformers. a**, Five RRE molecules in five distinct conformational states, C0–C4, captured by AFM (top), and their recapitulated structures (bottom). The raw single-molecule images were taken once, thus without signal averaging. The red and blue coloured regions indicate the locations of the IIB and IA Rev-binding sites. See also Supplementary Videos 1–5. Scale bar, 10 nm. **b**, Violin plots showing the estimated accuracy by HORNET for each of the five conformers (Extended Data Fig. 10a) with minimum values of 3.0, 4.3, 3.1, 3.7, 3.0 Å; maximum values of 10.4, 9.4, 8.1, 9.2 and 9.7 Å; mean values of 6.9, 7.5, 4.8, 5.4 and 6.9 Å; and $n = 2{,}809{,}043$, $1{,}242{,}907$, $1{,}389{,}161$, $2{,}101{,}611$ and $1{,}388{,}409$ structural models for C0, C1, C2, C3 and C4, respectively. **c**, r.m.s.f. versus residue among the five conformers (left) and the 3D model (C0) in molecular surface superimposed with ribbon diagram (middle), and the RRE secondary structure (right). The residue regions with variations above 10 Å (r.m.s.f. plot, dotted line)

are coloured accordingly. **d**, Distance variation between the two known Rev-binding sites, IIB and IA, for each conformer. **e**, Design scheme of a branched peptide (top left), with single-bond lysine linkage, offering flexibility to adapt to the conformationally heterogeneous RRE (bottom left), and the end-to-end distance distribution between the two parallel ARMs simulated by a 3.0-μs molecular dynamics calculation (right). **f**, Electrophoretic mobility shift assay of branched peptide–RRE complexes. Lanes 1, 5 and 10: RRE; The ratios of RRE:peptide:Rev:RibA71 in lanes 2–4 and 6–9 are: 1:4:4:0, 1:4:8:0, 1:16:0 and 1:4:32:0, 1:4:64:0, 1:4:32:150 and 1:4:64:150, respectively. RibA71 comprises three duplexes, and is an adenine riboswitch RNA used as non-specific RNA competing with RRE for peptide binding. The feint and smeared bands in lanes 6–9 are oligomeric Rev–RRE complexes. Gel source data are shown in Supplementary Fig. 1.

RRE and Rev protein have remained the subject of intense debate. We have long suspected RRE to be conformationally heterogeneous, which limits applicability of crystallography and cryo-electron microscopy. The conformational heterogeneity of RRE poses a fundamental question about how the virus specifically recognizes RRE when both the RNA[52] and the Rev protein dimer are flexible[53,54]. We applied HORNET to five RRE conformers observed by AFM to ultimately demonstrate its capability to determine 3D models of an RNA whose structure is unknown and very different from any RNAs used for benchmarking (Fig. 6a,b, Extended Data Fig. 10a and Supplementary Table 7). The visualization by AFM shows that RRE folds into various A-like shapes (Fig. 6a and Extended Data Fig. 10b–d), which validates the average molecular envelope derived indirectly from SAXS data[52]. Furthermore, in all five conformers, the two known Rev binding sites face each other with inter-site distances varying between 45 and 70 Å (Fig. 6c,d and Supplementary Videos 1–5), further demonstrating RRE conformational heterogeneity and flexibility. In addition, the region showing the largest conformational flexibility and heterogeneity is between residues

100 and 190 in domains III–V (Fig. 6c). This region has been implicated in the resistance to disruption of viral export by the *trans*-dominant negative RevM10 by adopting an alternate conformation[55]. Given the conformational heterogeneity, especially with regard to the inter-site distances, we then designed a class of novel branched peptides mimicking the Rev dimer[52,53]. One such compound consists of two arginine-rich motifs (ARMs) joined in parallel through the main and side chains of a lysine (Fig. 6e). The linkage through the eight single bonds of the lysine ensures maximum flexibility and allows the two parallel ARMs to sample a wide range of interhelical angles and inter-site distances (Fig. 6e). We believe that the two parallel ARMs complement the topological arrangement of the two facing binding sites in RRE, and at the same time, the conformational flexibility of both RRE and peptide may lead to mutual adaptability and best fit. The topological complementarity and mutual adaptability may result in the compound being capable of out-competing the cognitive Rev with very high specificity. Indeed, the branched peptide is able to bind RRE with high specificity and affinity even in the presence of multiple-fold excess concentrations of Rev

and/or a non-specific RNA (Fig. 6f and Supplementary Fig. 1). The resulting complexes migrate faster in electrophoretic mobility shift assays, indicating less flexible and more compact structures, consistent with the AFM images of complex particles (Extended Data Fig. 10c,d) and the structure–dynamics results from HORNET (Fig. 6c,d). The direct visualization of the A-like shape unambiguously resolves the long-standing debate surrounding the RRE topological structure. Furthermore, the architectural complementarity and mutual conformational adaptability may explain how the HIV-1 virus recognizes RRE specifically despite its flexible and heterogeneous conformations.

## Discussion

HORNET addresses major challenges in studying topological structures of highly heterogeneous and flexible RNA molecules by obviating the dependency on signal averaging, the common approach in NMR, crystallography and cryo-EM. Although our method complements the existing high-resolution methods, the ability to recapitulate topological structures from AFM images of individual RNA conformers could markedly expand our knowledge of the heretofore uncharted RNA 3D conformational space, far beyond the few snapshots of static structures in databases. Given the abundance of structural elements in RNA, HORNET has the potential to accelerate our understanding of the conformational space of large RNAs with known biological significance, as illustrated in the HIV-1 RRE study. The topological complementarity and mutual conformational adaptability found in this study may be one of the general mechanisms that drive the RRE–Rev interaction in terms of both specificity and affinity. Furthermore, estimating the accuracy of an unknown conformer structure is a grand challenge in structural biology[56]. Recent success in protein[5] and progress in RNA structure prediction[28] is highly encouraging. Because of the conformational heterogeneity of RNA, methods such as HORNET that incorporate individual conformer-specific topographic global restraints are a viable approach to studying the conformational landscape of flexible RNA. Given a DNN model trained with a sufficient structure database that covers broad RNA conformational space, combined with experimental topographic information and secondary structural information, HORNET is capable of generating low-resolution topological structures of individual large RNA conformers in solution.

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

## Methods

### RNA structure calculation applying topographic restraint

Given that duplexes are well-conserved and they are the predominant building blocks in folded RNA structures by far[36], they can be considered semi-rigid bodies within a folded RNA structure. Since they are covalently connected, these duplexes can be treated as kinematic chains. Adding kinematic constraints between rigid bodies will significantly decrease the degrees of freedom of a rigid body system[57], and imposing the topographic constraints in addition to the kinematic constraints further reduces the degrees of freedom of sampling space.

A high-resolution AFM image is more than just a 'frame' of a molecule. The width and pitch of an A-form RNA duplex are ~25 and 30 Å, respectively, which are on a similar scale to a sharp AFM probe and sensitive to detection. Thus, given an achievable imaging resolution of 10–15 Å[34] (Fig. 2a,b), major structural features such as grooves and pitches of long duplexes, along with molecular shapes and topological folds of larger structured RNAs in solution, are discernable in high-resolution AFM images[58–61]. Thus, a high-resolution AFM image of a molecule is a 3D frame with details about topographic information on individual molecules. The explicit expression of the physical relationship between a molecular structure and the topographic molecular surface is defined. A detailed description of the implementation is provided in Supplementary Information.

### Unsupervised machine learning

Our UML approach assumes that the classical molecular dynamics simulation guided by topographic information can sample the real native conformational space of the RNA, and that the correct models can be identified based on the established hierarchical folding principle[62], energetics[63] and agreement with topographic restraints. Our UML algorithm is able to decipher the underlying correlation of the dataset, resulting in the recognition of generalizable models without pre-training or data labelling. Each analysed dataset (trajectory) is unique, and the machine does not have any expected pre-labelled output from a given input. Our UML algorithm consists of three main steps: (1) energy filtering; (2) PCA and clustering; and (3) cohort model selection. A detailed description of the UML is provided in Supplementary Information.

### Supervised DNN

Based on the question of whether the most fundamental characteristics of models such as their energetics and known topology of a structure contained in the AFM experimental data would provide enough information to consistently determine the r.m.s.d. between the structural model and an unknown ground-truth structure, we designed a DNN[64] to explore how these fundamental characteristics could be intrinsically correlated. A detailed description of the supervised DNN is provided in Supplementary Information.

### Underfitting and overfitting

To avoid overfitting and to be able to keep increasing the complexity of our ANN, we added regularization penalties to the training. Within the known regularizers, we evaluated training using ridge regression (L2 regularizer) and the dropout technique. Ridge regression adds a penalty to the loss function term for all the weights squared, preventing the weights from assuming excessively high values. In the dropout technique, in each step of the training optimization, some neurons have a given (set) chance to be turned off. We also tested increasing the size and variety of the dataset by adding more data (trajectories) (Supplementary Table 3).

### Optimized architecture

To train the DNN and assess its performance, we split dataset BM0 (which contains only one kind of RNA) into two parts: the training, and the initial training test set (called 'training validation'), where we could check the regularization effect over the same trajectory and assure that the regularization was blocking the train from overfitting the trajectory over the split, thus providing similar loss on both sets. The training set had 80% of the 3.5 million trajectory models, while the training-validation set had 20%. The optimized training dataset that yielded the best performance was built using all data from the BM0, with an additional 5% of data from each of the trajectories of BM1 and BM2 (Extended Data Fig. 5).

The validation set was created by using a different RNA trajectory simulation, BM5, so that the best loss on the validation set would point to the place where the training and learning with a given RNA was still generalized to another RNA trajectory, applying early stopping on the evaluated loss considering r.m.s.d. values up to 10 Å to weight a better performance on smaller r.m.s.d. values than larger ones. Hence, the validation set was used for both tuning the hyperparameters and for selecting the best-trained model, while further tests over the benchmarks address if our model can generalize its findings and learnings to other RNAs not contained in the training data, with different RNA sizes and folds, assessing what would be the real performance of our model to other unknown RNAs and trajectories than the one used for training.

We optimized the architecture for this work by many step-by-step random searches and subsequent fine-tuning of the hyperparameters, which include the number of layers, the number of neurons per layer, weight initialization, neuron activations, regularization penalties and types, the optimizer algorithm as well as the learning rate. Additionally, more than 50 different compositions (data, kappa and noise) of the training dataset were used for training the models (Supplementary Table 3). The number of hidden layers tested (also by a random search) was between one and ten hidden layers. The number of neurons in each layer, on the other hand, was tested basically in 3 types: (1) starting with a high number of neurons in the first layer and decreasing this as the number of layers increases; (2) starting with a medium number of neurons in the first layer, and increasing the number of neurons on the next layers until reaching the middle layer, then decreasing as we continue to the last layer; and (3) through a random search, where the number of neurons per layer was picked randomly as a multiple of 8, being able to assume values from 8 to 256 neurons per layer. For architectures with five or more layers, we included batch normalization within layers.

The non-linear activations tested were relu, leaky-relu, elu and gelu for each layer separately, or a selu[65] activation set for all layers. For regularization, each layer could use either the ridge regression and/or a dropout[66] chance (for selu the alpha dropout[65] was used instead of Dropout to keep the self-normalizing properties). Our optimized architecture has only 3 hidden layers with a decreasing number of neurons, 128 in the first layer, 64 in the second, and 16 in the third, using elu activation with a common dropout rate of around 20% as the regularizing agent. Deeper networks also had a good performance, but with the cost of many weights to train without clear improvement. The total number of trainable parameters with the current architecture is around 11k. Within initializations, we tested Glorot uniform, Lecun normal and He normal, with the latter achieving the best performance as the weight initializer and using Adam as the optimizer algorithm with a standard learning rate of 0.001, with the mini-batch size of 128 and using Huber loss.

The models were trained using NIH-HPC (Biowulf) k80/k100x nodes: K100x node: 36 × 2.3 GHz (Intel Gold 6140), hyperthreading, 25 MB secondary cache, 4 x NVIDIA V100-SXM2 GPUs (32 GB VRAM, 5120 cores, 640 Tensor cores); K80 node: 28 × 2.4 GHz (Intel E5-2680v4), hyperthreading, 35 MB secondary cache, 2 x NVIDIA K80 GPUs with 2 x GK210 GPUs each (24 GB VRAM, 4992 cores).

### RNA sample preparation

**RPR.** The RPR was prepared as described[33]. In brief, the RPR was transcribed in vitro with recombinant T7 phage RNA polymerase from a double-strand DNA template that was amplified by PCR from linearized

DNA plasmid, which encodes a full-length RPR from *B. stearothermophilus* with an upstream T7 RNA polymerase promoter. Transcribed RNA was purified by denaturing polyacrylamide gel electrophoresis containing tris-borate with EDTA (TBE) and 8 M urea. The RNA was excised and eluted from the gel in RNA elution buffer (300 mM Sodium acetate pH 5.3, 0.1 mM EDTA) for 12 h at 4 °C. The eluted RNA was filtered using a 0.2-µm Ultrafree-MC centrifugal filter device (Millipore). Purified RNA was subjected to several buffer exchanges using a Centricon unit (Millipore) with 30 kDa molecular weight cut-off membrane against refolding buffer (50 mM MES buffer pH 6.8, 100 mM KCl, 1 mM MgCl$_2$), then concentrated to 2 µM, aliquoted, and stored at −80 °C before utilization.

For AFM experiments, the RNA sample at 2 µM concentration was annealed in the refolding buffer (50 mM MES buffer pH 6.8, 100 mM KCl, 10 mM MgCl$_2$) at 65 °C for 2 min followed by stepwise cooling to 37 °C over 30 min, and then kept at 4 °C before AFM measurements. To dilute the RNA sample to the required concentration (20 nM) for AFM, 1:100 volume of low-salt buffer (50 mM MES buffer pH 6.8, 10 mM KCl, 1 mM MgCl$_2$ (preequilibrated at 4 °C) was used, and the diluted sample was immediately deposited onto mica pre-treated with 1-(3-aminopropyl) silatrane (APS) for immobilization[26]. The functionalization of mica with APS is widely used for the nondisruptive immobilization of nucleic acids primarily through the electrostatic interactions between protonated amino groups of the APS-mica substrate and the negatively charged nucleic acid backbone.

**RRE RNA.** RRE sample was prepared following the same protocol described previously in detail[52]. The fresh sample was used for the AFM experiments with a concentration of 20 nM in 50 mM MES buffer pH 6.8, 10 mM KCl, 1 mM MgCl$_2$. The sample was loaded on a freshly cleaved mica pre-treated with APS and incubated for 30 min before imaging.

## Peptide design, synthesis of P46 and modelling
The two ARMs (H2N-RRRDRRLRQRARRRAAAA-COOH) are joined by the amino groups of a lysine main and side chains via chemical synthesis (Shengnuo). This compound is patented under US Patent Number 10,464,970.

The monomeric ARM structural model was built using Pep-Fold[67]. Then, two ARM structural models were linked in parallel using the bond build function of PyMol (PyMol Molecular Graphics System, version 2.0 Schrödinger). A ~3.0-µs coarse-grained molecular dynamics simulation using CafeMol[31,32] was performed to obtain the distance distribution between the 2 ARMs. The molecular dynamics trajectory was generated applying constant temperature simulation of 300 K Langevin dynamics and *Go* model for a total of 60 × 10$^6$ steps.

## Electrophoretic mobility shift assay
RRE (1 µM) was mixed with various ratios of P46 and Rev protein in a buffer containing 10 mM HEPES (pH 7.5), 300 mM KCl, 1 mM MgCl$_2$, 0.5 mM EDTA, 0.1 µg µl$^{-1}$ BSA, 0.2 µl SUPERase•In RNase Inhibitor (Thermo Fisher Scientific). The total reaction volume was 10 µl. The reactions were incubated at room temperature for 30 min, then 4 µl of each reaction was loaded into a Novex 6% TBE gel (Thermo Fisher Scientific). The gel was run for 80 min at 120 V, and the image was taken using a Gel Doc EZ Imager (Bio-Rad) after staining with SYBR Gold Nucleic Acid Gel Stain (Thermo Fisher Scientific). Adenine riboswitch (RibA71) RNA (150 µM), consisting of three helices, was used for competitive non-specific binding via peptide–major groove interactions. Samples of the Rev protein and RibA71 were prepared as reported previously[53,68]. The uncropped gel image file is provided in Supplementary Fig. 1.

## AFM experiments and image processing
**Experimental AFM image acquisition.** The detailed procedure for the AFM image acquisition is described elsewhere[2]. Full-length RPR particle images, P1, P2 and P3 (Extended Data Fig. 1), were recorded under the solution conditions described above using a Cypher VRS AFM (Asylum Research, Oxford Instrument) at 4 °C with amplitude-modulated AC mode at a scan rate of 1 Hz (commonly known as tapping mode) using FASTSCAN-D-SS probes (Bruker). For RNA immobilization, 50 mM APS stock was freshly diluted 300-fold in deionized water right before use and then used to coat a freshly cleaved muscovite mica (Grade V1) (Ted Pella) and incubated for 30 min, followed by rinsing the mica surface with deionized water and drying gently with filtered nitrogen gas.

**AFM noise estimation.** For quantification of the noise present in the *z* coordinates, we used the cropped single molecule from the full recorded AFM topography as input, and the *z* values were collected for defined *x* and *y* coordinates of the 'empty' area around the molecule (Extended Data Fig. 2a–d). The *z*-coordinate values of the empty horizontal and vertical spaces can be described by a normal function, where the mean value of this distribution represents the mean noise and the uncertainty as the standard deviation (sigma). The mean noise value and uncertainty were evaluated for P1, P2 and P3 before and after image processing (Extended Data Fig. 2a–d). In this analysis, we are considering all the experimental sources that result in noise randomly distributed over all recorded data as a background signal.

**AFM image resolution estimation.** The topography resolution assessment was performed using an ACV approach[34]. There are two principal steps to be performed in this method. First, using the processed image, we calculate the 2D FFT of the AFM image and a defined ring-size (in pixels) cut-off filter is applied to select a portion of the image in Fourier space (Extended Data Fig. 2e–g). Afterwards, the image is back-calculated to real space; this step is described as a low-pass filtered Fourier ring. In the second step, we calculate the ACV between the original image (R) and the resulting one from the inverse fast Fourier transform for each of the low-pass filtered rings (R′). The comparison between the original image with its resulting image from the low-pass filter is performed using the auto-correlation equation (Supplementary Information). A loop interaction was applied starting from low to high frequency, where the ACV starts from low correlation values up to values near to 1.0 where the low-pass cut-off is close to the particle dimension in real space. In Fig. 2, we demonstrate some intermediate steps of the Fourier ring filter applied to P1, P2 and P3 particles. The formula for estimation of ACV value is provided in Supplementary Information.

**AFM image processing.** The detailed procedure for AFM image processing is described elsewhere[2]. In brief, raw images were first processed using SPIP (Scanning Probe Image Processor) software (https://www.imagemet.com/products_/spip/): plane levelling to the particle-free region by applying third-order polynomial, followed by spike filtering to remove artefact streaks, and fast Fourier transform to remove high-frequency noise (Extended Data Fig. 2). The final image resolution was increased to 4,096 × 4,096 pixels by doubling the number of pixels twice. Single-particle images were cropped from the processed images and converted to pseudo-AFM with a digital resolution of 5-Å per pixel in MountainsSPIP software (https://www.imagemet.com/products_/spip/) for structure calculation.

## Benchmark information and design
**BM0.** BM0 was designed to provide the bulk of the training data by using a representative RNA with a known structure at an acceptable resolution. For this purpose, we chose the crystal structure of the RPR catalytic domain (PDB: 3DHS)[40]. The residues that were missing in 3DHS due to insufficient electron density were modelled using SimRNA[69] and further refined using Coot[70]. This model, representing the ground-truth structure for BM0, was subjected to coarse-grained dynamic fitting in CafeMol[31,32] to an experimental AFM image of this RNA,

and the trajectory of models was scored using ARES[28]. The model with the best ARES score of 9.9 from this pool of models, named k158597 (Extended Data Fig. 6a,b), was then used as the initial model for BM0. The r.m.s.d. between k158597 and the ground-truth structure is 21.4 Å. AFM images of the ground-truth model were calculated using a resolution of 5.0 Å per pixel, with 7 different simulated Gaussian noise levels added—that is, 5, 10, 15, 20, 30, 40 and 50% of the maximum z height (Fig. 2f). The dynamic fitting using k158597 as the initial model and the AFM topography of the ground-truth structure was performed for a total of $20 \times 10^6$ steps (~0.9 μs) for all noise levels (Fig. 2h).

**BM1 and BM2.** BM1 and BM2 were designed to tackle cases in which only the primary sequence and secondary structure information may be known, and the starting model must be predicted. For this task, we first applied the FARFAR2[46] - rna_denovo application, generating 10000 structure models of RPR catalytic domain, using the primary sequence, secondary structure, and atom pair distance constraints of the well-known loop interaction L15.1–L5.1 described in detail previously[71] (Supplementary Table 10). For structure refinement the minimize_rna function was applied as a potential during the FARFAR2 structure prediction run, using parallel jobs on a 28-core 2.3 GHz x2695 processor.

The FARFAR2 scoring function was calculated for all the predicted models and analysed as a function of the main energy terms. The sampled refined structures show a range of r.m.s.d. with a maximum of 46 Å and a minimum of 14 Å from the crystal model (PDB: 3DHS). We selected three models from all predicted structures from FARFAR2 using the following criteria: one model with the best ARES score (S257), one being located in the region of both ARES and FARFAR2 lowest scores and an r.m.s.d. from the ground-truth structure of at least 20 Å (S1076), and one model with the lowest r.m.s.d. (S142), (Extended Data Fig. 6).

ARES selected model S257 as the best model from the FARFAR2 ensemble, a model that presents dramatically different folds from the crystal model, with an r.m.s.d. of ~30 Å (Extended Data Fig. 6d). Using an r.m.s.d. threshold of 20 Å and scoring the models using the energetic scoring function of FARFAR2 and final score of ARES, the best model was S1076 with an r.m.s.d. of 22.0 Å, where this model shows a folding similar to the crystal model (Extended Data Fig. 6e).

**BM3.** We applied our method to the adenosylcobalamin riboswitch (Cbl) using the crystal structure (PDB: 4GMA), which has a folding and size (210 nt) different from the RNA used in training (BM0–B2). The structure calculation was performed with a total of $20 \times 10^6$ steps (~0.9 μs) using an AFM topography generated with 5 Å per pixel (Extended Data Fig. 8a). The final trajectory consisted of ~6.6 million models, which were analysed using UML and DNN (Figs. 3, 4).

**BM4 and BM5.** These two benchmarks were designed to test our method using initial models determined using low-resolution experimental data. In this case, we used the topological structures of RPR (298 nt) and group II intron (387 nt), generated by RS3D[48,72] using secondary structure information and SAXS data simulated from their respective crystal structures, PDB: 2A64[33] and PDB: 4E8K[49]. The ground-truth AFM images were calculated using the crystal models for the RNAs with a resolution of 5 Å per pixel (Extended Data Fig. 8b,c). The structure determination in each case was performed using a trajectory with a total of $60 \times 10^6$ steps (~2.7 μs). The final trajectories consisted of ~13.4 million models, which were analysed using UML and DNN (Figs. 3 and 4).

## Reporting summary

Further information on research design is available in the Nature Portfolio Reporting Summary linked to this article.

## Data availability

Data for the calculations are available at the National Cancer Institute institutional data site: BM0, BM1, BM2, BM3, BM4 and BM5 energy information, https://home.ccr.cancer.gov/csb/pnai/data/HorNet/Energies_For_Benchmarks/; BM0, BM1, BM2, BM3, BM4 and BM5 pdbs: https://home.ccr.cancer.gov/csb/pnai/data/HorNet/pdbs_For_Bench-Marks.tar.gz; BM0, BM1, BM2, BM3, BM4 and BM5 AFM images, noisy data, and initial structures, https://home.ccr.cancer.gov/csb/pnai/data/HorNet/AFM_For_BenchMarks.tar.gz; P1, P2 and P3 experimental data and structures, https://home.ccr.cancer.gov/csb/pnai/data/HorNet/P1_P2_P3_data.tar.gz; RRE experimental data and analysis files with top selected structures, https://home.ccr.cancer.gov/csb/pnai/data/HorNet/RRE_data_results.tar.gz.

## Code availability

All software and scripts are available at Zenodo (https://zenodo.org/records/10637777)[30].

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

**Acknowledgements** The authors thank R. J. L. Townshend and R. O. Dror for assistance in using ARES; NIH-HPC staff for computing resources; and V. Dwivedi and A. S. Prado for testing the HORNET software. This work was supported by the fund for the NIH Intramural Research Program (Y.-X.W., J.Z., A.R. and P.T.W.) and by NCI contract no. 75N91019D00024 (W.F.H.).

**Author contributions** Conceptualization and peptide design: Y.-X.W. Methodology: Y.-X.W., M.S.F.D. and H.F.D. AFM data acquisition: J.D., Y.-T.L., M.S.F.D. and W.F.H. Machine learning: M.S.F.D. and H.F.D. Software: M.S.F.D., H.F.D., Y.R.B. and C.D.S. Sample preparation: Y.-T.L., J.D., P.Y., N.R.W. and P.T.W. Testing software: P.Y. and J.R.S. Writing, original draft: Y.-X.W., M.S.F.D. and H.F.D. Writing, revision and editing: all authors

**Competing interests** The branched peptide used in this study is patented (US Patent no. 10,464,970) (Y.-X.W.). The other authors declare no competing interests.

**Additional information**
**Correspondence and requests for materials** should be addressed to Yun-Xing Wang.

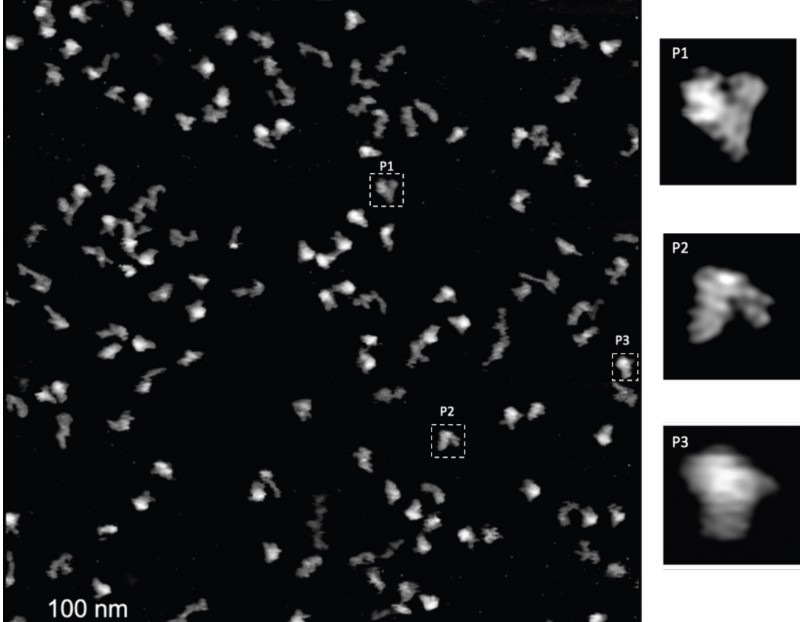

**Extended Data Fig. 1 | Experimental AFM image recorded for the full-length RNase P RNA.** Three particles (P1, P2 and P3) were selected for single-molecule structure determination using HORNET. The raw single-molecule images were taken once thus without signal-averaging. Right panels are enlarged views of the boxed particles in the AFM image.

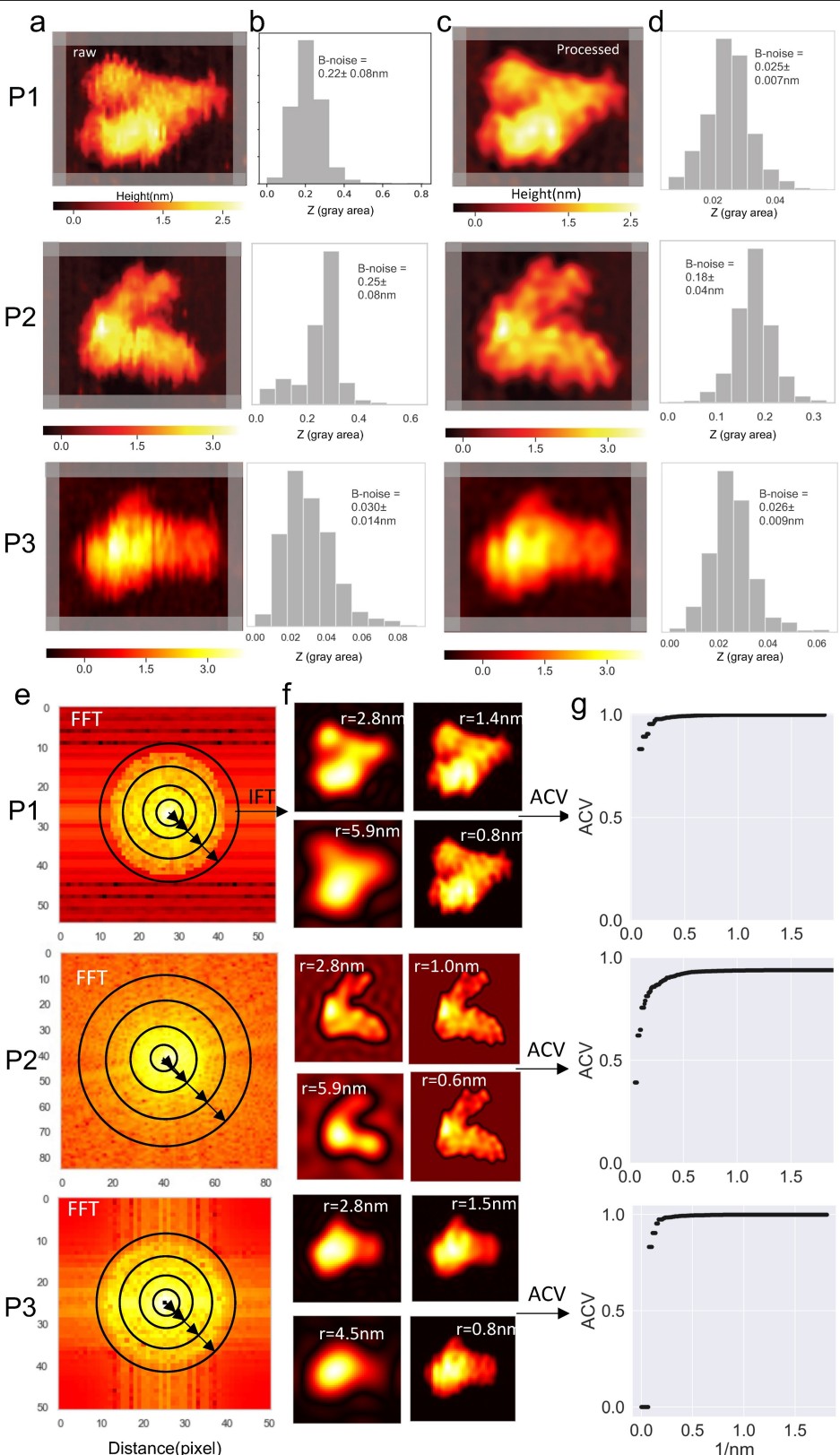

**Extended Data Fig. 2 | Noise and Resolution estimation for AFM-imaged particles of RNase P RNA. a**, Cropped raw AFM images of the selected particles: P1, P2 and P3. The empty area around the perimeter of each cropped particle (gray) was used to calculate the background noise. **b**, Noise assessment (z-value distribution) of the gray areas in the raw images. **c**, AFM images after processing (leveling, spike-filtering, FFT-filtering). **d**, Noise assessment of the gray areas in the processed images. **e**, FFT images of the three particles (P1, P2 and P3) after image processing. A low-pass Fourier filter was applied as concentric rings (black) spaced in 5-pixel increments. **f**, Inverse FFT (IFFT) images, in real space, of the Fourier-filtered particles at each respective Fourier-ring cutoff radius (r). **g**, Auto-correlation value (ACV) profiles, in Fourier space, after the applied low-pass cut-off filters.

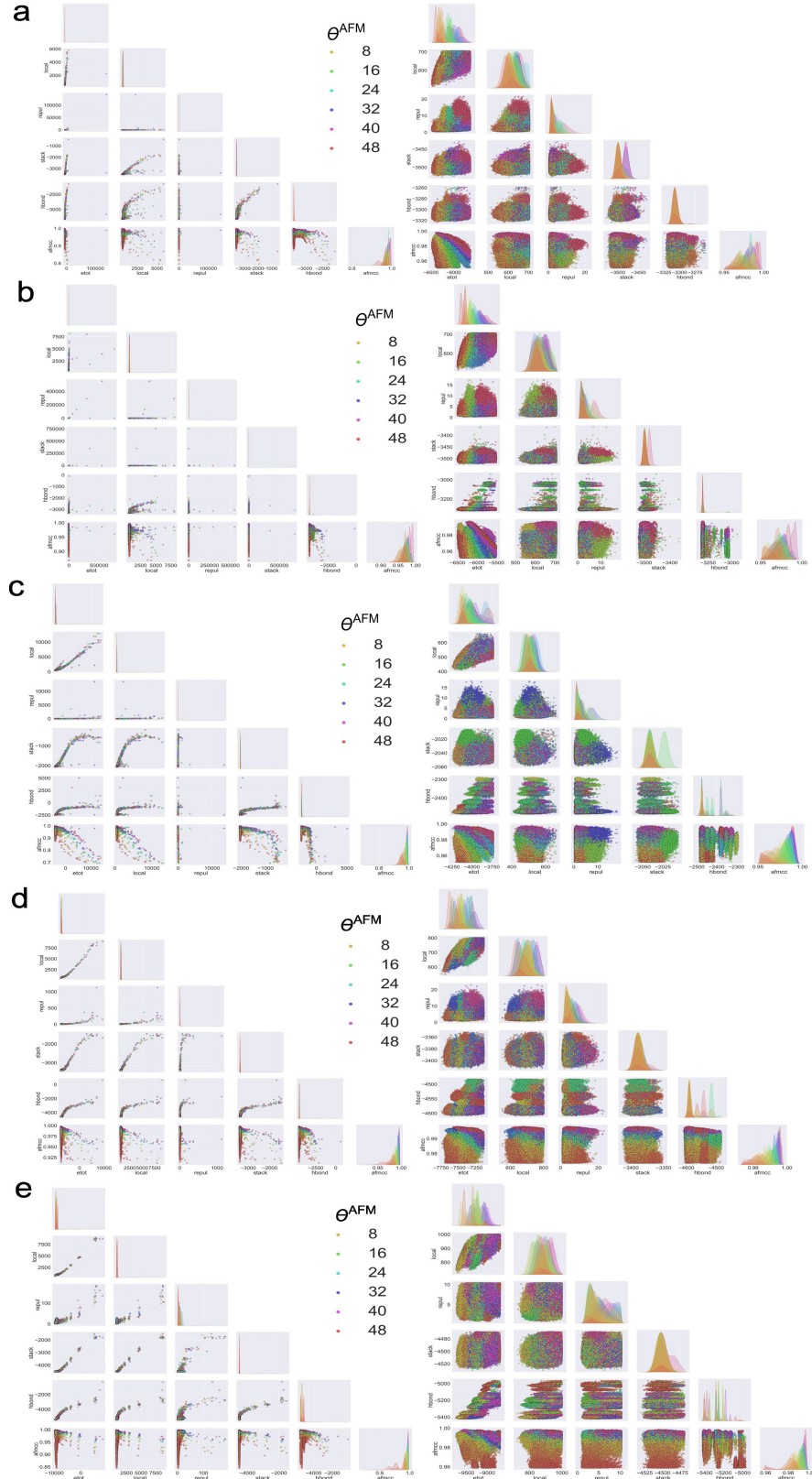

**Extended Data Fig. 3 | Energy and topography pair-correlation plots obtain during HORNET-UML analysis.** Left: raw trajectory. Right: after HORNET-UML energy-filtering for: **a**, Benchmark 1 (BM1); **b**, Benchmark 2 (BM2); **c**, Benchmark 3 (BM3); **d**, Benchmark 4 (BM4); **e**, Benchmark 5 (BM5).

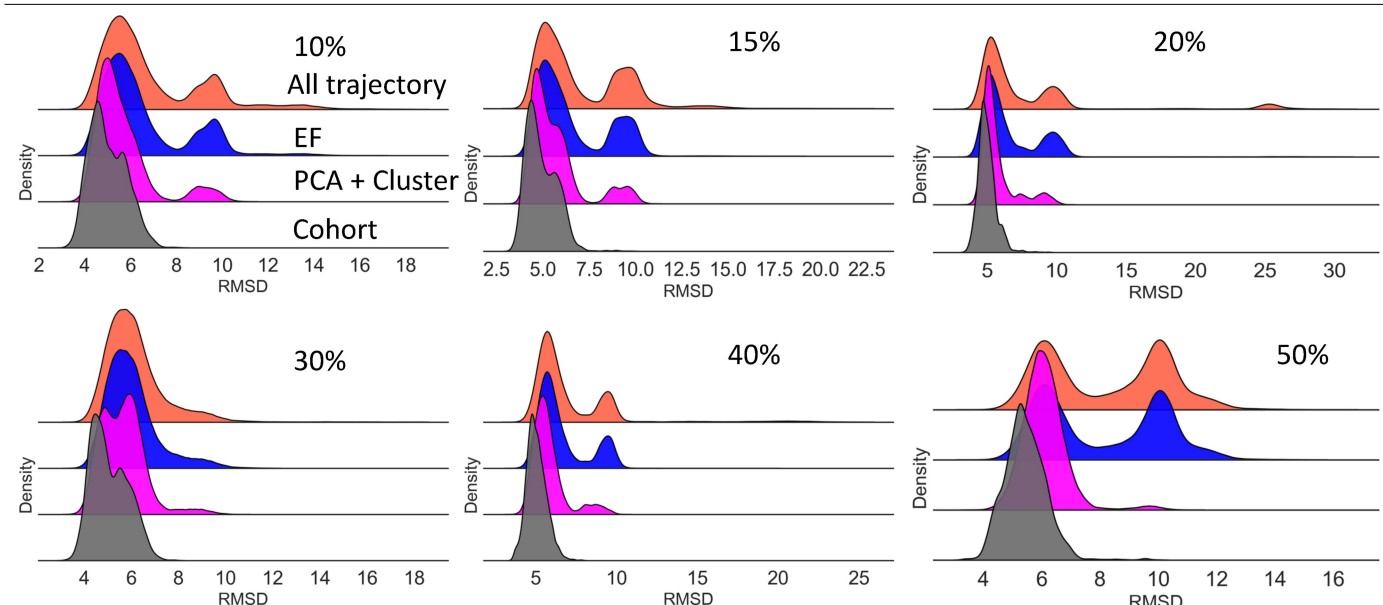

**Extended Data Fig. 4 | HORNET-UML analysis of Benchmark 0 (BM0).**
RMSD distribution at each stage of UML for the BM0 trajectories with different applied Gaussian noise levels (10–50%). Orange: all models generated during the dynamic fitting. Blue: remaining models after energy-filtering. Magenta: the selected cluster after PCA and clustering analysis. Gray: final cohort of models selected from the cluster, based on energetic and topographic information.

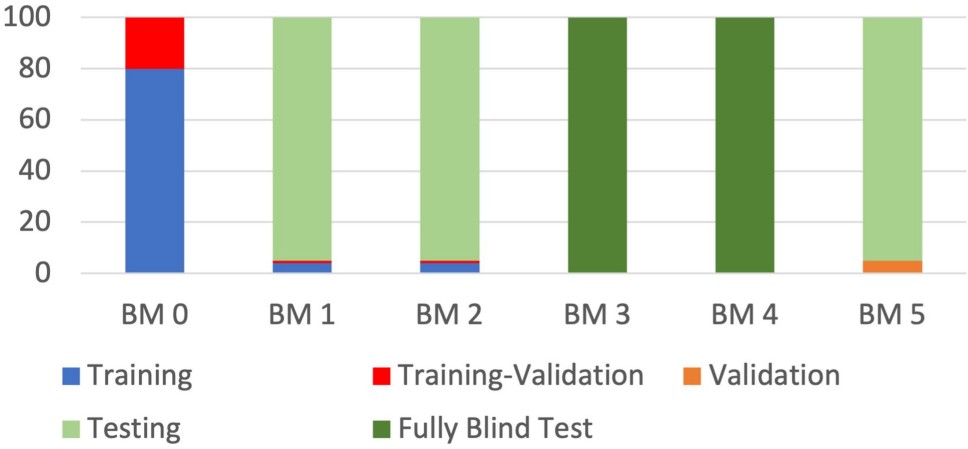

**Extended Data Fig. 5 | Benchmarks for training, validation, and testing HORNET.** Percentages of the total simulated data used for training (pseudo-structure database: 80% BM0, 5% BM1, 5% BM2), training-validation (20% BM0, 1% BM1, 1% BM2), validation (5% BM5), blind testing (100% BM3 and BM4) and testing (all remaining data).

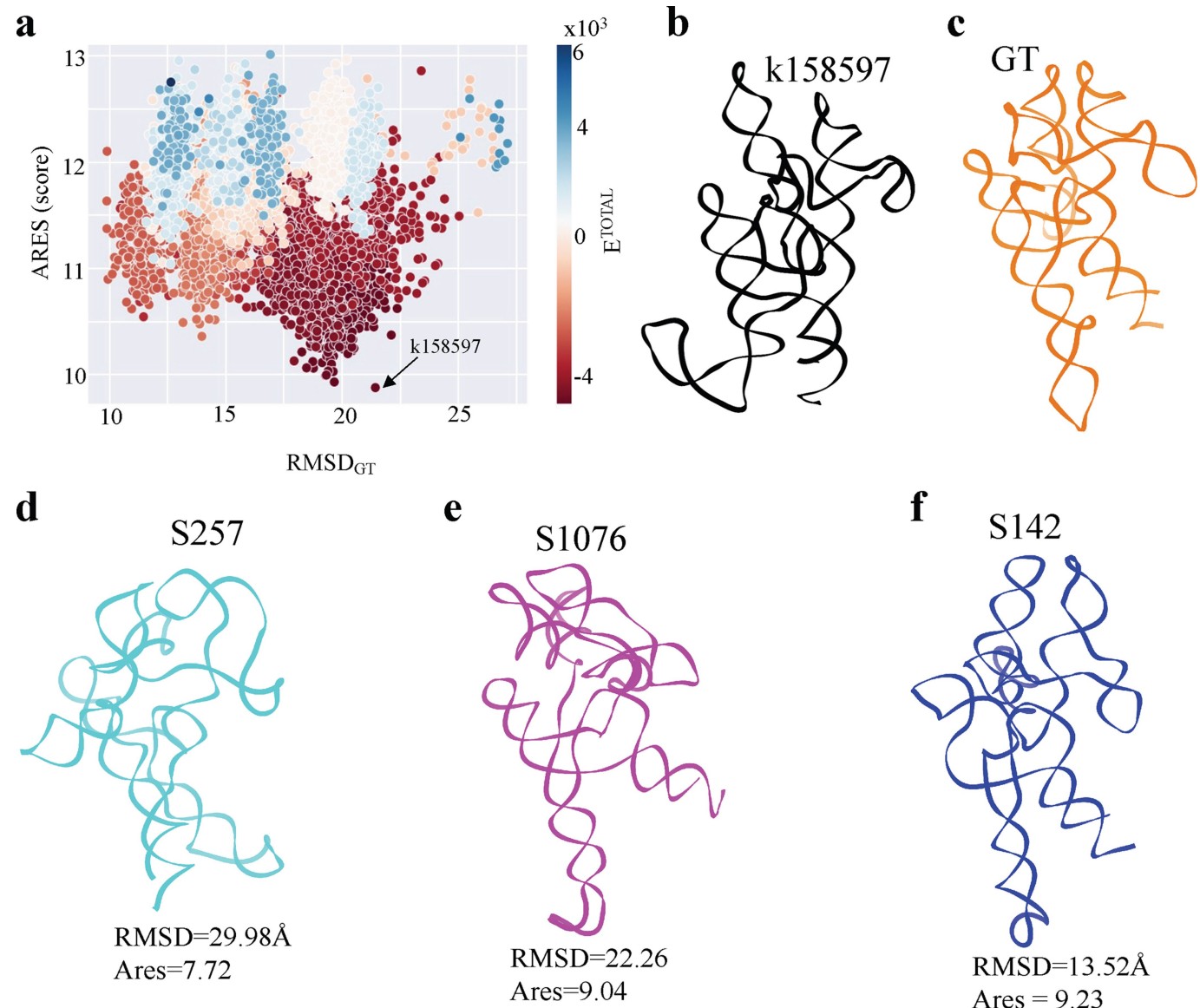

**Extended Data Fig. 6 | The initial structural models used for benchmarking.** **a**, ARES score vs RMSD of the dynamic fitting trajectory for RNase P RNA catalytic domain (see Methods). ARES scores were calculated using geometric deep learning[28]. Models are colored as a function of their total energy: dark red for lower energies and dark blue for higher energies. **b**, Ribbon diagrams of the trajectory model with the lowest ARES score, k158597 (orange), used as the initial structure for BM0, with an RMSD of 21.4 Å for 3DHS (black) crystal struture. **c**, Crystal structure of the RNase P RNA catalytic domain (PDB 3DHS)[40] with missing residues modeled, which was used as the GT structure. The

models from FARFAR2 structure calculation using the secondary structure information (**d**) S257, (**e**) S1076 (BM2), and (**f**) S142 (BM1) were selected from the pool of 10,000 models based on, respectively, the best ARES score (S257), best combined ARES and FARFAR2 scores with RMSD > 20 Å from the GT structure(S1076), or best combined ARES and FARFAR2 scores and RMDs relative to the GT structure of > 10 Å and <15 Å (S142). The ARES scores and the RMSDs to the GT structure are indicated. The missing residues in the crystal structure were modeled and energy-minimized using PHENIX.

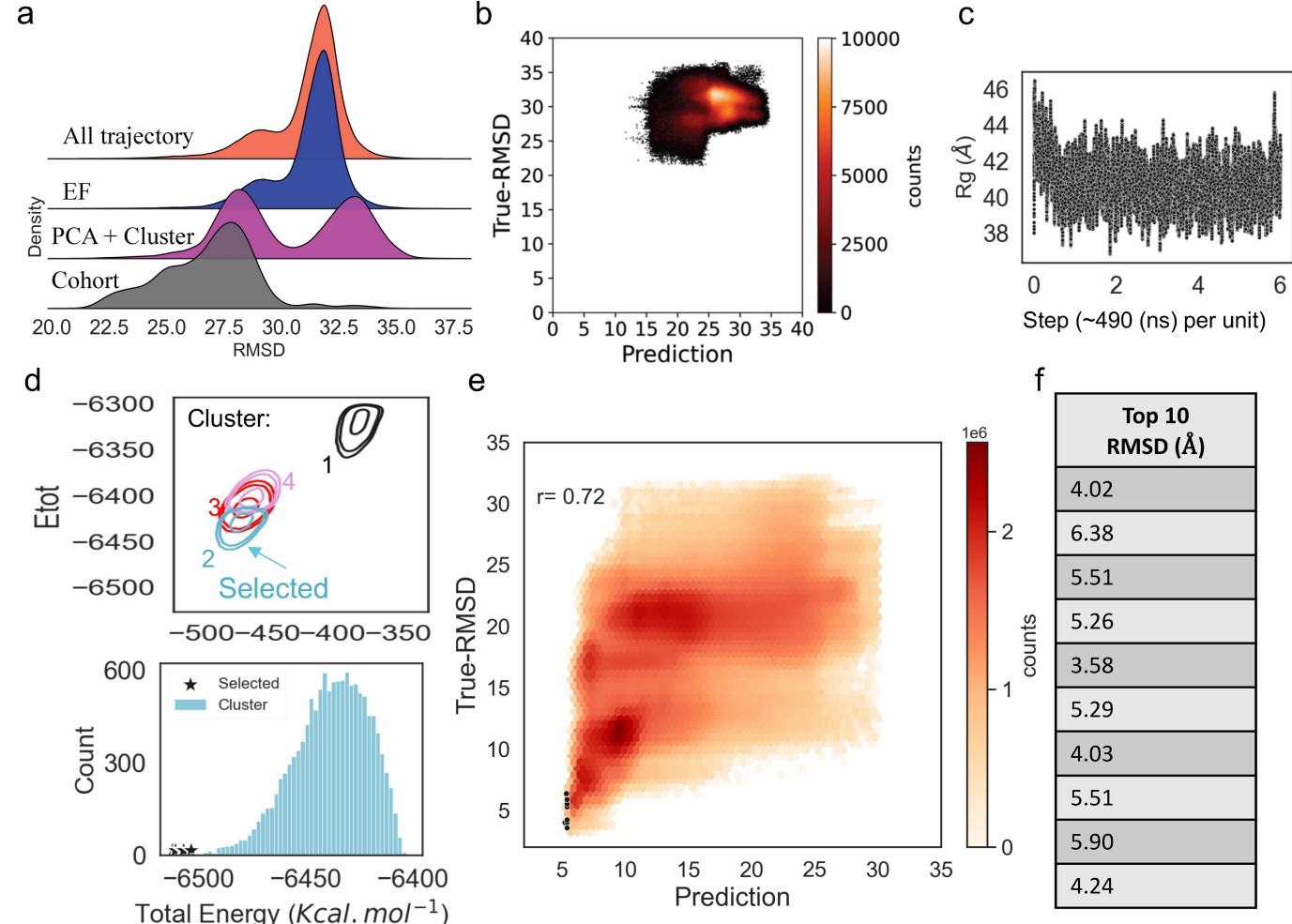

| Top 10 RMSD (Å) |
|---|
| 4.02 |
| 6.38 |
| 5.51 |
| 5.26 |
| 3.58 |
| 5.29 |
| 4.03 |
| 5.51 |
| 5.90 |
| 4.24 |

**Extended Data Fig. 7 | Structure calculation using FARFAR2 prediction S257 as starting model for RNase P RNA catalytic domain. a**, HORNET-UML analysis of non-convergence for S257. Even though S257 shows a good FARFAR2 score and the best ARES score, this structural model exhibits a fold completely different from the GT structure. We applied HORNET to S257 using simulated AFM topography of the GT structure (PDB 3DHS with missing residues modeled) as the driving force for dynamic fitting. For the trajectory time of ~11μs (estimation using CafeMol[31] unit time of 49 fs), the structure could not converge on a conformation close to the target structure, showing a minimum true RMSD of 21.4 Å and a major population centered at 27.5 Å RMSD in the final cohort of models (gray). **b**, Actual RMSD vs estimated RMSD using HORNET for the S257 dynamic fitting trajectory, the minimum estimated RMSD using HORNET, for the entire trajectory of ~16 million models, is 12.4 Å. However, this minimum is a clear outlier in the distribution, which is centered at ~27 Å.

HORNET was able to show that the trajectory of S257 could not converge to a structure resembling the GT structure. **c**, radio of gyration ($R_g$) as function of a coarse-grained MD simulation of ~2.9 $\mu s$ trajectory using CafeMol[31]. This simulation was performed to optimize the energy minimization of S257. **d**, To select the minimized structure from the MD trajectory, we applied the HORNET-UML approach without the AFM restraint, and the cluster with structures having the lowest Total and Go energies were selected (top). Next, we selected the five models having the lowest total energy (star symbol) with RMSDs relative to the GT structure of 19.3, 17.8, 19.0, 19.2 and 14.4 Å (below). **e**, Correlation between the true and estimated RMSDs by HORNET relative to the GT structure for all models from the five independent trajectories using the five selected the lowest-total-energy. The black dots indicate the top 10 HORNET-derived models having the lowest estimated RMSD. **f**, RMSD of the top 10 HORNET-derived models vs the GT structures.

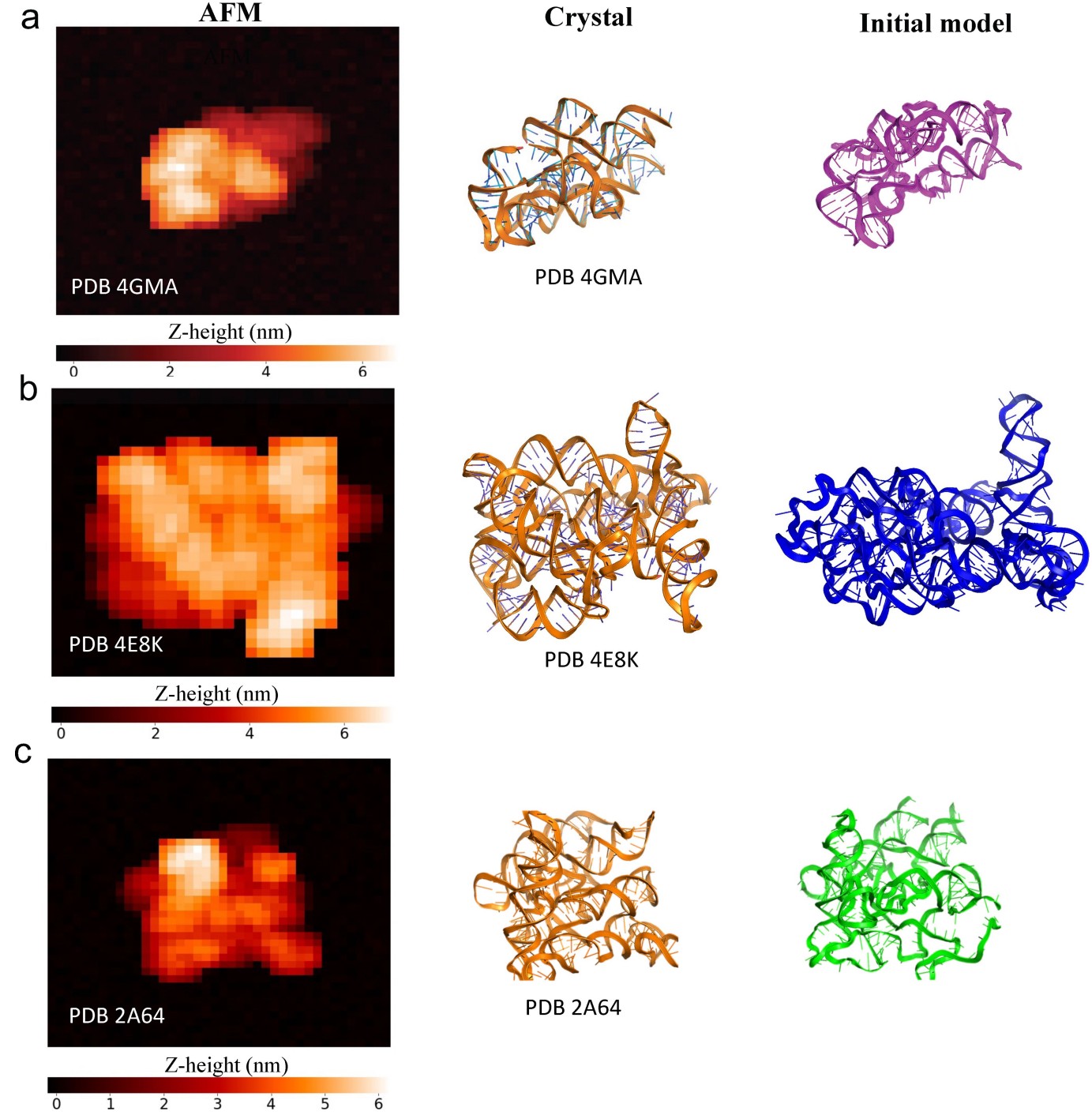

**Extended Data Fig. 8 | AFM images and initial models for BM3, BM4, and BM5.** The crystal structures of **a**, adenosylcobalamin riboswitch (PDB 4GMA)[47], **b**, group II intron (PDB 4E8K)[49], and **c**, RNase P RNA (PDB 2A64[33] with missing residues modeled) were used as the GT structures for BM3, BM4, and BM5, respectively. The simulated AFM images (left) were generated at 5 Å/pixel using the GT structures. The starting models used (right) and their RMSDs to the GT structures (middle) are as follows. BM3: MD trajectory model (magenta), RMSD of 10.8 Å; BM4: model derived using RS3D-simulated SAXS data (blue), RMSD of 16.1 Å; BM5: model derived using RS3D-simulated SAXS data (green), RMSD of 14.0 Å.

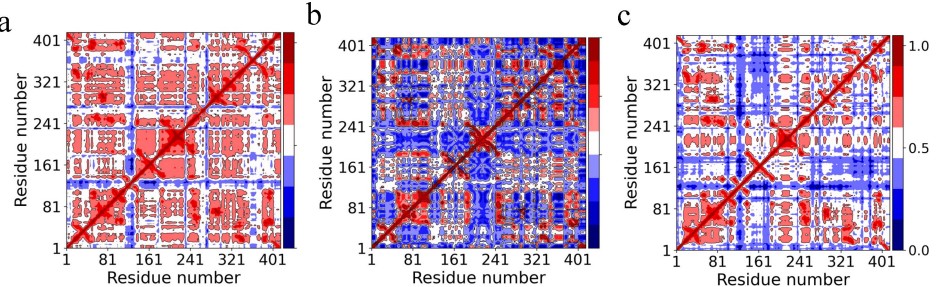

**Extended Data Fig. 9 | Dynamic cross-correlation maps (DCCMs) for the full-length RNase P RNA test case.** The DCCMs for the dynamic fitting trajectories of RNase P RNA AFM particles: **a**, P1. **b**, P2. **c**, P3.

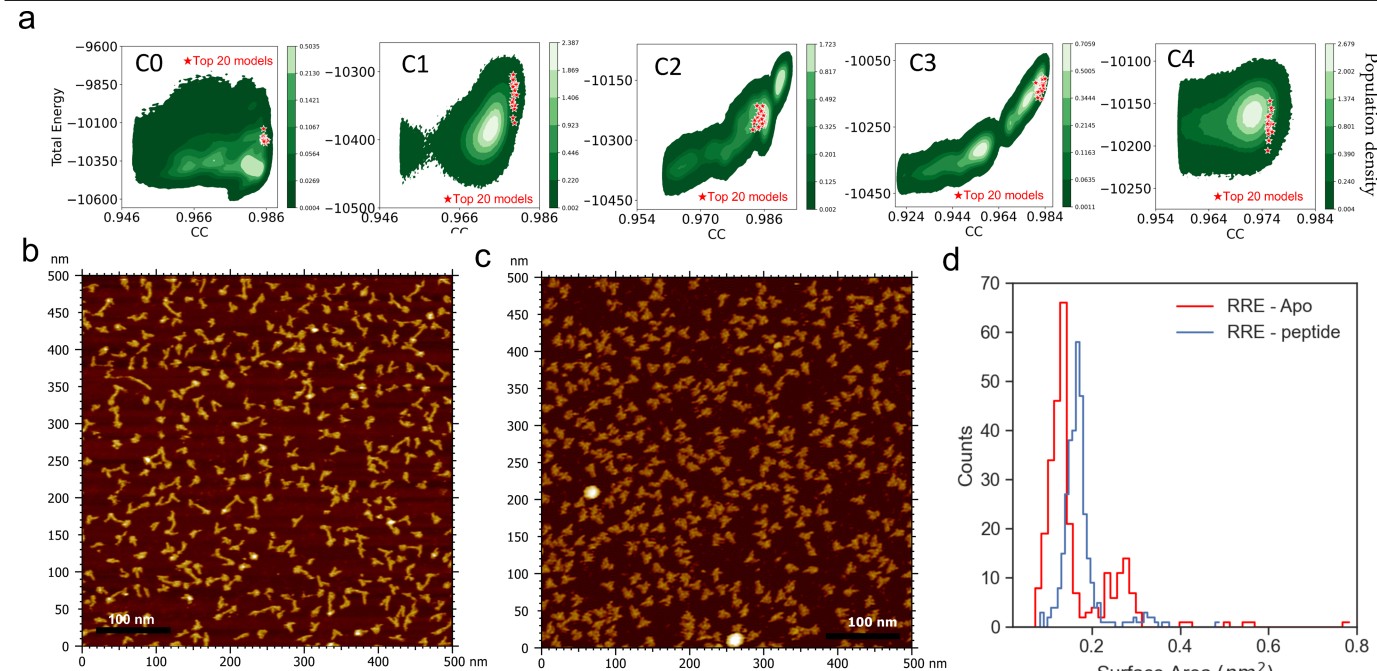

**Extended Data Fig. 10 | RRE structure calculation from experimental AFM data using HORNET and conformational heterogeneity. a**, Surface plot, showing the population density as a function of total energy and goodness-of-fit to the AFM molecular surface (CC), for the trajectory models from each of the five selected RRE conformers from left to right: C0, C1, C2, C3, and C4. The star symbols represent the top-20 scored models, based on estimated RMSD. **b**, AFM image of HIV-1 RRE RNA. **c**, AFM image of the RRE RNA in the presence of peptide. **d**, AFM tally of the surface area using the raw image of RRE in the absence (red) or presence (blue) of the peptide. The RRE shows a more homogenous conformational distribution in the presence of the peptide. The raw images in **b** and **c** were taken once thus without signal-averaging.

# Reporting Summary

## Statistics

For all statistical analyses, confirm that the following items are present in the figure legend, table legend, main text, or Methods section.

| n/a | Confirmed | |
|---|---|---|
| ☐ | ☒ | The exact sample size (*n*) for each experimental group/condition, given as a discrete number and unit of measurement |
| ☐ | ☒ | A statement on whether measurements were taken from distinct samples or whether the same sample was measured repeatedly |
| ☒ | ☐ | The statistical test(s) used AND whether they are one- or two-sided<br>*Only common tests should be described solely by name; describe more complex techniques in the Methods section.* |
| ☐ | ☒ | A description of all covariates tested |
| ☐ | ☒ | A description of any assumptions or corrections, such as tests of normality and adjustment for multiple comparisons |
| ☐ | ☒ | A full description of the statistical parameters including central tendency (e.g. means) or other basic estimates (e.g. regression coefficient) AND variation (e.g. standard deviation) or associated estimates of uncertainty (e.g. confidence intervals) |
| ☒ | ☐ | For null hypothesis testing, the test statistic (e.g. *F*, *t*, *r*) with confidence intervals, effect sizes, degrees of freedom and *P* value noted<br>*Give P values as exact values whenever suitable.* |
| ☒ | ☐ | For Bayesian analysis, information on the choice of priors and Markov chain Monte Carlo settings |
| ☐ | ☒ | For hierarchical and complex designs, identification of the appropriate level for tests and full reporting of outcomes |
| ☐ | ☒ | Estimates of effect sizes (e.g. Cohen's *d*, Pearson's *r*), indicating how they were calculated |

*Our web collection on statistics for biologists contains articles on many of the points above.*

## Software and code

Policy information about availability of computer code

| Data collection | Asylum atomic force microscope was used to acquire AFM images, and CafeMol (V. 3.0.2.0000) was used for creating the Dynamic Fitting data |
|---|---|
| Data analysis | Spip (V. 6.7.8) and MountainsSpip (V. 8) for AFM data processing, the HORNET python analysis code available at Zenodo (https://doi.org/10.5281/zenodo.10637777) |

For manuscripts utilizing custom algorithms or software that are central to the research but not yet described in published literature, software must be made available to editors and reviewers. We strongly encourage code deposition in a community repository (e.g. GitHub). See the Nature Portfolio guidelines for submitting code & software for further information.

## Data

Policy information about availability of data

All manuscripts must include a data availability statement. This statement should provide the following information, where applicable:
- Accession codes, unique identifiers, or web links for publicly available datasets
- A description of any restrictions on data availability
- For clinical datasets or third party data, please ensure that the statement adheres to our policy

- B0,B1, B2, B3, B4 and B5 energies information: https://home.ccr.cancer.gov/csb/pnai/data/HorNet/Energies_For_Benchmarks/
- B0,B1, B2, B3, B4 and B5 pdbs: https://home.ccr.cancer.gov/csb/pnai/data/HorNet/pdbs_For_BenchMarks.tar.gz
- B0,B1, B2, B3, B4 and B5 AFM images, noisy data, and initial structure: https://home.ccr.cancer.gov/csb/pnai/data/HorNet /AFM_For_BenchMarks.tar.gz

- P1, P2 and P3 experimental data and structure files: https://home.ccr.cancer.gov/csb/pnai/data/HorNet/P1_P2_P3_data.tar.gz
- RRE experimental data and analysis files with top selected structures: https://home.ccr.cancer.gov/csb/pnai/data/HorNet/RRE_data_results.tar.gz

# Research involving human participants, their data, or biological material

Policy information about studies with [human participants or human data](link). See also policy information about [sex, gender (identity/presentation), and sexual orientation](link) and [race, ethnicity and racism](link).

| | |
|---|---|
| Reporting on sex and gender | N/A |
| Reporting on race, ethnicity, or other socially relevant groupings | N/A |
| Population characteristics | N/A |
| Recruitment | N/A |
| Ethics oversight | N/A |

Note that full information on the approval of the study protocol must also be provided in the manuscript.

# Field-specific reporting

Please select the one below that is the best fit for your research. If you are not sure, read the appropriate sections before making your selection.

☒ Life sciences　　☐ Behavioural & social sciences　　☐ Ecological, evolutionary & environmental sciences

For a reference copy of the document with all sections, see [nature.com/documents/nr-reporting-summary-flat.pdf](link)

# Life sciences study design

All studies must disclose on these points even when the disclosure is negative.

| | |
|---|---|
| Sample size | Sample sizes refer to numbers of structural models in each case in unit of millions. BM0: 2 M; BM1: 14 M; BM2: 15M; BM3: 6.6 M; BM4: 1.7 M; BM5: 13.4 M; RNaseP RNA (P1, P2 and P3): 180M; RRE RNA (C0,C1, C2, C3, C4): 100M. |
| Data exclusions | No data were excluded from analysis. |
| Replication | The training was replicated more than 500 times for hyperparemeter tunning. The performance of the trained model could be reproduced in blind tests. |
| Randomization | For data splitting we used the pandas.DataFrame.sample method. |
| Blinding | The percentage of data used for blind tests are: 100% of BM3 and BM4, 20% of BM0, 95% of BM1 and BM2, 95% of BM3 . |

# Reporting for specific materials, systems and methods

We require information from authors about some types of materials, experimental systems and methods used in many studies. Here, indicate whether each material, system or method listed is relevant to your study. If you are not sure if a list item applies to your research, read the appropriate section before selecting a response.

## Materials & experimental systems

| n/a | Involved in the study |
|---|---|
| ☒ ☐ | Antibodies |
| ☒ ☐ | Eukaryotic cell lines |
| ☒ ☐ | Palaeontology and archaeology |
| ☒ ☐ | Animals and other organisms |
| ☒ ☐ | Clinical data |
| ☒ ☐ | Dual use research of concern |
| ☒ ☐ | Plants |

## Methods

| n/a | Involved in the study |
|---|---|
| ☒ ☐ | ChIP-seq |
| ☒ ☐ | Flow cytometry |
| ☒ ☐ | MRI-based neuroimaging |

