## [Peer Review File · Nature]

Manuscript Title: Determining structures of RNA conformers using AFM and deep neural networks

Editorial Notes: Redactions – unpublished data

Reviewer Comments & Author Rebuttals

Reviewer Reports on the Initial Version:

Referees' comments:

Referee #1:

The authors propose a new method called HORNET (Holistic RNA structure determination method using AFM unsupervised machine learning and deep neural NETWORKS) that aims to determine the 3D topological structure of RNA conformers based on AFM data. This paper uses coarse-grained MD simulations to obtain a large number of molecular trajectories restricted by covalent, non-covalent and AFM energy terms to reduce the degrees of freedom of the simulations and thus the searchable conformational landscape. Initial structures are determined such that they satisfy the weighted AFM pseudo-potentials and classical Gibbs free-energy descriptions, ensuring that the observed AFM data and the principles of stability and energy distribution are fairly considered. The list of suitable structures is then further refined using unsupervised machine learning methods to identify a cluster of structures with the lowest energetic distribution and the highest cross-correlation value. Once this cluster has been determined, supervised learning is used to estimate structure accuracy through prediction of RMSD. The authors then proceed to use both methods on a range of RNA conformers, including the RNase P RNA and HIV-1 RRE RNA, one which has a known crystal structure and other unknown conformers.

The work shown is of high quality and is an exciting step forward for the use of AFM in structural biology. The rationale for using AFM is clear, due to the flexibility of the RNA structures, which require the high signal-to-noise ratio of AFM. The main impact of this paper is in the automation for picking of individual particles, and close integration of simulations into a full pipeline for obtaining atomistic structures from AFM volumes, an exciting step forward which doesn't currently exist for this field.

The work presented here is very well considered, with training, test and validation data sets broad enough to avoid under- or over-fitting models, and thorough evaluation of model performance included throughout. The methods section was detailed and helpful, and suitable references to the methods were made within the main text throughout. The authors successfully described how their method adds to the current analysis approaches within the field and the potential for HORNET to be used to expand knowledge of the connection between topological structures and biological significance.

The authors have recently published another paper that also focuses on recapitulation of 3D topological structures from AFM images (<https://www.nature.com/articles/s41467-023-36184-x>). I would say the novelty of the HORNET software resides in its ability to automate detection of most suitable structures and integrate this with their method for "correlation between a topographic AFM image and the

underlying atomistic topological structure, and its use for recapitulation of individual RNA conformer structures have not been explored” (p. 5, line 2). This paper is only briefly referred to within the publication, “The crystal structure of BM3 was used as the GT model and the initial structure is one of the conformers revealed by AFM” (p. 15, line 1). More information as to how this builds on the authors’ initial study should be included to demonstrate how HORNET is a step forward, the workflow of the software and the potential for its use. How broadly could HORNET be used/applied? This would be further improved by more references to current availability of software/ML tools of image analysis in the field of AFM, and other examples of high resolution imaging/and or analysis of nucleic acids.

To have a wide-ranging impact, it would be useful to know whether it is possible to use existing simulations, crystal structures or EM structures for fitting and what the opportunities are for this across the AFM field. The ability to automate detection of 3D topological structures from AFM images holds significant value within the field; however it is possible that the software is currently inaccessible to its intended user base.

Along these lines, it is unclear to an uninitiated reader how this software is designed to be used. Fig. 1 is overly convoluted and confusing. What inputs can HORNET take? Can it take a wide range of inputs, e.g. coarse-grained simulations, atomistic simulations, EM/X-ray structures?

The large size (>75 GB) of the included folder of .pdb files, and also the requirement for HPC computing make this software challenging to test at the review stage and could potentially discourage usage following publication. The GitHub repository does contain a README file but more detailed information such as expected input/output file formats is required to aid usability and make this more accessible to the intended user base. For example, step 2 within the README involves copying .ts files as .txt files, yet the prerequisite .ts files are not provided. Further reading of CafeMol documentation suggests that such files can be output from there with the .pdb files used as CafeMol input, although this is not explained within the HORNET documentation. The README does include step-by-step instructions for working through the analysis pipeline and the codes run successfully up to the final DNN stage where a KeyError: ‘baseP’ error occurs.

Further documentation would significantly improve the usability of the software, including further details of the data repository file structure, the expected input/output file formats for each stage of the workflow and how these can be obtained, as well as a summary of the output .csv files from the UML stage and the variables they contain. Many of the intended users may not have a computational background and so ensuring that the software is user-friendly and accessible is paramount for the paper’s impact.

In addition it would be useful if the authors could expand on the below points:

Is the structural difference observed in figure 2 conformational or deposition based?

All structures look similar (although as stated - not identical), could this be due to the AFM surface? Is there a way to quantify the impact of the surface potential provided by the AFM?

How “not identical” are these states? Is there a measure or just taking their word for it?

Could the authors clarify that the resolution analysis (Fig. 2) is calculated for the (XY?) direction, or XZ, YZ directions, as these should have variable precisions due to the higher precision in the AFM Z-piezo? Is this the resolution value being used for the dynamic fitting uncertainty taking both resolution values into account? Along these lines, would FSC (Fourier shell correlation) be a better metric instead of FRC (Fourier ring correlation) for this?

What are the trajectory models as an input? Are these a list of coarse-grain coordinates?

This point is picked up further below, but mainly arises due to the lack of clarity of the workflow. Is this: PDB -> Do Sims -> Get Energies -> Cluster / DNN? As stated earlier, clarification of this would be very useful for interested users. Clarification of both what is used and what is needed would be extremely helpful.

Fig. 4 states that the model with lowest val loss is used; is this epoch 6? Has the model just got lucky at this early stage of learning as it is showing the lowest loss or is the model that’s really used one from the epoch nearer the centre of the graph?

UML cutoffs are empirically chosen after testing on BM 0(Methods), but how different can these get in the wider picture of fitting different pdb structures, i.e. opposed to the smaller BM5 / a different sample? Do these need to be recalculated for each different sample?

Normalisation values are based on the empirically chosen values of "theta_c = 5, theta_stacking = 9 and theta_pairing = 9" to "achieve the optimal balance between enforcing the integrity of primary and secondary structures (the hierarchical principle) and achieving the best fit to the topological restraints at the same time". It is unclear how these were determined in the methods.

Minor comments:

The caption of Fig. 2d states the clear kinks for P1, P2 and P3 are highlighted by the dotted lines at $\sim 0.87 \text{ nm}^{-1}$ (11.5 \AA), 0.87 nm^{-1} (11.5 \AA), and 0.87 nm^{-1} (11.5 \AA), respectively. This is true for P1 but needs to be corrected for the P2 and P3 values to match the graphs.

Page 15, line 20, typo: Determining RNsae P RNA structures of heterogeneous conformations.

Fig. 6b: The y-axis is “HORNET - score”, which refers to RMSD. Would it be better to change this axis label to “Predicted RMSD” as in Fig. 5b?

URL links within the reference list appear to not be working.

There are some domain-specific terms (“weighted AFM pseudo potentials”, “AFM force potential”, “trajectory”, “trajectory structure model”) that, if briefly explained, would improve the readability of the

paper to those without extensive expertise.

Referee #2:

Degenhardt et al. use AFM and machine learning to address RNA structure and conformational heterogeneity. I want to make clear that I think AFM is a phenomenal and underutilized approach for RNA and so I think the authors' development of this approach is quite important. That said, I have questions and concerns on multiple levels about the manuscript; these are listed below in rough order of occurrence in the manuscript, and I have tried to address the level of importance of each concern. Overall, I think the level of clarity for the general reader is currently insufficient, both to make clear what was done and how it was done. It is possible that with greater clarity (or additional time spent on my part) some of these issues identified below would be resolved, so the authors should definitely respond to these points.

1. Setting up the work and its importance. Overall I think the authors make a good logical case for the need for their approach (or additional approaches in general that can address RNA heterogeneity at the single molecule level. That said I have several comments and criticisms:

i. The authors seem to assume (and cite one paper suggesting) that most mRNAs are structured; it is not clear to me that this is valid and there are multiple questions about the data that have been used to support this.

ii. The single molecule studies of heterogeneity cited may have been subject to covalent differences and surface interactions.

Important: The possibility of surface or other handling effects does not seem to have been addressed in this work.

iii. There are other studies not cited that seem highly relevant and arguably more relevant than many that are cited. I hesitated to raise these as they are from my work and those of collaborators, so I apologize if the list is biased and if I am over-emphasizing their value:

a. NMR RDCs and chemical shifts have been used to obtain atomic-level ensembles for an RNA element. In addition to this precedent, the degree to which atomic-level information will be needed to extract information of value or high value is not addressed (i.e., the authors correctly state that AFM can give helices, but how much physical and biological information can be derived at this level is not addressed).

Shi, H., Rangadurai, A., Assi, H.A., Roy, R., Case, D.A., Herschlag, D., Yesselman, J.D., Al-Hashimi, H.M. (2020) Nat. Commun. 11, 5531. Rapid and Accurate Determination of Atomistic RNA Dynamic Ensemble Models Using NMR and Structure Prediction. PMID: 33139729.

Ken, M.L., Roy, R., Geng, A., Ganser, L.R., Manghrani, A., Cullen, B.R., Schulze-Gahmen, U., Herschlag, D., Al-Hashimi, H.M. (2023) Nature 617, 835-841. RNA Conformational Propensities Determine Cellular Activity. PMID: 37198487.

b. As the authors are aware, the Au-SAXS approach developed by Harbury provides a distribution of distances that directly reflects the distribution in solution. This information is more precise than the averaging in 'regular' SAXS and has revealed heterogeneity in the past. It would seem that using RNAs with prior Au-SAXS data and using Au-SAXS to test structural models would be better validations and tests of the approach than what was done in this manuscript. In addition one could use different solution conditions and ensure (test) that the molecular structures obtained change as predicted.

This point raises an additional weakness (as I understand it) as individual molecules are chosen for deeper study: do the authors obtain information about the entire population of molecules? That would seem needed for the approach to have significant value, and one would need to show that the population obtained on the grid reflects that in solution (so a great way to do this would seem to be to vary the solution conditions for RNAs like kink turns that have a distribution of shapes are known to change shape with salt).

Without information about the distribution of shapes how does one know that the three molecules chosen for this study are representative? This is an issue (unresolved) in cryo-EM as well; there researchers do obtain multiple states but also leave out many (most?) of the molecules and also may approximate continuous distributions by a discrete number of states that are models that sample the more continuous range of conformers explored.

Shi, X.S., Huang, L., Lilley, D.M.J., Harbury, P.A.B., Herschlag, D. (2016) Nat. Chem. Biol. 12, 146-152. The Solution Structural Ensembles of RNA Kink-turn Motifs and Their Protein Complexes. PMID: PMC4755865.

See also: Wang, Y-X. (2016) Nat. Chem. Biol. 12, 126-127. News and Views. RNA Conformation: Lightning Up Invisible States.

The authors suggest that heterogeneity is the cause of the limited number of RNA structures in the PDB; the possibility that there may be (many) fewer structured RNAs than proteins and recent breakthroughs in RNA cryo-EM are not noted:

"Thus, a single snapshot structure falls short of accurately describing the conformational landscape associated with its function as clearly demonstrated recently in a cellular context²."

Here the authors do not cite a recent Nature paper (by us) that shows that accounting for the landscape -via NMR approaches- can quantitatively predict cellular function. It is reasonable to suggest that additional (and even better) approaches are needed, but not to imply that current approaches cannot do this at all.

NOTE: Journal rules I believe limit the number of citations allowed and that may be responsible for this omission. Such rules by journals are counter to scientific communication and should be abolished.

2. The approach. Some points relevant to this were brought up in #1 above. The main additional point is concerns about mixing the data (AFM) with models (forces, 2 σ structure predictions or assumptions, AI models) so that in the end one doesn't have a model that comes from the data but rather some mixture

of data and model; that (I think) is hard or impossible to unravel. While the test cases used are needed they are not sufficient as they are idealized and do not demonstrate that the process works with real RNA molecules and real data, only idealized cases.

The limitation alluded to above of mixing AI models with data is in part analogous to limitations from a recent Science paper (Das & Dror) that showed that AI can help select from FARFAR-derived structures to find the correct ones. However, most reading the paper seems to conclude that AI can now predict RNA structure, which is very different than what was shown. Here again, it seems that structure is not predicted but rather structural models generated from knowledge-based models (ironically, based on RNA PDB structures) can be sorted and selected by applying AI to AFM data.

The possibility of alternative secondary structures does not seem to be taken into account. Also, there seems to be an assumption that species fall along (rather than some being orthogonal to) a folding pathway that progresses from unfolded to full folded.

There does not seem to be discussion or controls for surface effects.

Returning to establishing the method, there does not seem to be “ground truth” data of a multi-state system that was used and independently determined via AFM-AI. While the results are reasonable, they appear to not meet this highest standard. Arguably this was done in the authors’ Nat. Commun. paper with a riboswitch (WT and mutants and showing transitions upon ligand binding) so minimally discussion of this critical technical point I think is important to include.

Validation (i.e., testing) via SAXS seems less than ideal, given that it does not return or encompass (much) atomic-level structural information. It seems that Au-SAXS results could be predicted and tested, though sample preparation for that technique is time consuming (though can still be done in <1-2 weeks).

The main conclusion seems to be that the structural results of an RRE dimer with binding sites facing it led to the design of tethered ligands. The authors state that a “long-standing mystery” is resolved, but do not clearly articulate what this mystery (perhaps for reasons of space limitations, but this is needed to evaluate the significance of this results). Also, I was not able to (readily) glean how the conclusion was reached of less heterogeneity after ligand binding (in supplemental material); minimally clearer presentation would help. A related question that the authors likely have information for is whether different A-like structures are favored with different length tethers; such information would provide information about the energetics of the conformational landscape, albeit on a crude level.

The tethered peptide experiment is somewhat analogous to an old Uhlenbeck hammerhead ribozyme experiment where they tethered two hammerhead RNA arms (by making it a continuous oligo I think) to assess the bend angle of the active form. That experiment was not fully successful but might be considered and cited here. More broadly, could the tethering experiment been done “blindly” (or based on the A-like structure derived from SAXS) without AFM to see what length tethers give the strongest

binding?

3. The presentation.

More information for figures and more table would help. E.g., how many total images; how many processed; labels on some of the figures are missing or not fully defined. While this is hard to do with so much data and so many steps, in some ways it is even more important.

As a ML/AI novice, I was not able to readily follow the approaches used, the reasons for these approaches (over others), what the evidence is that they “really” work, and what information (if any beyond empirical) is derived from the AI. Related to this, while numbers are improved (R^2 etc) it is not clear (to me) what a significant improvement is or whether this was considered. E.g., there are some important papers in algorithm development (I think for protein structure prediction) that suggest most “improvements” from new algorithms instead arise from noise in fitting and so are not real.

4. The overarching goal seems to be to develop a means to model RNA (at the individual molecule level) so that this information can be used more broadly and generally to solve the (multi-state) RNA folding problem. Given that the data are not on the atomic scale and given that the structures obtained are derived from a mixture of the data and the model, it is not clear to me that this goal is achievable by following this path or worth achieving alone without thermodynamic and/or kinetic data. E.g., will AFM be more valuable for dissecting transitions between states (over conditions and time), providing a highly direct connection of structures formed to their underlying energetics? Further, such AFM exploration could identify conditions that favor different states and thus can be used to learn about each of the underlying states.

I am not trying to rewrite the authors’ paper but rather describe my reaction to it with the goal of helping the authors improve the ability of their manuscript to reach broad and expert audiences. Again, there is much here I may have missed, and I hope that those cases will also help the authors clarify points.

Daniel Herschlag

Referee #3:

Degenhardt et al. introduced an advanced method, named HORNET, that leverages atomic force microscopy (AFM) images along with classical unsupervised and supervised machine learning techniques to elucidate the three-dimensional structure diversity of RNAs. In this work, the authors demonstrated HORNET on RNase P RNA particles (P1, P2 and P3), cobalamin-sensing riboswitches (rCbl), and the potential conformational heterogeneity of HIV-1 Rev response element (RRE) RNA.

Overall, this work brought up some interesting ideas in combining the atomic force microscopy (AFM)

images with machine learning techniques to determine RNA conformational diversity. However, the machine learning techniques used in this work were very classical and simple. There are several concerns about the modelling itself regarding both the features and the sample/testing dataset. It is also not convincing that this method can be widely applied to dissect RNA structure conformations. It is also not clear about the novel impacts of this method, e.g. whether this method could determine new functional RNA structure conformation that was not possibly determined previously, and whether it can determine the ratio between different conformations. Also, the software, "HORNET", is not user-friendly, and the authors should comprehensively fix their codes.

My review mainly focuses on both the machine learning techniques (including unsupervised and supervised learning) and the software code provided in this work. The authors used the classical PCA and K-means algorithms for unsupervised learning. Then the lowest energy clusters were obtained by cohort selection. For the supervised learning, the authors traversed the parameter space and generated a "pseudo-structure database" due to the lack of large amounts of RNA structure data. Then they adopted the classical dense network modelling to predict the difference between the real structure and the pseudo-structure. A similar approach to the one used in this study (Townshend et al., Science, 2021) was implemented in the ARES software to overcome the problem of insufficient 3D RNA structure data." While going through the codes in this manuscript, I found them overall not user-friendly. Either necessary example files were missing, or certain documentation was not clear. At least it should be possible for the user to run through the entire process with sample data. Here are the detailed comments based on my testing.

In the unsupervised learning part, the code performs the overall following steps: reading and cleaning the data, filtering by energy, conducting PCA analysis and clustering, and performing cluster selection and cohort selection. The RNA conformation represented by the AFM can be classified according to unsupervised learning, and the top structure model of the real structure under the topological surface is then obtained by energy selection. Finally, the filtered data, selected cluster data, and cohort data are saved as separate CSV files. In the deep learning component, the authors employed the Keras framework to construct a sequential model consisting of a three-layer dense network, encompassing a total of 10,721 parameters. The training input data comprised 16 extracted features, namely "frame," "radg," etc., while the output corresponded to the "rmsd" value. The authors have demonstrated that the model is not over-fitted using the 'dropout' and 'L2' regularisation at the model construction level and validated with both untrained and blind datasets. However, based on the datasets in this work, the authors might run into the issue of "undetectable overfitting" with two concerns:

Firstly, I noted the high correlations observed between "etot", "hbond", "local" and "stack" among the 16 features extracted by the authors, exceeding 0.9. Such high correlations raise the issue that these features convey redundant information that can particularly lead to ineffective modelling.

Secondly, among the 200,000 rows of data used in the example, I extracted 100,000 samples and randomly correlated them two by two, achieving an average correlation coefficient of 0.94. In this case, even if the training and test sets were randomly assigned, they will reach very similar outputs that lead

to a bias in the accuracy of modelling. The authors should find a better way to validate the accuracy of their model.

Also, considering that these data emanate from a "pseudo-structure database," the authors should ensure uniform sampling across the parameter space. Additionally, in the script of the model training, the authors used a fixed number of 5 rounds. Is there any justification of using 5 rounds? Some methods such as "earlystop" may help determine an optimal number of rounds.

Regarding the software part that I have tested, I am disappointed that most of the functions did not work. It is mainly due to the lack of sample files (like: "ts" file, AFM data, and sampling data...). Therefore, I could not evaluate these functions. In addition, when testing the individual scripts, the "README" instructions did not allow the correct configuration of all the code. Therefore, I installed the required packages manually. If the software is to be formally released, I strongly recommend the software should be packaged as a real Python software package (rather than a collection of different bioinformatic scripts) and may consider releasing the package on public platforms such as PYPI, Anaconda, Docker, etc. The authors should perform appropriate Python unit tests to determine the usability.

Here are some specific detailed issues regarding the software use I have encountered.

1. Following the current instructions to create a conda environment for "python 3.7", it will cause compatibility problems during the second "pip" installation dependency. The Error is: "ERROR: Ignored the following versions that require a different python version," "ERROR: Could not find a version that satisfies the requirement numpy==1.24." I get the same error on both mac and Linux terminals. Please modify the installation instructions in the README.
2. After I had installed all the dependent packages manually, I found that although the introduction mentions the format of the input files, a runnable example file was missing. An example "ts", "bases_k<kappa>.txt" should be given in the "Step 2 - Input Preparation" section. etc. to make the code for the examples runnable. I then installed the cafeMol software and followed the "protocol.txt" step-by-step in the folder. However, in the first step "path/cafemol input.inp" gives an error ("can not open the file: ./output/out_name.data").
3. The example file, Full_Trajectory.csv, is not given, or relies on the output of step 2, so I was unable to run the unsupervised learning part (step 3) of the code. For step 4, there is no example file or dependency on the data from the previous step.
4. DNN's train.py is runnable. But there are still some issues. If "-l huber" is used in the parameter, then the error "ValueError: invalid literal for int() with base 10: 'huber'" will be reported. Delete this parameter or modify the code to work correctly.
5. The file named 'RRE.tar.gz.zip' downloaded from the provided link

("https://home.ccr.cancer.gov/csb/pnai/data/HorNet/") requires a password when being unzipped.
Please fix this issue.

Author Rebuttals to Initial Comments:

Referee #1:

*The authors propose a new method called HORNET (Holistic RNA structure determination method using AFM*
*unsupervised machine learning and deep neural NETWORKs) that aims to determine the 3D topological structure of*
*RNA conformers based on AFM data. This paper uses coarse-grained MD simulations to obtain a large number of*
*molecular trajectories restricted by covalent, non-covalent and AFM energy terms to reduce the degrees of freedom*
*of the simulations and thus the searchable conformational landscape. Initial structures are determined such that*
*they satisfy the weighted AFM pseudo-potentials and classical Gibbs free-energy descriptions, ensuring that the*
*observed AFM data and the principles of stability and energy distribution are fairly considered. The list of suitable*
*structures is then further refined using unsupervised machine learning methods to identify a cluster of structures*
*with the lowest energetic distribution and the highest cross-correlation value. Once this cluster has been determined,*
*supervised learning is used to estimate structure accuracy through prediction of RMSD. The authors then proceed to*
*use both methods on a range of RNA conformers, including the RNase P RNA and HIV-1 RRE RNA, one which has a*
*known crystal structure and other unknown conformers.*

*The work shown is of high quality and is an exciting step forward for the use of AFM in structural biology. The*
*rationale for using AFM is clear; due to the flexibility of the RNA structures, which require the high signal-to-noise*
*ratio of AFM. The main impact of this paper is in the automation for picking of individual particles, and close*
*integration of simulations into a full pipeline for obtaining atomistic structures from AFM volumes, an exciting step*
*forward which doesn't currently exist for this field.*

*The work presented here is very well considered, with training, test and validation data sets broad enough to avoid*
*under- or over-fitting models, and thorough evaluation of model performance included throughout. The methods*
*section was detailed and helpful, and suitable references to the methods were made within the main text throughout.*
*The authors successfully described how their method adds to the current analysis approaches within the field and*
*the potential for HORNET to be used to expand knowledge of the connection between topological structures and*
*biological significance.*

We thank the reviewer for many encouraging comments.

*The authors have recently published another paper that also focuses on recapitulation of 3D topological structures*
*from AFM images (<https://www.nature.com/articles/s41467-023-36184-x>). I would say the novelty of the HORNET*
*software resides in its ability to automate detection of most suitable structures and integrate this with their method*
*for "correlation between a topographic AFM image and the underlying atomistic topological structure, and its use*
*for recapitulation of individual RNA conformer structures have not been explored" (p. 5, line 2). This paper is only*
*briefly referred to within the publication, "The crystal structure of BM3 was used as the GT model and the initial*
*structure is one of the conformers revealed by AFM" (p. 15, line 1). More information as to how this builds on the*
*authors' initial study should be included to demonstrate how HORNET is a step forward, the workflow of the*
*software and the potential for its use. How broadly could HORNET be used/applied? This would be further*
*improved by more references to current availability of software/ML tools of image analysis in the field of AFM, and*
*other examples of high resolution imaging/and or analysis of nucleic acids.*

We thank the reviewer for the suggestion. We have revised the Introduction (p. 3-4) with explicit
descriptions of what has or has not been done in the field, as well as the novelty of HORNET, the significance of its
capabilities, and the breadth of its potential applications. For clarification, there is no available method for
recapitulating structures from experimental topographical images of individual RNA molecules with any measurable
or meaningful level of accuracy. HORNET is the first technology to do this, whose deep neural network identifies
the top cohort of structures that best fit the experimental data, while also providing an estimated accuracy of each.
Such an approach is broadly applicable to studying RNAs with unknown and elusive structures, mapping
heterogeneous conformations, and correlating motions and RNA function. **Redacted**

We have adopted the reviewer's suggestion and added a workflow (Fig.1).

About "*How broadly could HORNET be used/applied?*", we demonstrate the HORNET impact by solving three
novel structures of the full-length (417-nt) RNase P RNA conformers. It is noteworthy to mention that the crystal
structure only consists of 298-nt and the rest could not be modeled due to a poor electron density map (Kazanitsev,
2005, PNAS). Furthermore, we also solved five novel structures of the HIV-1 RRE RNA conformers and addressed
several long-standing mysteries about RRE's function. **Redacted**

In conclusion, the answer is “Yes”. HORNET is applicable to determine 3D topological
structure of large RNA. We have stated this explicitly in Conclusion of the manuscript.

Regarding the suggestion *“This would be further improved by more references to current availability of
software/ML tools of image analysis in the field of AFM, and other examples of high-resolution imaging/and or
analysis of nucleic acids.”*, we thank the reviewer for this suggestion. We have properly cited all software for the
processing and statistical analysis of high-resolution AFM images of macromolecules (p. 46 in Methods). However,
there is no available software/ML analysis that performs structure determination and evaluation on a single molecule
level (i.e., without averaging) using an AFM image. This is one of novelties of HORNET. We have revised the
Introduction to clearly state the novelty (p. 4&6).

*“To have a wide-ranging impact, it would be useful to know whether it is possible to use
existing simulations, crystal structures or EM structures for fitting and what the opportunities
are for this across the AFM field. The ability to automate detection of 3D topological structures
from AFM images holds significant value within the field; however it is possible that the
software is currently inaccessible to its intended user base.”*

We thank and agree with the reviewer on the potential wide-ranging impact of our method. Indeed, our
study demonstrates that HORNET can recapitulate 3D topological structures from topographic AFM images of
individual molecules using initial structural models derived from a variety of methods, including simulated
structures, crystal structures, and cryo-EM structures. As illustrated in our study (Figs. 3 and 4), the 6 benchmarks
were performed using starting models obtained from a crystal structure (BM0), FARFAR2 prediction (BM1, BM2),
a coarse-grained MD trajectory structure BM3), and SAXS (BM4, BM5), all of which differed significantly from
their respective ground-truth structures, but the initial model could be from any source. The use of crystal or cryo-
EM structures as initial models represents relatively easy scenarios. Importantly, the initial model can be even in a
completely different fold from the ground-truth structure underneath the AFM surface, as we have shown with RNA
S257 (Extended Data Figs.11-15). The detailed demonstration of the potential impact of HORNET is beyond the
scope of the current manuscript that focuses on the dissemination and validation of HORNET. **Redacted**

In regard to the reviewer’s comment about automation, the work of streamlining the whole process, from
particle detection to recapitulating 3D topological structures in an automated fashion using software and high-
performance computing, is ongoing in our lab and will be released in the future version of HORNET.

*“The large size (>75 GB) of the included folder of .pdb files, and also the requirement for HPC
computing make this software challenging to test at the review stage and could potentially
discourage usage following publication.”*

The large size is due to the inclusion of all codes, test cases, and relevant output files necessary for this
review purpose in our institutional repository site: <https://home.ccr.cancer.gov/csb/pnai/data/HorNet/>). Specifically,
this includes all benchmarking data associated with the training, testing, and validation of HORNET, as well as all
results obtained for the experimental test-cases, RNase P and HIV-1 RRE RNAs. At the user level, however, the
HORNET package available on GitHub is relatively small due the space limit and now includes tutorial data only,
with accompanying instructions in the README file, to demonstrate the use of HORNET and its functions.
Importantly, although the dynamic fitting step could be run on a personal computer, a single simulation can take
from some hours to even days to finish, depending on the size of molecules. At the scale demonstrated in this paper,
it would be best suited by using clusters or HPCs, which could be normally found in University or Research centers.
On the other hand, the structural analysis using HORNET after the dynamic fitting does not require HPC and can
easily be run on a personal computer in ~5 min for an entry of 20 million models. Structure evaluation by HORNET
includes structure parameters that are written in a tabular file, which is relatively small (on the order of MB). In
terms of output models, only the top 10 candidate structures are written out by default to the final coordinates file
(hundreds of KB). To ensure our instructions in the README files are clear, we have also requested a cell biologist
and a biochemist to use our program. Neither of them has any experience or training in computational structural
biology, yet both could independently run HORNET successfully after reading the README instructions, without
any interaction with us.

*“The GitHub repository does contain a README file but more detailed information such as expected input/output*
*file formats is required to aid usability and make this more accessible to the intended user base. For example, step 2*
*within the README involves copying .ts files as .txt files, yet the prerequisite .ts files are not provided. Further*
*reading of CafeMol documentation suggests that such files can be output from there with the .pdb files used as*
*CafeMol input, although this is not explained within the HORNET documentation. The README does include step-*
*by-step instructions for working through the analysis pipeline and the codes run successfully up to the final DNN*
*stage where a KeyError: ‘baseP’ error occurs.*
*Further documentation would significantly improve the usability of the software, including*
*further details of the data repository file structure, the expected input/output file formats for*
*each stage of the workflow and how these can be obtained, as well as a summary of the*
*output .csv files from the UML stage and the variables they contain. Many of the intended*
*users may not have a computational background and so ensuring that the software is user*
*friendly and accessible is paramount for the paper’s impact.”*

We thank the reviewer for the very helpful suggestions. The problems that the reviewer mentions are all
related to some required input and data files that were misplaced in different folders. We have reorganized the
HORNET package to make the software more streamlined and user-friendly, including more detailed README
files for running the program with tutorial data. In addition, input and data files for all steps are provided in the same
TUTORIAL subfolder, which allows the user to perform any step in the pipeline without having to generate requisite
input files. Regarding the KeyError, we thank the reviewer for catching this. This error occurred because the
secondary structural information was not being carried over to the FinalCohort.csv output file, but only to the
AllTrajectory.csv file. This has been revised in the updated version.

*In addition it would be useful if the authors could expand on the below points:*

*Is the structural difference observed in figure 2 conformational or deposition based?*
*All structures look similar (although as stated - not identical), could this be due to the AFM*
*surface? Is there a way to quantify the impact of the surface potential provided by the AFM?*
*How “not identical” are these states? Is there a measure or just taking their word for it?*

We appreciate the reviewer's concern that the RNA molecules may deposit onto the AFM surface via
different molecular interfaces, and that the three selected particles in Fig. 2 could, in theory, be three different
orientations of the same/similar conformation. First, when it comes to AFM imaging, one of the advantages of most
RNA molecules is that they are relatively flat (i.e., not globular) with large and continuous polyanionic surfaces in
two dimensions, which become preferential and stable contact surfaces for electrostatic interaction with the
positively-charged APS-coated mica. Since both surfaces are universally charged, the deposition of RNA molecules
energetically favors maximal surface contact and, therefore, exhibits a very limited number of orientations. This is
well-supported in our AFM study of the cobalamin riboswitch (Nat Commun. 2023 Feb 9;14(1):714), where
thousands of particles were individually examined and classified into a limited number of classes, nearly all of
which laid flat on the mica surface. Furthermore, it has been shown abundantly that immobilization on a mica
surface does not alter structures of minor/major grooves or Holliday junctions, suggesting minimal perturbations to
nucleic acid structure (see references below).

- • Atomic Force Microscopy Imaging and Probing of DNA, Proteins, and Protein-DNA Complexes:
Silatrane Surface Chemistry. In: Leblanc, B., Moss, T. (eds) DNA-Protein Interactions. Methods in
Molecular Biology™, vol 543. Humana Press. 2009. https://doi.org/10.1007/978-1-60327-015-1_21
- • DNA structure and dynamics: an atomic force microscopy study. Cell Biochem Biophys 2004;41(1):75-
98. doi: 10.1385/CBB:41:1:075

Our findings demonstrate clearly that the three particles (P1, P2, P3) shown in Fig. 2 are in very different
conformations. The recapitulated structures of P1, P2, P3 show poor CC^{AFM} scores with the molecular surfaces *not*
used for fitting in each case (e.g., the CC score for the model of P1 against the molecular surfaces of P2 and P3 are
0.8 and 0.4, respectively), and have RMSDs ranging from 34 to 42 Å. **Redacted**

*Could the authors clarify that the resolution analysis (Fig. 2) is calculated for the (XY?)*
*direction, or XZ, YZ directions, as these should have variable precisions due to the higher*
*precision in the AFM Z-piezo? Is this the resolution value being used for the dynamic fitting*

*uncertainty taking both resolution values into account? Along these lines, would FSC (Fourier shell correlation) be*
*a better metric instead of FRC (Fourier ring correlation) for this?*

The spatial information used for dynamic fitting by CafeMol is from XYZ, where Z heights are the pixel
intensity. Therefore, FRC is the most appropriate metric in this case. The local resolution analysis was done in the
XY-plane using a Fourier filter with concentric rings spaced in 5-pixel increments (Extended Data Fig. 3). Of note,
this analysis is not intended to be used in the context of determining the resolution of the structure. Here, we are
simply using the local 3D spatial information (in pixels) to estimate the spatial resolution limit (in Å) of the AFM
image, which is used to guide the dynamic fitting through appropriate force potentials and fitting measurements.

*What are the trajectory models as an input? Are these a list of coarse-grain coordinates?*
*This point is picked up further below, but mainly arises due to the lack of clarity of the*
*workflow. Is this: PDB -> Do Sims -> Get Energies -> Cluster / DNN? As stated earlier,*
*clarification of this would be very useful for interested users. Clarification of both what is used*
*and what is needed would be extremely helpful.*

We thank the reviewer for the suggestion, and we have revised Fig.1 to clearly illustrate the HORNET
workflow. In summary, the structure determination by HORNET is performed in three main steps: 1) structure
calculation using a starting model and the AFM image to drive dynamic fitting, 2) structure evaluation and filtering
by DNN or UML/DNN, and 3) writing to file the final PDB coordinates of the top-scored structures. Regarding the
reviewer's question, the trajectory models input file used in DNN contains a list of energies terms and topology
scores of the models with their respective energy terms and CC^{AFM} scores.

*Fig. 4 states that the model with lowest val loss is used; is this epoch 6? Has the model just*
*got lucky at this early stage of learning as it is showing the lowest loss or is the model that's*
*really used one from the epoch nearer the centre of the graph?*

The selected trained model was from epoch 5, as it had the lowest loss value for the validation set. The
general behavior observed for the loss functions is as expected. In the early stages of learning, both the training and
the validation losses decrease when the learning is generalizable. After some time, however, the training loss
continues to decrease while the validation loss starts to increase, indicating that new information being learned by
the machine is no longer generalizable, resulting in overfitting of the model to the training set.

Although epoch 5 seems apparently to be very early in the learning, it is not. To achieve 5 epochs, all the
training data (~3.5 million entries) must have been used 5 times, amounting to ~17.5 million data processed. If we
divide this by the batch size of 128, it means that the neural network has optimized its weight parameters
(forward+backward propagation) approximately 137,000 times (iterations).

*UML cutoffs are empirically chosen after testing on BM 0(Methods), but how different can*
*these get in the wider picture of fitting different pdb structures, i.e. opposed to the smaller*
*BM5 / a different sample? Do these need to be recalculated for each different sample?*

All pre-UML cutoffs are done by removing outliers with high energy scores (for example, unstable
secondary structure) and/or poor CC^{AFM} scores (the structure does not correlate with the AFM topology). In this step,
all filters are applied using an average of less than 1σ of the total distribution. After PCA and clustering, the cluster
that has both the lowest energy scores (*Go, local, Total*) and the highest CC^{AFM} score is the winner (Fig. 1). In this
step, we are selecting the average of 1σ from all these components. As all cutoff filters used are intrinsic to RNA
structure and independent of RNA size, they should be generally applicable to any case. As the reviewer points out,
the standardized cutoff values were tested using BM0. However, extensive tests with both BM1 to BM5 (not only
BM5) showed good performance using the same cutoff values, indicating that the selected filters are generally
applicable (Fig.3). Nevertheless, the UML approach gives the user the flexibility to modify PCA analysis, cluster
size, and cutoff filters, as all these parameters could be further refined if necessary in the UML analysis code as an
input parameter.

*Normalisation values are based on the empirically chosen values of "theta_c = 5,*
*theta_stacking = 9 and theta_pairing = 9" to "achieve the optimal balance between enforcing*
*the integrity of primary and secondary structures (the hierarchical principle) and achieving the*
*best fit to the topological restraints at the same time". It is unclear how these were determined*
*in the methods.*

In simple cases of molecular dynamics simulations, structure-based potentials and constraints may be
sufficient to guide RNA structure. However, in the case of HORNET, the AFM pseudo-potential (*kappa* value) that
is imposed using the experimental image is critical to driving the trajectory toward a correct structure. Therefore, the
structure-based potentials must be properly weighted against the AFM-based potential in order to maintain the
integrity of the secondary structure as well as covalent geometry. We varied these scale factors (from 1 to 10) and
determined that the values of $\theta_c = 5$, $\theta_{stacking} = 9$ and $\theta_{pairing} = 9$ (p27 in Methods) were the most
suitable for achieving the proper balance within the range of kappa values (up to 50). Importantly, *Go* potential is set
to the lowest value (1.0) so that the calculation is not biased towards the initial structural model. Although we
expect this set of values to be universally applicable, it is possible that the use of different weighting factors may
achieve similar results in specific cases. For such cases, we have provided a script (with instructions in the
README) for training HORNET with data generated using different scaling factors. All this information has been
provided in the Methods and README files in the HORNET package.

For clarity, we have now included the update list for the HORNET package. This list is attached to the end
of this rebuttal as an Appendix.

*Minor comments:*

*The caption of Fig. 2d states the clear kinks for P1, P2 and P3 are highlighted by the dotted lines at ~0.87 nm-1*
*(11.5Å), 0.87 nm-1 (11.5Å), and 0.87 nm-1 (11.5Å), respectively. This is true for P1 but needs to be corrected for the*
*P2 and P3 values to match the graphs.*

*Page 15, line 20, typo: Determining RNase P RNA structures of heterogeneous conformations.*

*Fig. 6b: The y-axis is "HORNET - score", which refers to RMSD. Would it be better to change this axis label to*
*"Predicted RMSD" as in Fig. 5b?*

*URL links within the reference list appear to not be working.*

*There are some domain-specific terms ("weighted AFM pseudo potentials", "AFM force potential", "trajectory",*
*"trajectory structure model") that, if briefly explained, would improve the readability of the paper to those without*
*extensive expertise.*

We thank the reviewer for pointing these out. We have revised the manuscript and addressed all comments
accordingly.

35 36 **Appendix: HORNET Package –Update list**

The complete code review and changes can be found in the repository: [https://github.com/PNAI-CSB-NCI-](https://github.com/PNAI-CSB-NCI-NIH/HORNET/pull/8)
[NIH/HORNET/pull/8](https://github.com/PNAI-CSB-NCI-NIH/HORNET/pull/8). All data for the calculations are available at
<https://home.ccr.cancer.gov/csb/pnai/data/HorNet/>

To give an overview of what was changed at an upper level, please find a list of modifications below:

- • README file review: The readme file was carefully followed and reviewed by three persons who have
limited computation skills and know nothing about HORNET. The complete comparison between the
previous readme and the current can be found here (by pressing **ctrl** and double click it for **Windows**)
- • Tutorial Files: All the intermediate files necessary to run the package were included in a folder called
data/TUTORIAL. Now, all the initial, intermediate, and even final files are there to serve as examples. For
clarity, a list of input files is shown below, and black-colored names are the files that were already there,
and green ones are those included at this time (even files that were optional and not needed to run the
package).
 - ○ For input preparation:
 - ■ `En_allk14.txt`
 - ■ `En_allk22.txt`
 - ■ `Bases_k14.csv` (was an optional entry but included anyway)

- Bases_k22.csv (was an optional entry but included anyway)
 - For UML analysis:
 - Full_Trajectory.csv (intermediate file created in the input preparation)
 - For the Deep Learning predictions:
 - Prediction_sample.csv
 - Full_Trajectory.csv (intermediate file created in the input preparation)
 - Filtered_Data.csv (intermediate file created in the UML analysis)
 - Select_Cluster.csv (intermediate file created in the UML analysis)
 - Final_Cohort.csv (intermediate file created in the UML analysis)
 - For the Deep Learning training:
 - Train_sample1.csv
 - Train_sample2.csv
 - Validation_sample.csv (sample of our validation set)
 - As final outputs from the prediction:
 - Final_Cohort_Top10.csv (examples of output from prediction)
 - Final_Cohort_prediction.csv (examples of output from prediction)
- Package Install: Since users asked for it to be an installable package, we restructured the code to be in an installable format, mainly regrouping the previous scripts under UML and DNN folder to another folder called src/hornet. A setup.py file was included in the repository so that a user could simply install the package using the pip command from Python. Although we would not see this step as crucial to the work as it does not interfere with the ability of one to run the package, we agree that it brings more clarity and organization, being the main reason to update the code but as it takes important time to do when considering a publication, had not been done before.
- Unit Tests: Users requested the package to have unit tests, mainly to assure the operability of the scripts to run a complete pipeline as a test. We went further than just testing a pipeline of the 4 steps by creating a separate set of unit tests for all the functions we have in each module. This gives a total of:
 - 14 tests contained in 4 testing files for the INPUT module (data engineering).
 - 21 tests contained in 8 testing test files for the UML module (UML analysis).
 - 49 tests contained in 11 testing test files for the MODEL module (deep learning).
 - 52 tests contained in 4 testing test files for the wrapper of the scripts, where each complete step and parameter options of the pipeline is tested in each file, finally completing the whole pipeline of 4 steps using the data provided in the tutorial, assuring all of the code has been tested and can be run throughout without issues.

The bug found by users in one of the wrapper scripts was fixed and would not interfere with the inner core functionality. No new bug was found during the process of creating the unit tests. All the code modifications can be found in the PR link provided above.
- Containerization: Users had trouble installing the environment (install the environment is different from installing the python package we developed), so we created a docker image with a prepared environment already installed and containing the HORNET package inside as well. If one had trouble installing the environment this time, we are offering a clean environment for the package to be used, making it practically impossible for someone to fail the test. We also make the Docker File available, so any user can simply create their own containers if somehow needed, given that we provided the docker commands in the README to follow through. Importantly, the unit tests are automatically performed during the docker image building process. The complete building pipeline can be seen in the Docker file. *IMPORTANT NOTE: We do not have Copy Rights to distribute the Cafemol package (the dynamic fitting software), so it is not installed in the docker image we are releasing. For Cafemol, please go to <https://www.cafemol.org/> for instructions and the download.*
- We included in the updated code the selection of the best model giving the best loss on the validation set in the training script, also offering a sample of our validation set to follow exactly what was done in the paper.
- On the Dynamic Fitting side, the README file was also updated to include more explanations, and the complete comparison of the previous file and the new one can be found here.
- Practically all the Dynamic Fitting scripts were refactored to allow a user to run it locally, instead of using High Performance Computing environments, and all necessary example files and scripts were added to bring more clarity and convenience.

- To make it fully reproducible, we included a complete toy example where the data, simulated AFM image, initial crystal structure file (already centered over the AFM image) and configuration files are already set up for someone to just run the dynamic fitting using those files as input.
- For conversion of CG to all-atom model we provide a separate instruction with one example file that can be found here.

**Referee #2:**

*Degenhardt et al. use AFM and machine learning to address RNA structure and conformational heterogeneity. I*
*want to make clear that I think AFM is a phenomenal and underutilized approach for RNA and so I think the*
*authors' development of this approach is quite important. That said, I have questions and concerns on multiple*
*levels about the manuscript; these are listed below in rough order of occurrence in the manuscript, and I have tried*
*to address the level of importance of each concern. Overall, I think the level of clarity for the general reader is*
*currently insufficient, both to make clear what was done and how it was done. It is possible that with greater clarity*
*(or additional time spent on my part) some of these issues identified below would be resolved, so the authors should*
*definitely respond to these points.*

We thank the reviewer for many of his positive/constructive comments/suggestions. We also agree on the
need for clarifications in multiple places. We address the comments/questions/criticisms.

*The authors seem to assume (and cite one paper suggesting) that most mRNAs are*
*structured; it is not clear to me that this is valid and there are multiple questions about the*
*data that have been used to support this.*

After searching the text, we could not find any language or citations that make any assumption regarding
the percentage of mRNAs that are structured. Our entire study is completely based on, and only applicable to, folded
RNAs. This is clearly stated in the Abstract (lines 2-4) and Introduction (p. 3&4). If an mRNA of interest were
unstructured, this would become quickly evident upon inspection by AFM or other methods, and there would be no
reason to pursue structure calculation using HORNET.

*The single molecule studies of heterogeneity cited may have been subject to covalent*
*differences and surface interactions. Important: The possibility of surface or other handling effects does not seem to*
*have been addressed in this work.*

We appreciate the reviewer's concern regarding the possible effects of surface interactions. However, this
concern has been addressed in our previous publication on the conformational heterogeneity of the cobalamin
riboswitch (Nat. Comm 2023 Feb 9; 14(1):714) as well as many other publications by others that we have cited
adequately (see also the below for a few). Thus, for this reason, and because of space limitations, we did not address
this question in this manuscript but will do so here for review purposes.

The functionalization of mica with APS is widely used for the immobilization of nucleic acids primarily
through the electrostatic interactions between protonated amino groups of the APS-mica substrate and the negatively
charged nucleic acid backbone. The low charge density of APS-mica allows reliable imaging of nucleic acids,
protein-nucleic acid complexes and other biological samples, which has been reported extensively in the literature (a
few examples are provided below). Furthermore, it should be mentioned here that the reviewer's concern about
possible alteration of structure (both secondary and tertiary) has also been addressed extensively in the literature.
The structural features of minor/major grooves, Holliday junctions, and intricate interactions are intact on mica
surfaces, and the immobilized particles can even be induced to undergo structural transitions (DNA structure and
dynamics: an atomic force microscopy study. Cell Biochem Biophys 2004;41(1):75-98. doi:
10.1385/CBB:41:1:075.). Surface immobilization for AFM imaging is completely non-covalent, and there is no
reason to suspect covalent alterations in RNA structure during this procedure.

- • Shlyakhtenko, L.S. et al. Silatrane-based surface chemistry for immobilization of DNA, protein-DNA
complexes and other biological materials. Ultramicroscopy 97, 279-87 (2003).
- • Lyubchenko, Y.L., Shlyakhtenko, L.S. & Gall, A.A. Atomic force microscopy imaging and probing of
DNA, proteins, and protein DNA complexes: silatrane surface chemistry. Methods Mol Biol 543, 337-51
(2009).
- • Stumme-Diers, M.P., Stormberg, T., Sun, Z. & Lyubchenko, Y.L. Probing The Structure And Dynamics Of
Nucleosomes Using Atomic Force Microscopy Imaging. J Vis Exp (2019).
- • Shlyakhtenko and Lyubchenko. Mica Functionalization for Imaging of DNA and Protein-DNA Complexes
with Atomic Force Microscopy. Methods Mol Biol. 2013; 931: 10.1007/978-1-62703-056-4_14.

*There are other studies not cited that seem highly relevant and arguably more relevant than many that are cited. I*
*hesitated to raise these as they are from my work and those of collaborators, so I apologize if the list is biased and if*
*I am over-emphasizing their value:*

*a. NMR RDCs and chemical shifts have been used to obtain atomic-level ensembles for an RNA element. In addition*
*to this precedent, the degree to which atomic-level information will be needed to extract information of value or high*

*value is not addressed (i.e., the authors correctly state that AFM can give helices, but how much physical and*
*biological information can be derived at this level is not addressed).*

*Shi, H., Rangadurai, A., Assi, H.A., Roy, R., Case, D.A., Herschlag, D., Yesselman, J.D., Al-Hashimi, H.M. (2020)*
*Nat. Commun. 11, 5531. Rapid and Accurate Determination of Atomistic RNA Dynamic Ensemble Models Using*
*NMR and Structure Prediction. PMID: 33139729.*

*Ken, M.L., Roy, R., Geng, A., Ganse, L.R., Manghrani, A., Cullen, B.R., Schulze-Gahmen, U., Herschlag, D., Al-*
*Hashimi, H.M. (2023) Nature 617, 835-841. RNA Conformational Propensities Determine Cellular Activity. PMID:*
*37198487.*

We have added a statement in the Introduction citing the suggested references (p3/ln11). It is noteworthy to
point out that most of the current methods and techniques are suitable for probing ensemble behaviors of dynamic
molecules, as opposed to 3D structure determination of individual RNA conformers. smFRET probes
conformational dynamics at a single molecular level by providing a set of sparse distances but does not generate
explicit 3D atomistic structures. NMR-RDC or Au-SAXS experiments are useful for studying ensemble behaviors
of the total population in solution, and are thus conceptually different from single-molecule behavior using single-
molecule approaches. Moreover, there are many limitations associated with some of those methods, especially when
studying large RNA molecules. It would be impractical to measure RDCs given the expected short T2 and signal
overlapping of a large RNA such as the 417-nt RNase P RNA, as opposed to small RNA molecules like the HIV
TAR.

*In addition to this precedent, the degree to which atomic-level information will*
*be needed to extract information of value or high value is not addressed (i.e., the authors*
*correctly state that AFM can give helices, but how much physical and biological information*
*can be derived at this level is not addressed)."*

Specifically, HORNET needs an initial structure, which could differ from the ground-truth structures as
much as ~30 Å (S257, Extended Fig.11). The initial structures could be generated using programs such as FARFAR,
which only requires input of secondary structure information (Benchmarks 0, 1, 2 and S257), crystal/cryo-EM
structures (Case 1), or hybrid methods such as combining modeling with SAXS (Benchmarks 4 and 5; Case 2)
(Extended Fig.16). We used six benchmark cases to demonstrate our method. The degree to which atomic-level
information is needed for HORNET to work is fully illustrated. With all benchmarks plus two case studies, we show
HORNET is capable of determining topological structures with an accuracy of around 5 Å. At this accuracy, domain
motions in the RNase P RNA case and flexibility among five conformers in the HIV-1 RRE RNA can be seen (see
the response to the relevant comments in the later sections). The results reveal the RNase P RNA in three different
conformational states and the structure-dynamics basis for the trans-dominant negative RevM10 mutant activity and
the architectural complementarity and mutual conformational adaptability between RRE and the Rev protein. These
findings are highly significant in HIV biology and could not be obtained from any other methods.

As we have demonstrated in this study, HORNET's ability to determine 3D topological structures of
individual RNA conformers is highly significant for the following reasons. 1. It shows heterogeneous RNA
conformations in solution at the single-molecule level with an unprecedented level of detail. Prior to this study, our
knowledge about RNA conformational heterogeneity mostly came from indirect measurements of ensemble
behaviors, such as RDC/chemical shifts/PRE from NMR. Single molecule FRET (smFRET) is about the only
experimental approach for probing RNA conformational dynamics at the level of individual molecules, but such
information only provides conformational dynamics in terms of fluctuating distances between FRET pairs, not
explicit atomistic 3D structural models. 2. From AFM images, our method makes it possible to map the full
conformational space of RNA, from which one can derive motion amplitudes and directions of structural elements,
correlations between the RNA conformational dynamics and activity, and between the sequence conservation and
3D conformational conservation. **Redacted**

*As the authors are aware, the Au-SAXS approach developed by Harbury provides a*
*distribution of distances that directly reflects the distribution in solution. This information is*
*more precise than the averaging in 'regular' SAXS and has revealed heterogeneity in the*
*past. It would seem that using RNAs with prior Au-SAXS data and using Au-SAXS to test*

*structural models would be better validations and tests of the approach than what was done in*
*this manuscript. In addition one could use different solution conditions and ensure (test) that*
*the molecular structures obtained change as predicted.*

We do not dispute the usefulness of Au-SAXS in describing an ensemble behavior of conformational
diversity of small and simple systems such as the TAR RNA. However, it is not practical to study much larger and
more complex RNAs with multiple flexible domains. This is because Au-SAXS requires specific labeling to
incorporate pairs of Au nanoparticles, which could be practically challenging for large RNAs. In addition, the
flexibility of the linkers could complicate data interpretation. Probably for these reasons, no Au-SAXS experiment
has been demonstrated for a large RNA, such as the 417-nt RNase P RNA used in our studies.

With regard to the utility of SAXS and Au-SAXS for the validation of HORNET-derived structures, the
authors see no difference in these two methods for such a purpose. However, As we have illustrated in one of our
recent publications (Nat Commun. 2023 Feb 9;14(1):714), orthogonal analysis using SAXS corroborated by ITC
data is a simpler and more practical approach to validating structural models and populations.

*This point raises an additional weakness (as I understand it) as individual molecules are chosen for deeper study:*
*do the authors obtain information about the entire population of molecules? That would seem needed for the*
*approach to have significant value, and one would need to show that the population obtained on the grid reflects*
*that in solution (so a great way to do this would seem to be to vary the solution conditions for RNAs like kink turns*
*that have a distribution of shapes are known to change shape with salt).*

The reviewer brings up a good point, i.e., validation of conformational space consisting of all RNA
conformers using an orthogonal method. We have addressed this question in a previous publication (Nat Commun.
2023 Feb 9;14(1):714), where the SAXS curves of the cobalamin riboswitch in both the absence and presence of
ligand show that the ensemble behavior of the heterogeneous conformation populations in solution cannot be fit by
any single conformer, including the crystal structure. Instead, these data could only be fit using a curve that was
back-calculated from the volume fraction of the various conformers observed by AFM. The percentages of each
conformer population tallied from the AFM images were consistent with those derived from the SAXS data. The
focus of this current manuscript, however, is to establish a novel method for recapitulating 3D structures from AFM
images and estimate their accuracy, and to test the robustness and utility of the methodology in various realistic
scenarios. As such, the potential utility of HORNET to study systematically ensemble populations is outside the
scope. For an example of this application, including the use of SAXS for validation, we refer the reviewer to the
unpublished study of “*Conformational space of RNA in solution*” using HORNET.

*Without information about the distribution of shapes how does one know that the three molecules chosen for this*
*study are representative? This is an issue (unresolved) in cryo-EM as well; there researchers do obtain multiple*
*states but also leave out many (most?) of the molecules and also may approximate continuous distributions by a*
*discrete number of states that are models that sample the more continuous range of conformers explored. Shi, X.S.,*
*Huang, L., Lilley, D.M.J., Harbury, P.A.B., Herschlag, D. (2016) Nat. Chem. Biol. 12, 146-152. The Solution*
*Structural Ensembles of RNA Kink-turn Motifs and Their Protein Complexes. PMCID: PMC4755865. See also:*
*Wang, Y-X. (2016) Nat. Chem. Biol. 12, 126-127. News and Views. RNA Conformation: Lightening Up Invisible*
*States.*

The three RNase P RNA particles shown in Fig.2 and Fig.5 were selected because they represent three
different conformational states, and were therefore useful for demonstrating the utility of HORNET to determine
distinct conformations of an RNA whose complete structure has never been reported. Once again, the primary focus
of this manuscript is to present a full description and validation of a method that can recapitulate 3D topological
structures of individual conformers using AFM and DNN. We did not claim that the three conformers represent the
total conformational space of the RNA.

The use of AFM to capture all representative conformers has already been demonstrated and validated in
our previous publication (Nat Commun. 2023 Feb 9;14(1):714). The heterogeneous conformation landscape of
cobalamin riboswitch RNA is also consistent with our cryo-EM study (Nucleic Acids Research, 2023.
<https://doi.org/10.1093/nar/gkad651>). Therefore, the inclusion of such information the reviewer is requesting would
be redundant with work that is already published. **Redacted**

*The authors suggest that heterogeneity is the cause of the limited number of RNA structures in the PDB; the*
*possibility that there may be (many) fewer structured RNAs than proteins and recent breakthroughs in RNA cryo-EM*

*are not noted:*

*“Thus, a single snapshot structure falls short of accurately describing the conformational landscape associated with*
*its function as clearly demonstrated recently in a cellular context².”*

In the original version of the manuscript, we cited the recent progress in cryo-EM by Reference 11 and the
reference reporting conformational propensity in the cellular environment by Reference 3.

*Here the authors do not cite a recent Nature paper (by us) that shows that accounting for the landscape -via NMR*
*approaches- can quantitatively predict cellular function. It is reasonable to suggest that additional (and even better)*
*approaches are needed, but not to imply that current approaches cannot do this at all.*

*NOTE: Journal rules I believe limit the number of citations allowed and that may be responsible for this omission.*
*Such rules by journals are counter to scientific communication and should be abolished.*

In the original version of the manuscript, the paper Ken, M. L. *et al.* RNA conformational propensities
determine cellular activity. *Nature* (2023) was cited.

*The approach. Some points relevant to this were brought up in #1 above. The main*
*additional point is concerns about mixing the data (AFM) with models (forces, 2o structure*
*predictions or assumptions, AI models) so that in the end one doesn't have a model that*
*comes from the data but rather some mixture of data and model; that (I think) is hard or*
*impossible to unravel. While the test cases used are needed they are not sufficient as they*
*are idealized and do not demonstrate that the process works with real RNA molecules and*
*real data, only idealized cases.*

There seems to be a point of confusion regarding how HORNET is used in deriving 3D structures from
AFM data. Incorporating information from multiple sources (e.g., experimental data, starting models, appropriate
constraints/restraints, even AI) is fundamental to any structure-determination method, including NMR,
crystallography, and cryo-EM. Dynamic fitting of a preliminary model to an AFM-imaged particle, using
appropriate energetic and geometric constraints, is no different than fitting and refining a model (now almost
exclusively generated by AI for proteins) against an X-ray or cryo-EM electron density map. The same is true for
structure determination using NMR-derived restraints mixed with standard covalent geometry, such as torsion
angles, dihedral angles and bond lengths, and conformation restraints such as H-bond distances and planarity of base
pairs in duplexes. Then, these structures are optimized or validated based on 1) agreement with experimental data
(e.g, Rfactor, *CC*), and 2) model accuracy by comparison to expected/known values (e.g., RMSD bonds/angles,
clash score). In the same way, HORNET-derived structures are selected/validated based on 1) agreement with the
AFM topography (*CC^{AFM}*), and 2) minimal violations or deviations from expected values (lowest energy scores),
based on a generalizable neural network trained using known structural information. As we have mentioned in the
manuscript, no structure of a biomacromolecule has ever been determined using experimental data truly from a
single molecule, as existing structure-determination methods require coherent signals averaged over many thousands
of molecules in near-identical conformations.

For the benchmark RNAs, the crystal structures were used to assess the accuracy of models generated by
HORNET (within ~5 Å RMSD). Based on its trained DNN, HORNET thus provides an estimated accuracy for each
model, even in cases where the ground-truth structure is unknown. This leads to the reviewer's last point in the
comment regarding the insufficient nature of RNAs used to test and validate the method. We are unaware of any
RNA or experimental data used in this study that would be considered idealized and not suitable for representing
“real” case scenarios. The smallest benchmark RNA used is 210 nt, which is far from “ideal”, and its folding
architecture is far more complex than, for example, two duplexes connected via a flexible internal asymmetrical
bulge that has been studied extensively. Thus, it would have been much easier (but less significant) for the authors to
have selected small RNAs with very simple structures to benchmark and test HORNET. The benchmark RNAs
(RNase P, cobalamin riboswitch, and group-II intron) are very diverse in shape, size, and function, and are the most
representative RNAs > 200 nt in the current structure database. Moreover, the starting models for these benchmarks
were generated from three different low-resolution methods, including predicted models of RNase P RNA using the
popular program FARFAR, a trajectory model of the cobalamin riboswitch derived from MD trajectory, and models
of RNase P RNA and Group II Intron derived from SAXS data. Notably, all these initial structural models are far
from their respective ground-truth crystal structures (Extended Data Fig.11), and HORNET was capable of
determining the structures with an accuracy better than 5 Å RMSD. Lastly, HORNET was further tested using
experimental AFM data for two very different large RNAs (417-nt RNase P RNA and 232-nt RRE RNA). Of note,
the full-length structures of these two RNAs have never been determined, and we presume, therefore, that they
would meet any definition of “real” test cases. Importantly, the AFM-imaged particles of RRE and their structures

calculated by HORNET are fully supportive of the SAXS-derived topological structure of RRE (Fang, et al., Cell,
2013, V. 155), which serves as a meaningful validation for an RNA with an undetermined 3D-structure and that was
never seen by the trained DNN.

*The limitation alluded to above of mixing AI models with data is in part analogous to limitations from a recent*
*Science paper (Das & Dror) that showed that AI can help select from FARFAR-derived structures to find the correct*
*ones. However, most reading the paper seems*
*to conclude that AI can now predict RNA structure, which is very different than what was shown. Here again, it*
*seems that structure is not predicted but rather structural models generated from knowledge-based models*
*(ironically, based on RNA PDB structures) can be sorted and selected by applying AI to AFM data.*

Regarding the confusion with “mixing AI models with data,” please also see the response to one of the
previous comments (p11/line25). As for the comparison between HORNET and the ARES approach (Townshend et
al., Science, 2021), here are some key differences and hope the reviewer will appreciate the novelty of our work:

1. Conceptually, ARES is a structure prediction program for small RNA based on the information from structural
databases, whereas HORNET determines 3D structures of large RNA based on experimental AFM images;

2. ARES followed the general concept, assumption and methodology of AlfaFold for structure prediction, i.e., for one
primary sequence, there is THE one corresponding 3D structure, whereas HORNET totally differs from AlfaFold and
ARES, and considers one primary sequence could fold into different conformations and determines structures of
individual conformers. The notion of one sequence to one RNA structure/fold/conformation is incorrect for RNA,
especially for large multi-domain RNAs as we have demonstrated in a recent publication (Ding, et al., Nat Commun.
2023).

3. ARES uses conventional discrete fragments from structural databases and can only sample a limited and discrete
conformational space, whereas HORNET uses a pseudo-structural database with continuous conformational
trajectories.

4. Considering neural networks, HORNET uses a novel architecture that is totally different from that of ARES and
performs remarkably better than ARES. It is noteworthy to mention that the intended only goal of the ARES approach
is to use the ARES scoring function to estimate the accuracy of a predicted structural model. As shown in their
manuscript in the Supplementary Information, the ARES scores predict RMSD poorly in general and fail in cases of
larger RNA, indicating that either the model of their geometric deep learning is inadequate, or the fragment-based
structure database is not sufficient, or both. In contrast, HORNET is capable of estimating accuracy (RMSD) in all
benchmarking and testing cases. A more detailed comparison in terms of accuracy prediction is listed at the end of
the response.

For more detailed comparison, please see **Appendix** at the end of the rebuttal.

*The possibility of alternative secondary structures does not seem to be taken into account.*
*Also, there seems to be an assumption that species fall along (rather than some being*
*orthogonal to) a folding pathway that progresses from unfolded to full folded.*

The dynamic fitting by HORNET takes all available information, including the secondary structure. In the
current version of HORNET, it only takes in one secondary structure scheme into calculation. In the cases in which
the AFM particle exhibits alternative secondary structures, HORNET will not reach convergence on a model that
adequately fits the molecular surface. In future releases, we could in principle incorporate the information on
multiple secondary structures as input and let the DNN decide.

With regard to folding pathways, HORNET applies energy constraints that enforce the RNA hierarchical
folding principle together with the AFM restraint. While it is possible that some RNAs may not fold along a
“correct” pathway without sufficient restraints, we have found that with a long trajectory of dynamic fitting together
with appropriately applied strong AFM potential, the initial model with wrong folding could still be folded into the
correct final structure as we have shown in this study with RNA S257 as an example (p16 lines 9-19 , Extended
Data Fig.11, p13-15). One point we need to make here is that HORNET is not designed to “fold” an RNA starting
from an unfolded primary structure, as in the case of structure prediction.

*There does not seem to be discussion or controls for surface effects.*

This comment has already been addressed in the response to a previous comment by the reviewer. We
have added a paragraph to the main text (p4 lines1-2), with appropriate citations, to address this concern.

*Returning to establishing the method, there does not seem to be “ground truth” data of a*
*multi-state system that was used and independently determined via AFM-AI. While the results*
*are reasonable, they appear to not meet this highest standard. Arguably this was done in the*
*authors’ Nat. Commun. paper with a riboswitch (WT and mutants and showing transitions upon ligand binding) so*
*minimally discussion of this critical technical point I think is important*
*to include.*

We respectfully disagree with the reviewer on this point. Indeed, an adequate number of multi-state systems
with independently determined structures for each state would provide the best standard for developing and testing
the model. However, no such set of high-resolution structures has been determined by using information solely from
a “single” molecule (as opposed to information from the ensemble average) that covers an RNA’s conformational
landscape. Using HORNET/AFM-derived structures as the ground-truth data to establish and validate the method
would be a circular and invariable approach. This is why the crystal structures of large RNAs (RNase P, Group II
Intron and cobalamin riboswitch) with diverse starting conformations (BM0-BM5) were used to generate the >65
million initial trajectory structural models for training and benchmarking the DNN, which is designed to reliably
cover a broad conformational space and to be generalizable to all RNAs.

We believe that the ultimate test for validating HORNET is to reliably determine (estimated accuracy < 5
Å) the atomistic 3D structures of biologically significant RNAs whose complete structures have never been
determined. Our results meet this standard. In this study, we have determined novel structures of the full-length
RNase P RNA in three distinct conformations and novel structures of the HIV RRE RNA in five different
conformational states, all of which have not been possible to determine by other methods previously.

**Redacted** The observation of the heterogeneous RRE conformers in terms of
the distances between the two (IA and IIB) Rev binding sites and the heterogeneous distances between the two
arginine-rich regions at the N-termini of the Rev dimer led to a hypothesis of a novel recognition mode between the
HIV-1 RRE and the Rev dimer, which is achieved via architectural complementarity and mutual conformational
adaptability. To confirm this hypothesis, we designed a novel peptide where two arginine-rich motifs (ARMs) are
linked by a flexible linker. Our simulated calculation indicates that the distance between the two ARMs in the
peptide can sample a wide range of distances ranging from 20-70 Å, similar to the range of distances between the
two Rev binding sites in the RRE conformers (Fig. 6d&e). This architectural complementarity and mutual
conformational adaptability between the RRE and Rev dimer ensures high specificity and affinity, as shown by the
gel electrophoresis mobility shift assay. The faster mobility of the RRE-peptide complex indicates a more compact
overall structure, which is confirmed by the direct visualization by AFM of RRE particles with and without peptide
(Extended Data Fig. 19). Moreover, the highest degree of conformational flexibility, indicated by the highest RMSF,
was discovered in the region containing residues 100-190 (Fig.6c). This region has been implicated in forming
alternative conformations associated with the trans-dominant negative behavior of the RevM10 mutant (Sherpa, C.
et al., *NAR*, 2015. Vol. 43).

*Validation (i.e., testing) via SAXS seems less than ideal, given that it does not return or encompass (much) atomic-*
*level structural information. It seems that Au-SAXS results could be predicted and tested, though sample*

*preparation for that technique is time consuming (though can still be done in <1-2 weeks).*

This has been addressed in the response to a previous comment regarding the use of SAXS data (p10, in
this rebuttal). **Redacted**

*The main conclusion seems to be that the structural results of an RRE dimer with binding sites facing it led to the*
*design of tethered ligands. The authors state that a “long-standing mystery” is resolved, but do not clearly*
*articulate what this mystery (perhaps for reasons of space limitations, but this is needed to evaluate the significance*
*of this results). Also, I was not able to (readily) glean how the conclusion was reached of less heterogeneity after*
*ligand binding (in supplemental material); minimally clearer presentation would help. A related question that the*
*authors likely have information for is whether different A-like structures are favored with different length tethers;*
*such information would provide information about the energetics of the conformational landscape, albeit on a crude*
*level.*

*The tethered peptide experiment is somewhat analogous to an old Uhlenbeck hammerhead ribozyme experiment*
*where they tethered two hammerhead RNA arms (by making it a continuous oligo I think) to assess the bend angle of*
*the active form. That experiment was not fully successful but might be considered and cited here. More broadly,*

*could the tethering experiment been done “blindly” (or based on the A-like structure derived from SAXS) without*
*AFM to see what length tethers give the strongest binding?*

First, we are not sure what the reviewer meant by referring “RRE dimer”. The RRE RNA functions as a monomer.
Second, although the findings and conclusions pertaining to RRE are highly significant, and provide the valuable
test case for HORNET, they are not the main conclusion of the study as a whole as the reviewer refers. The main
focus and conclusion of our study is to establish and validate a novel method, HORNET, for studying RNA
conformational dynamics, and HORNET is the tool to determine 3D atomistic structures of individual conformers
with reasonable accuracy. This method uses experimental AFM images of RNA and a DNN to determine 3D
topological structures of individual RNA molecules in distinct conformations and estimate the accuracy of those
structures. The model was trained and validated using a total of six benchmarks, comprised of RNAs with different
sizes, biological functions, and methods for deriving their starting models. We then further demonstrated the
robustness of HORNET by determining three novel structures of the full-length RNase P RNA and five novel
structures of the HIV-1 RRE using experimental AFM images, the latter of which addresses important long-standing
questions about the structure-dynamic basis for the mode of binding between RRE and Rev protein. For a more
detailed discussion about the RRE results, please refer to our response to a previous comment (p13 in this rebuttal).

About “*Also, I was not able to (readily) glean how the conclusion was reached of less heterogeneity after*
*ligand binding (in supplemental material); minimally clearer presentation would help.*”, we have revised the
manuscript with a new figure (Extended Data Fig. 19) to address the reviewer’s suggestion.

About Uhlenbeck’s tethered ribozyme study, there are many tethered designs of various compounds or
molecules for various purposes and applications. Our novel peptide was designed to have flexibility that
complements both the architecture and dynamics of the RRE conformers and thus is unrelated to the Uhlenbeck’s
design and purposes in terms of both the applications and chemical makeup.

For a more detailed discussion about the RRE results, please also see our responses to previous comments.

*More information for figures and more table would help. E.g., how many total images; how*
*many processed; labels on some of the figures are missing or not fully defined. While this is*
*hard to do with so much data and so many steps, in some ways it is even more important.*

We thank the reviewer for this suggestion. We have provided additional details and clarity in the figure
captions, and we have added a table that describes the key information for each of the benchmarks and test cases
(Table of Content p50-52; Extended Data Table 1).

*As a ML/AI novice, I was not able to readily follow the approaches used, the reasons for these*
*approaches (over others), what the evidence is that they “really” work, and what information (if*
*any beyond empirical) is derived from the AI. Related to this, while numbers are improved (R^2 etc) it is not clear*
*(to me) what a significant improvement is or whether this was considered. E.g., there are some important papers in*
*algorithm development (I think for protein structure prediction) that suggest most “improvements” from new*
*algorithms instead arise from noise in fitting and so are not real.*

The authors understand the reviewer’s concerns and questions related to machine-learning and AI-based
models. We hope that our responses to the reviewer’s other comments have clarified any confusion. In contrast to
AlphaFold and FARFAR/ARES, HORNET is not a structure-prediction algorithm, as we have stated in the
Introduction. HORNET is a program for determining 3D structures of individual conformers observed by AFM and
estimating the accuracy of those structures using a DNN. It is paramount that this distinction be clearly understood.
The metric for “improvement” or whether HORNET really “works” is its ability to converge on a structural model
that simultaneously satisfies the imposed constraints (energy terms, geometry, etc.) and the molecular surface
derived from the experimental AFM image, each of which is independently scored and optimized. Then, the trained
AI model uses the rich signal-to-noise image to evaluate, sort, and filter those conformations to select the top
structures and provide an estimate of their accuracy. This is quite different from AI-driven structure prediction from
a primary sequence, where the maximum likelihood algorithms do not incorporate optimized fitting to experimental
data.

AlphaFold, indeed, has been a game-changer in protein structure prediction. However, RNA structure
prediction is far more challenging due to its chemical and structural degeneracy, often broad heterogenous folding
landscape, and the paucity of high-resolution structural information. The “one sequence = one structure” principle
that is mostly true for proteins is mostly *not* true for RNAs. Furthermore, FARFAR2 has shown to be successful in

cases of small RNAs (< 80 nt), but is not suitable for larger RNAs that have more complex and heterogeneous
folding landscapes. The authors of FARFAR2 even state that their goal for structure prediction was RNAs up to 200
nt (Watkins et al., 2020, Structure 28, 963–976). For larger RNAs, such as those presented in our study, experimental
data are necessary that can provide structural information about individual conformers.

Please also see our response to the previous related comment in p11-12.

*The overarching goal seems to be to develop a means to model RNA (at the individual molecule level) so*
*that this information can be used more broadly and generally to solve the (multi-state) RNA folding problem. Given*
*that the data are not on the atomic scale and given that the structures obtained are derived from a mixture of the*
*data and the model, it is not clear to me that this goal is achievable by following this path or worth achieving alone*
*without thermodynamic and/or kinetic data. E.g., will AFM be more valuable for dissecting transitions between*
*states (over conditions and time), providing a highly direct connection of structures formed to their underlying*
*energetics? Further, such AFM exploration could identify conditions that favor different states and thus can be used*
*to learn about each of the underlying states.*

We will address the individual comments in individual pieces.

*i. Given that the data are not on the atomic scale...*

Please see our response to the reviewer's 4th comment (p9 in this rebuttal).

*ii. ...given that the structures obtained are derived from a mixture of the data and the model...*

This comment has been addressed in response to earlier comments (p11 in this rebuttal).

*iii. ...it is not clear to me that this goal is achievable by following this path or worth achieving alone without*
*thermodynamic and/or kinetic data.*

It is clear that structural information alone from any source, either high- or low-resolution techniques, will
not be sufficient to understand the energetics of RNA folding pathways, and the addition of other data (biophysical,
thermodynamic, kinetic, etc.) would be required to understand the structures and conformational heterogeneity
observed. There is no dispute here.

*iv. E.g., will AFM be more valuable for dissecting transitions between states (over conditions and time), providing a*
*highly direct connection of structures formed to their underlying energetics? Further, such AFM exploration could*
*identify conditions that favor different states and thus can be used to learn about each of the underlying states."*

As this study represents the unveiling of HORNET, its scope of applicability and potential information
derived is uncharted territory. The reviewer's examples of such valuable information that AFM-derived structures
may provide are highly plausible. However, the authors choose not to (overly)speculate on this matter, but would
rather rely on experimental studies. **Redacted**

**Appendix. Comparing ARES and HORNET**

ARES represents a significant improvement over all existing programs for predicting RNA structures. Nevertheless,
for review purposes, we sum up the comparison of the two software. In doing so, we have no intention to criticize the
ARES approach but provide facts that could help reviewers understand why the pure-computational approaches will
fail and start to appreciate the urgent need for different thinking and approaches like HORNET.

Townshend et al. use the Atomic Rotationally Equivariant Scorer (ARES) function and machine learning approach to
estimate the accuracy of predicted structural models, whereas HORNET is the package for determining 3D atomistic
topological structures of individual RNA conformers. They differ in several fundamental and important aspects that
define their purposes and limitations.

**The purpose**

ARES is designed to estimate the accuracy of computationally predicted structural models using geometric deep
learning, whereas HORNET is the package to determine 3D atomistic topological structures and estimate the accuracy
of individual conformers using AFM images of particles and deep neural networks.

**Basic premise**

The ARES approach, like almost every other structural prediction software, assumes that for one primary sequence,
there is THE one corresponding 3D structure. HORNET has no such assumption and determines the structures of

individual conformers based on experimental topographic particle images. As we have demonstrated previously by
direct visualization, the assumption that one sequence leads to one 3D structure of RNA simply is contrary to
experimental observation via direct visualization by AFM and against RNA structural thermodynamics that govern
RNA folding. The fundamental reason for one RNA sequence to fold into possible multiple conformations is the
energetic degeneracy and geometric equivalency among RNA conformers due to its ragged energy landscape and
limited variety of the four building blocks. The only way to overcome energetic and geometric degeneracies is to
include global geometric/topographic constraints that delineate the topological folds of individual conformers. It is
noteworthy to point out that Townshend et al. appeared to realize the limitations of their approach and the importance
of experimental global constraints (the last second paragraph (*Townshend et al., Science, 2021*)).

**Structural database**

The ARES approach makes use of the fragment-based structural database (18 fragments), whereas HORNET uses a
pseudo structure database. The former is discrete and samples limited conformational space, whereas the latter is
continuous and samples much broader conformational space. As a result, the ARES approach may perform poorly
when a structure falls outside of its conformational space defined by its database such as it happens in some of the
puzzles.

**Deep learning**

The only similarity between the two programs is that both use deep learning. While ARES focused on the geometrical
aspects of the conformers for the scoring function hence applying a geometrical deep learning approach ending with
dense layers, HORNET focus purely on the energy and agreement to the experimental AFM image using dense layers
alone. In their paper, there is no plot showing the loss function as a function of the number of epochs since ARES was
trained using a single epoch - as stated in their supplementary materials (pg. 7). This means that it must have been
hard for them to validate their model as, even applying the best techniques, they were possibly already experiencing
overfitting in the very first epoch between the training set and validation set, pointing to a possibly less generalizable
model: “The set of hyperparameter values with the lowest loss was a batch size of 16, 1,000 randomly selected
structural models for each sequence, a learning rate of 0.01, and a single training epoch.” Nowadays, in light of the
new advancements in observing the dynamics of RNAs it is clear that an approach using a global constraint seems to
be more appropriate. In contrast, HORNET model shows all proofs that the model is being able to generalize its
learning, by following the general and expected loss pattern between datasets. With the loss curve we could find the
best weights given the validation set while also really tuning the hyperparameters, where the loss is shown in Figure
4.

More importantly, in estimating the accuracy of structural models using their AI model, which is the ultimate intended
goal of ARES, the ARES scores predict RMSD poorly in general and fail in cases of larger, indicating that either the
model of their geometric deep learning is inadequate, or the fragment-based structure database is not sufficient, or
both. In contrast, HORNET is capable of estimating accuracy (RMSD) well in all benchmarking and testing cases.

**Some detailed comparison**

Here are some detailed comparisons of the performance of ARES and HORNET. As one can see the key differences
in the fundamental aspects between the two programs lead to the dramatic difference in estimating accuracy of
structural models that are generated by either computational prediction or determination using AFM data.

Figure S4 from ARES supplementary information (pg. 15) shows the correlation between the ARES predictions
(score) vs the real RMSD for the 21 RNA puzzles (blind tests). As shown in their plot, the scoring function predicts
scores in the order of 3-10 Å where in many cases there are real RMSDs up to 40 Å, and ARES cannot accurately
predict the real value of the RMSD. This is fine as the authors did not claim in the end that ARES is predicting the
accuracy but rather acting as a scorer. Our model is not only acting as a scorer, but it can provide accuracy estimation
of the structure with reasonably small uncertainty (2.5 Å – see Fig. 4, pg. 14 of our manuscript) and even determine
nonconverged structure calculations such as S257 (Extended Data Figures 11, 12 and 13). More importantly, as RNA
gets larger, there could be more than one conformer for a given RNA primary sequence and ARES could not possibly
predict correct conformer structures because of energetic and geometric degeneracies.

Continuing in the same Figure S4 from ARES supplementary information (pg. 15), in most of the cases there is even
a poor correlation between ARES score and the true RMSD, which can be seen by eyes but also directly evaluated by
some very low Spearman correlation (average of Spearman correlation of 0.27 considering all puzzles). Our blind

tests show a clear relationship between the real and estimated RMSD, achieving a Pearson score of 0.77 and 0.84 on
our blind tests.
Table S5 from ARES supplementary information (pg. 29-33) compares the best ARES scored structures to other
scoring functions for 4 different RNAs for the RNA-puzzles (A, B, C and D). In summary, their results of the best
scored structures as a function of the real RMSD are as follows:
8 A: The average RMSD of the top 10 structures is ~ 14.8 Å, with structures ranging from 4.8 – 21.8 Å (111 nt, 36.2
9 kDa)
B: Average RMSD of top 10 structures is ~ 16.9 Å, with structures ranging from 12.5 – 22.7 Å (130 nt, 48.1 kDa)
C: Average RMSD of top 10 structures is ~ 12.2 Å, with structures ranging from 9.5 – 15.1 Å (230 nt, 78.9 kDa)
D: Average RMSD of top 10 structures is ~ 28.5 Å, with structures ranging from 14.5 – 35.9 Å (175 nt, 56.7 kDa)
In comparison, in our blind tests we were able to achieve the estimation of RMSD correctly (< 2.5 Å of difference)
and ALL the top 10 structures selected had < 5 Å of real RMSD to the GT, not only one or other structures.

**Referee #3:**

*Overall, this work brought up some interesting ideas in combining the atomic force microscopy (AFM) images with*
*machine learning techniques to determine RNA conformational diversity. However, the machine learning techniques*
*used in this work were very classical and simple. There are several concerns about the modelling itself regarding*
*both the features and the sample/testing dataset. It is also not convincing that this method can be widely applied to*
*dissect RNA structure conformations*

We thank the reviewer for appreciating the scope of this work. However, the reviewer’s comment “*the*
*machine learning techniques used in this work were very classical and simple,*” appears unwarranted since we make
no claim of novelty in developing new or sophisticated machine learning algorithms, nor is it the focus of the
manuscript. The authors would argue that the success of a machine-learning model is not determined by its level of
complexity or simplicity, but rather by its ability to achieve desired results. One could further argue that the simplest
model that achieves the desired results is actually the best model when considering computation time and resources.
If the reviewer is speaking generically about the complexity of the model in terms of hyperparameters, the authors
would argue that the complexity was determined by the long step of hyperparameter tuning of the network, not as a
choice. If the reviewer is speaking about the usage of more sophisticated architectures such as convolutional neural
networks, vision transformers or others, the authors would argue that since they are not directly dealing with the
image itself as an input parameter, but the cross-correlation value from the comparison of the structure to the AFM
image, these techniques are not appropriated to the current problem and input parameters. As the title of this paper
suggests, the authors used a purely holistic approach using energy-related and topological cross-correlation values,
and one of the novelties is that this information alone was already sufficient. A common practice in machine-
learning engineering is to increase the complexity of the model only if it fails to learn the data patterns on the
training set. Our model has complexity appropriate to perform the work for which it was designed, and is best suited
to address our regression problem, given the type of input parameters of this study. Of note, the technique we used
for dataset splitting is also a technique commonly used in complex models for computer vision and speech
recognition applications when dealing with data from different distributions.

Our method represents a novel application of machine learning to resolve an important problem in RNA
structural biology, the structure determination of individual RNA conformers. Moreover, as proof of concept, we
determined eight novel structures of RNA conformers, 3 RNase P RNA and 5 RRE RNA. Briefly, HORNET
requires only two things: 1) experimental AFM data, and 2) a starting model, obtained by any means, which can be
very different from the ground-truth structure. Given these two basic requirements, HORNET can be applied to
virtually any folded RNA for which an AFM image and starting model can be obtained. Thus, the authors would
argue that the method is widely applicable.

To help ensure the machine-learning was generalizable, we selected as benchmarks different types of RNAs
with available structures that were not redundant in terms of size, sequence, or fold. Those structures are the RNase
P RNA catalytic domain, Group II intron, and the cobalamin riboswitch. We then applied HORNET to two
experimental test cases: the full-length RNase P RNA, whose structure is only partially determined (298-nt of 417-37
nt) by crystallography, and the HIV-1 RRE, whose heterogeneous structures and conformers have never been
determined and are totally unrelated to all molecules in the training and benchmarking. In the case of RRE, the
general folding is in agreement with the previously determined RRE envelope. In each test case, HORNET was
applied to multiple particles (3 particles of RNase P, and 5 particles of RRE) to demonstrate the utility of HORNET
in deriving distinct structure conformations from individual AFM micrographs. **Redacted**
Instead, the current manuscript focuses on presenting the
methodology and required validation.

*It is also not clear about the novel impacts of this method, e.g. whether this method could determine new functional* 49
RNA structure conformation that was not possibly determined previously, and whether it can determine the ratio

*between different conformations.*

The direct response to the reviewer’s comment is “Yes” as we have shown with the case study of the HIV-1
RRE RNA. This is a highly functionally important RNA whose 3D structures of any conformers have been eluded
from determination by existing methods for several decades. We used HORNET to determine the novel structures of
the HIV RRE RNA in five different conformational states, all of which have not been possible by existing methods
previously. The flexible conformations in terms of the distances between the IA and IIB Rev-binding sites reveal a

novel recognition mode between the HIV RRE RNA and Rev protein, achieved through architectural complementarity
and mutual conformational adaptability, which may explain how the HIV-1 virus recognizes the RRE that exists in
flexible and heterogeneous conformations. Moreover, the highest degree of conformational flexibility, indicated by
the highest RMSF, was discovered in the region containing residues 100-190, which is associated with the trans-
dominant negative RevM10 mutant, and has been shown to form alternative conformations. Thus, our method made
novel advances and significant impacts in biology by determining structures of heterogeneous conformers of RRE that
have not been possible to determine previously.

The conformational heterogeneity is not limited to RNA like RRE. Even the well-structured RNA, such as RNase P
RNA, samples a large conformational space. As an example of HORNET capability, we determined three novel 11
structures of the RNase P RNA conformers. It is noteworthy to point out that the structure of 298-nt of the 417-nt 12 full
1-length RNase P RNA was determined using crystallography but the rest of the structure was not determined 13 because
the structure could not be modeled due to the poor electron density. **Redacted**

With regard to the comment, "*whether it can determine the ratio between different conformations*", the
answer is "Yes" as it has been demonstrated and validated in one of our previous publications (Nat Commun. 2023
Feb 9;14(1):714) **Redacted**

*My review mainly focuses on both the machine learning techniques (including unsupervised and supervised*
*learning) and the software code provided in this work. The authors used the classical PCA and K-means algorithms*
*for unsupervised learning. Then the lowest energy clusters were obtained by cohort selection. For the supervised*
*learning, the authors traversed the parameter space and generated a "pseudo-structure database" due to the lack of*
*large amounts of RNA structure data. Then they adopted the classical dense network modelling to predict the*
*difference between the real structure and the pseudo-structure. A similar approach to the one used in this study*
*(Townshend et al., Science, 2021) was implemented in the ARES software to overcome the problem of insufficient 3D*
*RNA structure data." While going through the codes in this manuscript, I found them overall not user-friendly.*
*Either necessary example files were missing, or certain documentation was not clear. At least it should be possible*
*for the user to run through the entire process with sample data. Here are the detailed comments based on my testing.*

The reviewer's comment "*A similar approach to the one used in this study (Townshend et al., Science, 2021)*
*was implemented in the ARES software to overcome the problem of insufficient 3D RNA structure data."* Is not
accurate. Our approach differs from ARES in three fundamental aspects where HORNET is novel. 1. Conceptually,
ARES is a structure prediction program for small RNA based on the information from structural databases, whereas
HORNET determines 3D structures of large RNA based on experimental AFM images; 2. ARES followed the general
concept, assumption and methodology of AlfaFold for structure prediction, i.e., for one primary sequence, there is
THE one corresponding 3D structure, whereas HORNET totally differs from AlfaFold and ARES, and considers one
primary sequence could fold into different conformations and determines structures of individual conformers. The
notion of one sequence to one RNA structure/fold/conformation is incorrect for RNA, especially for large multi-
domain RNAs as we have demonstrated in a recent publication (Ding, et al., Nat Commun. 2023). 3. ARES uses
conventional discrete fragments from structural databases and can only sample a limited and discrete conformational
space, whereas HORNET uses a pseudo-structural database with continuous conformational trajectories.

Even when considering neural networks, HORNET uses a novel architecture that is totally different from that
of ARES and performs remarkably better than ARES. It is noteworthy to mention that the intended only goal of the
ARES approach is to use the ARES scoring function to estimate the accuracy of a predicted structural model. As
shown in their manuscript in the Supplementary Information, the ARES scores predict RMSD poorly in general and
fail in cases of larger RNA, indicating that either the model of their geometric deep learning is inadequate, or the
fragment-based structure database is not sufficient, or both. In contrast, HORNET is capable of estimating accuracy
(RMSD) in all benchmarking and testing cases. A more detailed comparison in terms of accuracy prediction is listed
in **Appendix 1** at the end of this response.

One RNA primary sequence folding into multiple 3D conformations is the underlying premise of our study.
This is consistent with abundant existing studies but had never been directly visualized until recently after our
publication (*Ding et al., Nat. Comm., 2023*). Consequently, there are at least two important implications. 1. The
significance of a single experimentally determined or predicted structure. Our studies simply illustrate that a single
structure is not sufficient to describe the full conformational space of an RNA. This does not imply in any way that
snapshots of experimentally determined high-resolution structures are not important, but rather that the field needs to
take a fresh look at RNA in terms of its structure-dynamics landscape. But it does question the meaning of the accuracy
of a purely computationally predicted structural model, given that one sequence could fold into multiple
conformations: which structure of many possible conformers one is predicting? 2. Techniques/methods for structure
determination of heterogeneous RNA conformers, which is not to be confused with biophysical methods/techniques
for studying ensemble behaviors (as opposed to structure determination. It is clear that the current techniques/methods
are inadequate in determining the 3D structures of RNA heterogeneous conformers. All existing techniques/methods
for 3D structure determination rely on signal averaging over a large number of identical/very similar conformers and
thus are ill-suited for the structure determination of the heterogeneous RNA conformers. This is because the physical
signals from NMR, X-ray diffraction, or cryo-EM of a single molecule are too weak to be either detected (NMR or
crystallography) or interpreted (cryo-EM). AFM is currently the only biophysical method that is capable of directly
visualizing individual biomacromolecules and our study makes several very significant advances in RNA
conformational dynamics, which is one of the frontiers of RNA structural biology.

*Also, the software, "HORNET", is not user-friendly, and the authors should comprehensively fix their codes.*

We recognize that the software, at the time the manuscript was submitted, was not user-friendly. We have
ensured that the package now contains all necessary components and detailed instructions for a user to run the
software without any assistance from developers. To ensure that the README files instructions are clear, we have
requested a cell biologist, a biochemist and a psychologist to use our software. Neither of them has any experience
or training in computational structural biology, yet both could independently run HORNET successfully after
reading the README instructions. For clarity, we have provided the list of updates in the HORNET package and
the list is attached to the end of the response as **Appendix 2**.

*Firstly, I noted the high correlations observed between "etot", "hbond", "local" and "stack" among the 16 features*
*extracted by the authors, exceeding 0.9. Such high correlations raise the issue that these features convey redundant*
*information that can particularly lead to ineffective modelling.*

We thank the reviewer for running diligent tests and noting observations of concern regarding the
redundancy of features and the potential for overfitting; this is very important. First, we did not use 16 features for
training the model, but 10 in the DNN, as described in the methods section, and as can be verified inside the code.
These are: "etot", "local", "go", "repul", "stack", "hbond", "elect", "afmfit", "afmcc" and "cc7xEtot". In the
UML analysis, we used 11 features. The only difference is that "cc7xEtot" is replaced with "stage" and "kappa" in
the case of UML. The stage feature cannot be used for the training because this term is not generalizable to any RNA
molecule, as it depends on the molecular volume and shape derived from the AFM image, and "kappa" is not a
structure-related parameter but rather a scaling factor. Instead, "cc7xEtot" is used for training, which correlates the
total structural energy with the best agreement with the AFM image.

Importantly, the features available must allow for normalizations across different RNAs, which our study
demonstrates successfully. Some of the features, as you point out, are indeed highly correlated, as the total energy
(*Etot*) is the summation of all other energy terms, local and stack share some dependencies in their calculations, and
*cc7xEtot* is a combination of two terms. In light of these natural correlations, the features were explored deeply. In
fact, one of the first exercises we performed in the early stages of testing and development was to extensively
investigate how the model would behave when removing randomly each of the available features, or when including
combined features. Based on these tests, many original features were removed early on to optimize model
performance and the results of this study clearly demonstrate that the existing model performs extremely well. The
early stopping and dropout regularization functions we use are vitally important to avoid overfitting; they prevent
the neural network from giving too much attention to single features or specific neurons, and from making decisions
completely based on just a few learnings.

*Secondly, among the 200,000 rows of data used in the example, I extracted 100,000 samples and randomly*

*correlated them two by two, achieving an average correlation coefficient of 0.94. In this case, even if the training*
*and test sets were randomly assigned, they will reach very similar outputs that lead to a bias in the accuracy of*
*modelling. The authors should find a better way to validate the accuracy of their model.*

The two data sets that the reviewer refers to in the HORNET GitHub package are derived from the same set
of data at different noise levels (BM0). Therefore, they do not represent the whole range and distribution across all
datasets and different RNAs. As we could not provide all data used for training and testing in the GitHub package,
we have now made all data, including experimental data, available in our public repository:

<https://home.ccr.cancer.gov/csb/pnai/data/HorNet/>). We understand the reviewer's concern that the random splitting
of data alone is insufficient for validating the accuracy of the trained model. And that is precisely the reason we have
not used random splitting to split our data into training, validation, and testing, and that this was already extensively
described in the main text (p13, lines 11-23) and Session in Method: Optimized Architecture. To further clarify, we
will focus on how the model avoids bias through diverse data distributions and appropriate strategies for splitting the
data.

First, the benchmark simulations, which employ different starting models, data, and AFM potentials
(κ), produce trajectories with completely different energy ranges and data distribution. To ensure that a large
distribution of energies was sufficiently sampled, we trained and tested the model using a wide range of κ
values (2 to 50), and with different applied noise levels (5, 10 and 15%) (p69; Extended Data Table 3), where the
noise would be the equivalent of augmenting the dataset. Second, we used a common machine-learning approach to
split the benchmarking data into four groups—training, training-validation, validation, and testing—which is
necessary when using data from different sources (in this case, different RNAs) (p60; Extended Data Fig. 10). The
distributions were normalized based on the number of atoms of each RNA, or by other terms, providing an averaged
energy instead of a sum, which is then comparable across different RNAs that do not have the same distributions.
We then trained the model with the trajectories from a single RNA (100% of BM0 + 5% of BM1 + 5% of BM2),
whose data fractions were split into the training (80%) and training-validation (20%). The purpose of the training-
validation set is to ensure that the common regularization techniques are working and that the model is not
overfitting within the same data distribution (i.e., internal bias), but not to select the best epoch (Fig. 4b).

To prevent the biasing of trajectories of other RNAs toward those of the trained model, we must determine
the point at which the learning is no longer generalizable. This was done by comparing the loss functions for the
training set and the validation set (5% of BM5 alone), and select the weights of the network (the learning) at the
point at which the validation-set loss function hits a minimum (epoch 5), so that the learnings, even with a limited
parameter space of a single RNA, is still generalizable to other RNAs. This must be done because the usability of
the model is defined by its application to different RNAs never seen by the model. Finally, we blind-tested the
trained model with data from two completely different RNAs (BM3 and BM4) to ensure that the model performs
well with unseen data distributions. These successful blind tests, arguably, are the most rigorous tests for
demonstrating generalizability of the model. As such, the accuracy reached in our blind tests should, in principle, be
reproducible in other RNAs (p13&14;Method p39).

*Also, considering that these data emanate from a "pseudo-structure database," the authors should ensure uniform*
*sampling across the parameter space.*

Our design of the pseudo-structure database not only to just uniformly samples the energy landscape of one
RNA (RNase P RNA) but also to be able to learn the model generalizable to any other RNA's. Equally important,
our pseudo-structure database is continuous covering a wide range of conformational space (see the Appendix 2 for
details). To sample most of the possible structure energies features of RNA basis folding into 3D structure space, we
use three complete different sets of structures that started from three different initial structure of the same RNA
(RMSD among three initial structures are as large as $\sim 27\text{\AA}$ and the largest RMSD among structures in the 3.5
millions of the trajectory structures is 37\AA). However, the ultimate test for the robustness of the pseudo-structure
database is the benchmarking with never-seen structures. For this, we then tested the generalized performance of the
trained model for other never seen data, BM3 and BM4, which are also complete different RNA (Cbl Riboswitch
and Group II intron), in terms of size (210 and 387), structure features. These tests illustrate that the parameter
space sampled in the training set was adequate to ensure the applicability to unseen structures. This blind test is a
golden standard to prove the applicability to never-seen cases and illustrates the model is "learned" not
"memorized", and the model covers the "parameter space". Importantly, one of the main parameters is the overall
particle folding information embedded into AFM potential, which is a main key for "inform" the system about the
overall folding agreement between one model and what is observed under AFM. Nevertheless, the learned model
could be further improved, as we have indicated in the previous version of the manuscript.

*Additionally, in the script of the model training, the authors used a fixed number of 5 rounds. Is there any*
*justification of using 5 rounds? Some methods such as "earlystop" may help determine an optimal number of*
*rounds.*

The default value in the script for the maximum number of epochs is 300 (Fig. 4b). As mentioned in the
previous comment, the optimal number of epochs in the training is determined by the minimum loss on the
validation set. However, in the README file where an example was given for training a model with 5 epochs. To
avoid any confusion, the number of epochs in the README file is reset to 300. Furthermore, in the updated
package now includes the automatic selection of the optimal epoch in the training, whereby the program performs
the user-specified number of epochs but selects the epoch with the minimum value for the validation-set loss
function.

*Regarding the software part that I have tested, I am disappointed that most of the functions did not work. It is mainly*
*due to the lack of sample files (like: "ts" file, AFM data, and sampling data...). Therefore, I could not evaluate these*
*functions. In addition, when testing the individual scripts, the "README" instructions did not allow the correct*
*configuration of all the code. Therefore, I installed the required packages manually. If the software is to be formally*
*released, I strongly recommend the software should be packaged as a real Python software package (rather than a*
*collection of different bioinformatic scripts) and may consider releasing the package on public platforms such as*
*PYPI, Anaconda, Docker, etc. The authors should perform appropriate Python unit tests to determine the usability.*

The missing sample input files were misplaced in a different folder, and thus caused all the problems that
the reviewer mentions. We apologize for this oversight, and we have ensured that all necessary files have been
included in the updated package. This includes all the files needed for testing all our developed scripts, as well as
CafeMol input files with all optional arguments for running dynamic fitting and testing its standard functions. We
modified the dynamic fitting scripts for better suitability for running in non-HPC environments so the reviewers and
users can use and test locally, although the reproducibility of the extensive work presented in this paper would
require the usage of clusters/HPC for the dynamic fitting step. We also regrouped the scripts into a Python package,
so one can import the modules and create/adapt scripts as desired for individual cases. We adapted some scripts in
the scripts folder to be callable, making them easier to use without the need to modify the source code. We have
included python unit tests for all the functions of the modules and for the callable scripts, including necessary data
or mock data, as rightly suggested by the reviewer. Of note, we cannot distribute CafeMol's package because of
copyrights, so anything related to dynamic fitting was maintained outside the scope of HORNET installations and
script packages. We have also greatly improved the README file by including much more detail and examples to
make it more user-friendly. Lastly, we have ensured that HORNET can be installed and run by users without
intimate knowledge of the package by following the README file, and have provided a Docker image for
containerization if needed. For the summary of the revision regarding the software package, please also see the list
of changes at the end of this response (**Appendix 2**).

*Following the current instructions to create a conda environment for "python 3.7", it will cause compatibility*
*problems during the second "pip" installation dependency. The Error is: "ERROR: Ignored the following versions*
*that require a different python version, "ERROR: Could not find a version that satisfies the requirement*
*numpy==1.24." I get the same error on both mac and Linux terminals. Please modify the installation instructions in*
*the README.*

We thank the reviewer for pointing this out. We have upgraded Python to version 3.9 and modified the
package to be installed using pip, correcting any issue one might have. This was also updated in the README file.

*After I had installed all the dependent packages manually, I found that although the introduction mentions the*
*format of the input files, a runnable example file was missing. An example "ts", "bases_k<kappa>.txt" should be*
*given in the "Step 2 - Input Preparation" section. etc. to make the code for the examples runnable. I then installed*
*the cafeMol software and followed the "protocol.txt" step-by-step in the folder. However, in the first step*
*"path/cafeMol input.inp" gives an error ("cannot open the file: ./output/out_name.data").*

As stated above, the updated package now includes all necessary files, with example scripts, and
explanations in the README file. For completeness, we also included bases_k<kappa>.txt, although this file is not
mandatory to run the script. The issue seen by the reviewer in running the dynamic fitting has been corrected by
modifying the scripts to work locally.

*The example file, Full_Trajectory.csv, is not given, or relies on the output of step 2, so I was unable to run the*

*unsupervised learning part (step 3) of the code. For step 4, there is no example file or dependency on the data from*
*the previous step.*

With the scripts available in the previous version, the Full_Trajectory.csv was obtained by running step 2
with additional positional arguments, as described in the documentation. For full reproducibility, we included the file
“bases_k<kappa>.txt”. For convenience, a sample Full_Trajectory.csv is now provided in the /data/TUTORIAL
folder, to allow a user to run step 3 without having to run step 2. For step 4 (training and prediction using the deep
learning method), all the input data are in a single folder, as described in the documentation: two datasets for
training, one for validation, and another dataset to test predictions, and the script should have run without issue. As
such, we did not modify this part of the package.

*DNN's train.py is runnable. But there are still some issues. If "-l huber" is used in the parameter, then the error*
*"ValueError: invalid literal for int() with base 10: 'huber'" will be reported. Delete this parameter or modify the*
*code to work correctly.*

We thank the reviewer for pointing this out. The issue has been addressed in the updated package by fixing
the argument parsing layer that was placed on top of the scripts.

**Appendix 1. Comparing ARES and HORNET**

Reviewer 3 brought up the comparison of HORNET with ARES (*Townshend et al., Science, 2021*). ARES represents
a significant improvement over all existing programs for predicting RNA structures. Nevertheless, for review
purposes, we sum up the comparison of the two software. In doing so, we have no intention to criticize the ARES
approach but provide facts that could help reviewers understand why the pure-computational approaches will fail and
start to appreciate the urgent need for different thinking and approaches like HORNET.

Townshend et al. use the Atomic Rotationally Equivariant Scorer (ARES) function and machine learning approach to
estimate the accuracy of predicted structural models, whereas HORNET is the package for determining 3D atomistic
topological structures of individual RNA conformers. They differ in several fundamental and important aspects that
define their purposes and limitations.

**The purpose**

ARES is designed to estimate the accuracy of computationally predicted structural models using geometric deep
learning, whereas HORNET is the package to determine 3D atomistic topological structures and estimate the accuracy
of individual conformers using AFM images of particles and deep neural networks.

**Basic premise**

The ARES approach, like almost every other structural prediction software, assumes that for one primary sequence,
there is THE one corresponding 3D structure. HORNET has no such assumption and determines the structures of
individual conformers based on experimental topographic particle images. As we have demonstrated previously by
direct visualization, the assumption that one sequence leads to one 3D structure of RNA simply is contrary to
experimental observation via direct visualization by AFM and against RNA structural thermodynamics that govern
RNA folding. The fundamental reason for one RNA sequence to fold into possible multiple conformations is the
energetic degeneracy and geometric equivalency among RNA conformers due to its ragged energy landscape and
limited variety of the four building blocks. The only way to overcome energetic and geometric degeneracies is to
include global geometric/topographic constraints that delineate the topological folds of individual conformers. It is
noteworthy to point out that Townshend et al. appeared to realize the limitations of their approach and the importance
of experimental global constraints (the last second paragraph (*Townshend et al., Science, 2021*)).

**Structural database**

The ARES approach makes use of the fragment-based structural database (18 fragments), whereas HORNET uses a
pseudo structure database. The former is discrete and samples limited conformational space, whereas the latter is
continuous and samples much broader conformational space. As a result, the ARES approach may perform poorly

when a structure falls outside of its conformational space defined by its database such as it happens in some of the
puzzles.

4 **Deep learning**

The only similarity between the two programs is that both use deep learning. While ARES focused on the geometrical
aspects of the conformers for the scoring function hence applying a geometrical deep learning approach ending with
dense layers, HORNET focus purely on the energy and agreement to the experimental AFM image using dense layers
alone. In their paper, there is no plot showing the loss function as a function of the number of epochs since ARES was
trained using a single epoch - as stated in their supplementary materials (pg. 7). This means that it must have been
hard for them to validate their model as, even applying the best techniques, they were possibly already experiencing
overfitting in the very first epoch between the training set and validation set, pointing to a possibly less generalizable
model: “The set of hyperparameter values with the lowest loss was a batch size of 16, 1,000 randomly selected
structural models for each sequence, a learning rate of 0.01, and a single training epoch.” Nowadays, in light of the
new advancements in observing the dynamics of RNAs it is clear that an approach using a global constraint seems to
be more appropriate. In contrast, HORNET model shows all proofs that the model is being able to generalize its
learning, by following the general and expected loss pattern between datasets. With the loss curve we could find the
best weights given the validation set while also really tuning the hyperparameters, where the loss is shown in Figure
4.

More importantly, in estimating the accuracy of structural models using their AI model, which is the ultimate intended
goal of ARES, the ARES scores predict RMSD poorly in general and fail in cases of larger, indicating that either the
model of their geometric deep learning is inadequate, or the fragment-based structure database is not sufficient, or
both. In contrast, HORNET is capable of estimating accuracy (RMSD) well in all benchmarking and testing cases.

25 **Some detailed comparison**

Here are some detailed comparisons of the performance of ARES and HORNET. As one can see the key differences
in the fundamental aspects between the two programs lead to the dramatic difference in estimating accuracy of
structural models that are generated by either computational prediction or determination using AFM data.

Figure S4 from ARES supplementary information (pg. 15) shows the correlation between the ARES predictions
(score) vs the real RMSD for the 21 RNA puzzles (blind tests). As shown in their plot, the scoring function predicts
scores in the order of 3-10 Å where in many cases there are real RMSDs up to 40 Å, and ARES cannot accurately
predict the real value of the RMSD. This is fine as the authors did not claim in the end that ARES is predicting the
accuracy but rather acting as a scorer. Our model is not only acting as a scorer, but it can provide accuracy estimation
of the structure with reasonably small uncertainty (2.5 Å – see Fig. 4, pg. 14 of our manuscript) and even determine
nonconverged structure calculations such as S257 (Extended Data Figures 11, 12 and 13). More importantly, as RNA
gets larger, there could be more than one conformer for a given RNA primary sequence and ARES could not possibly
predict correct conformer structures because of energetic and geometric degeneracies.

Continuing in the same Figure S4 from ARES supplementary information (pg. 15), in most of the cases there is even
a poor correlation between ARES score and the true RMSD, which can be seen by eyes but also directly evaluated by
some very low Spearman correlation (average of Spearman correlation of 0.27 considering all puzzles). Our blind
tests show a clear relationship between the real and estimated RMSD, achieving a Pearson score of 0.77 and 0.84 on
our blind tests.

Table S5 from ARES supplementary information (pg. 29-33) compares the best ARES scored structures to other
scoring functions for 4 different RNAs for the RNA-puzzles (A, B, C and D). In summary, their results of the best-
scored structures as a function of the real RMSD are as follows:

50 A: The average RMSD of the top 10 structures is ~ 14.8 Å, with structures ranging from 4.8 – 21.8 Å (111 nt, 36.2
51 kDa)
52 B: Average RMSD of top 10 structures is ~ 16.9 Å, with structures ranging from 12.5 – 22.7 Å (130 nt, 48.1 kDa)
53 C: Average RMSD of top 10 structures is ~ 12.2 Å, with structures ranging from 9.5 – 15.1 Å (230 nt, 78.9 kDa)
54 D: Average RMSD of top 10 structures is ~ 28.5 Å, with structures ranging from 14.5 – 35.9 Å (175 nt, 56.7 kDa)
55

1 In comparison, in our blind tests we were able to achieve the estimation of RMSD correctly ($< 2.5 \text{ \AA}$ of difference)
2 and ALL the top 10 structures selected had $< 5 \text{ \AA}$ of real RMSD to the GT, not only one or other structures.
3

Appendix 2. HORNET Package –Update list

The complete code review and changes can be found in the latest Pull Request on the repository:
<https://github.com/PNAI-CSB-NCI-NIH/HORNET/pull/8>. all data for the calculations are available at
<https://home.ccr.cancer.gov/csb/pnai/data/HorNet/>

To give an overview of what was changed at an upper level, please find a list of modifications below:

- README file review: The readme file was carefully followed and reviewed by three persons who have limited computation skills and know nothing about HORNET. The complete comparison between the previous readme and the current can be found here (by pressing **ctrl** and double click for **Windows**)
- Tutorial Files: All the intermediate files necessary to run the package were included in a folder called data/TUTORIAL. Now, all the initial, intermediate, and even final files are there to serve as examples. For clarity, a list of input files is shown below, and black-colored names are the files that were already there, and green ones are those included at this time (even files that were optional and not needed to run the package).
 - For input preparation:
 - En_allk14.txt
 - En_allk22.txt
 - Bases_k14.csv (was an optional entry but included anyway)
 - Bases_k22.csv (was an optional entry but included anyway)
 - For UML analysis:
 - Full_Trajectory.csv (intermediate file created in the input preparation)
 - For the Deep Learning predictions:
 - Prediction_sample.csv
 - Full_Trajectory.csv (intermediate file created in the input preparation)
 - Filtered_Data.csv (intermediate file created in the UML analysis)
 - Select_Cluster.csv (intermediate file created in the UML analysis)
 - Final_Cohort.csv (intermediate file created in the UML analysis)
 - For the Deep Learning training:
 - Train_sample1.csv
 - Train_sample2.csv
 - Validation_sample.csv (sample of our validation set)
 - As final outputs from the prediction:
 - Final_Cohort_Top10.csv (examples of output from prediction)
 - Final_Cohort_prediction.csv (examples of output from prediction)
- Package Install: Since users asked for it to be an installable package, we restructured the code to be in an installable format, mainly regrouping the previous scripts under UML and DNN folder to another folder called src/hornet. A setup.py file was included in the repository so that a user could simply install the package using the pip command from Python. Although we would not see this step as crucial to the work as it does not interfere with the ability of one to run the package, we agree that it brings more clarity and organization, being the main reason to update the code but as it takes important time to do when considering a publication, had not been done before.
- Unit Tests: Users requested the package to have unit tests, mainly to assure the operability of the scripts to run a complete pipeline as a test. We went further than just testing a pipeline of the 4 steps by creating a separate set of unit tests for all the functions we have in each module. This gives a total of:
 - 14 tests contained in 4 testing files for the INPUT module (data engineering).
 - 21 tests contained in 8 testing test files for the UML module (UML analysis).
 - 49 tests contained in 11 testing test files for the MODEL module (deep learning).
 - 52 tests contained in 4 testing test files for the wrapper of the scripts, where each complete step of the pipeline is tested in each file, finally completing the whole pipeline of 4 steps using the data provided in the tutorial, assuring all of the code has been tested and can be run throughout without issues.

The bug found by users in one of the wrapper scripts was fixed and would not interfere with the inner core
functionality. No new bug was found during the process of creating the unit tests. All the code
modifications can be found in the PR link provided above.

- • Containerization: Users had trouble installing the environment (install the environment is different from the
one we've tested), we created a docker image with a prepared environment installed, already containing the
HORNET package inside as well. If one had trouble installing the environment this time, we are offering a
clean environment for the package to be used, making it practically impossible for someone to fail the test.
We also make the Docker File available, so any user can simply create their own containers if somehow
needed, given that we provided the docker commands in the README to follow through. Importantly, the
unit tests are automatically performed during the docker image building process. The complete building
pipeline can be seen in the Docker file. *IMPORTANT NOTE: We do not have Copy Rights to distribute the*
*Cafemol package (the dynamic fitting software), so it is not installed in the docker image we are releasing.*
*For Cafemol, please go to <https://www.cafemol.org/> for instructions and the download.*
- • We included in the updated code the selection of the best model giving the best loss on the validation set in
the training script, also offering a sample of our validation set to follow exactly what was done in the paper.
- • On the Dynamic Fitting side, the README file was also updated to include more explanations, and the
complete comparison of the previous file and the new one can be found here.
- • Practically all the Dynamic Fitting scripts were refactored to allow a user to run it locally, instead of using
High Performance Computing environments, and all necessary example files and scripts were added to
bring more clarity and convenience.
- • To make it fully reproducible, we included a complete toy example where the data, simulated AFM image,
initial crystal structure file (already centered over the AFM image) and configuration files are already set
up for someone to just run the dynamic fitting using those files as input.
- • For conversion of CG to all-atom model we provide a separate instruction with one example file that can be
found here.

Reviewer Reports on the First Revision:

Referees' comments:

Referee #1:

The authors' response to the previous round of reviewers' comments is thoughtful and considered, demonstrating that the authors have taken onboard the suggestions and have made significant revisions to their work in light of these. A major advancement in the revised work is the addition of improved GitHub documentation for running HORNET, refactoring of scripts and rectification of bugs that were identified during previous testing of the software. The software runs successfully and the provided training example can be worked through in ~5 min, which aligns with that estimated by the authors.

The authors state that their software was successfully run by a cell biologist and biochemist with no previous experience in computational structural biology and no additional support other than the online documentation. These examples demonstrate that the software now contains sufficient documentation that can be understood and followed by users with varying expertise. The authors have taken the time to restructure their GitHub file format to ensure all input and data files required for testing the software are contained in a "TUTORIAL" subfolder. This adds to the ease of testing the software as now any step in the pipeline can be performed without the need to generate prerequisite files.

The authors' revision of Fig. 1 provides clarification of required input data and the stages of the HORNET workflow that were ambiguous in the previous version and each stage of the workflow is now clearly explained, both visually and within the text. The pipeline stages are also well documented within the GitHub README, together with tutorials and customisable scripts so that users can use their own training data and set different epochs, kappa and frame limits, and loss functions etc. This is a significant addition to the original submission and the option for users to customise the code to suit their own applications will attract a much larger user base.

The authors effectively justify the novelty of their methods/software by referencing recent studies that focus on single snapshot structures, contrasting with their approach that considers the broader conformational landscape of RNA. The fact that the authors show that their workflow can recapitulate 3D topological structures from AFM images using initial structural models from a range of sources such as simulations, crystal structures and cryo-EM represents a really promising advancement in the field and suggests that this work will have broad impact. Furthermore, the authors provided an unpublished study that makes use of HORNET to map the conformational space of RNA which provided a good exemplification of how the tool can support and advance our understanding of RNA structure and dynamics and increased the novelty.

Regarding automation of the full pipeline (from particle detection through to recapitulating 3D topological structures), the authors state that is a current work in progress and will likely be released in a future version of HORNET. This would significantly improve the usability of the tool.

Referee #1 (remarks on code availability):

These are detailed above but can be summarised as this is now usable and well documented.

Referee #2 (paraphrased by editor):

This referee did not provide direct comments for the authors, but noted a few items in the confidential comments. First, s/he notes that it is not necessarily true that because prior AFM work with DNA or nucleosomes showed no difference due to the surface does not mean that there are not some RNAs (that are more inherently floppy) might not show some effect of the surface. If surface binding favors a large, flat surface area within a particular conformer, then it will also favor conformers with larger flat surfaces and thus is expected to perturb equilibria. Whether or not this happens is would depend on the size of the preferences and the kinetics of interconversion vs surface adhesion, features that likely need empirical measurement and may vary from one RNA to another. Therefore, you should ensure this possibility is clear, as empirical measurements have not been done. Second, this referee had noted that it has not been determined whether the AFM approach (and data pipeline) can detect a moderately broad ensemble and changes across such an ensemble. Consequently, there should be a clear statement of not only the capabilities but also the potential limitations, and also of what has been directly tested and what is postulated.

Referee #3:

I would like to express my gratitude to the authors for their insightful and detailed response. I appreciate, as highlighted in the authors' reply, that the principal objective of this research lies not in pioneering new machine learning algorithms but rather in the adept application of these techniques to address a specific scientific enquiry. The method of combining atomic force microscopy (AFM) images with the machine learning approach, HORNET, to investigate RNA conformational diversity is indeed fascinating. Perhaps there was a misinterpretation in my earlier remarks; my intention was not to suggest that the authors employ more complex models with additional parameters. Instead, I aimed to highlight the existence of alternative effective methods currently available in the field (for instance, algorithms to identify optimal parameters within machine learning's parameter space, models based on pre-training neural networks, and attention-based frameworks in deep learning). Of course, it is not my desire to advocate for a switch to a different model, especially given that the applicability of HORNET in this context has been convincingly demonstrated by the authors.

I am delighted to acknowledge the authors' efforts in demonstrating the general applicability and accuracy of HORNET. Their detailed explanation regarding the various RNA types used for benchmarking, coupled with the rigorous testing protocols, significantly bolsters the credibility and potential impact of their approach. The successful identification of eight novel RNA structures is

particularly laudable and highlights the importance of this research in the field of structural biology.

The response to the concerns regarding the software is also well-received. The authors' dedication to refining their software package, ensuring it is thoroughly documented and user-friendly, represents a commendable step towards broadening the accessibility of their research.

In conclusion, the authors have thoughtfully addressed the initial concerns raised. The improvements in the software, together with the methodological rigour evident in their study, are commendable. I have no further comments to add at this stage.

Referee #3 (remarks on code availability):

Based on my review of the code and its functionalities, I can confirm that it is well-structured and accessible for community use. The inclusion of a README file, which offers comprehensive instructions for installation and execution, significantly enhances its usability. Additionally, the presence of example files serves as a practical guide for users, aiding them in understanding and utilizing the code effectively.

Author Rebuttals to First Revision:

Referee #1:

The authors' response to the previous round of reviewers' comments is thoughtful and considered, demonstrating that the authors have taken onboard the suggestions and have made significant revisions to their work in light of these. A major advancement in the revised work is the addition of improved GitHub documentation for running HORNET, refactoring of scripts and rectification of bugs that were identified during previous testing of the software. The software runs successfully and the provided training example can be worked through in ~5 min, which aligns with that estimated by the authors.

The authors state that their software was successfully run by a cell biologist and biochemist with no previous experience in computational structural biology and no additional support other than the online documentation. These examples demonstrate that the software now contains sufficient documentation that can be understood and followed by users with varying expertise. The authors have taken the time to restructure their GitHub file format to ensure all input and data files required for testing the software are contained in a "TUTORIAL" subfolder. This adds to the ease of testing the software as now any step in the pipeline can be performed without the need to generate prerequisite files.

The authors' revision of Fig. 1 provides clarification of required input data and the stages of the HORNET workflow that were ambiguous in the previous version and each stage of the workflow is now clearly explained, both visually and within the text. The pipeline stages are also well documented within the GitHub README, together with tutorials and customisable scripts so that users can use their own training data and set different epochs, kappa and frame limits, and loss functions etc. This is a significant addition to the original submission and the option for users to customise the code to suit their own applications will attract a much larger user base.

The authors effectively justify the novelty of their methods/software by referencing recent studies that focus on single snapshot structures, contrasting with their approach that considers the broader conformational landscape of RNA. The fact that the authors show that their workflow can recapitulate 3D topological structures from AFM images using initial structural models from a range of sources such as simulations, crystal structures and cryo-EM represents a really promising advancement in the field and suggests that this work will have broad impact. Furthermore, the authors provided an unpublished study that makes use of HORNET to map the conformational space of RNA which provided a good exemplification of how the tool can support and advance our understanding of RNA structure and dynamics and increased the novelty.

Regarding automation of the full pipeline (from particle detection through to recapitulating 3D topological structures), the authors state that is a current work in progress and will likely be released in a future version of HORNET. This would significantly improve the usability of the tool.

Referee #1 (remarks on code availability):

These are detailed above but can be summarised as this is now usable and well documented.

We thank the referee for the kind words and constructive review comments/suggestions.

Referee #2 (paraphrased by editor):

This referee did not provide direct comments for the authors, but noted a few items in the confidential comments. First, s/he notes that it is not necessarily true that because prior AFM work with DNA or nucleosomes showed no difference due to the surface does not mean that there are not some RNAs (that are more inherently floppy) might not show some effect of the surface. If surface binding favors a large, flat surface area within a particular conformer, then it will also favor conformers with larger flat surfaces and thus is expected to perturb equilibria. Whether or not this happens is would depend on the size of the preferences and the kinetics of interconversion vs surface adhesion, features that likely need empirical measurement and may vary from one RNA to another. Therefore, you should ensure this possibility is clear, as empirical measurements have not been done. Second, this referee had noted that it has not been determined whether the AFM approach (and data pipeline) can detect a moderately broad ensemble and changes across such an ensemble. Consequently, there should be a clear statement of not only the capabilities but also the potential limitations, and also of what has been directly tested and what is postulated.

Referee #2 original concern was about the distortion of RNA conformation by covalent attachment of RNA to a mica surface (not sure where he got the idea of covalent attachment to a mica surface), and we have addressed his question in the Rebuttal. Now, the reviewer brought up another question. First, the statement that *it is not necessarily true that because prior AFM work with DNA or nucleosomes showed no difference due to the surface does not mean that there are not some RNAs (that are more inherently floppy) might not show some effect of the surface.* In our prior publications (Ding, et al., Nat. Comms, 2023; Ding et al., NAR, 2023), we have corroborated our AFM studies of cobalamin riboswitch RNA with results from orthogonal techniques such as SAXS, ITC (Ding, et al., Nat. Comms, 2023) and cryo-EM (Ding et al., NAR, 2023). Should RNA molecules be distorted significantly on a mica surface, the synthesized SAXS curves and the orthogonal ITC data would not possibly have agreed with the conformation population tallies made from AFM images. The structural and conformational heterogeneity of cobalamin riboswitch RNA observed by AFM is also seen in our cryo-EM data, where this is no surface immobilization.

Second, about the statement *If surface binding favors a large, flat surface area within a particular conformer, then it will also favor conformers with larger flat surfaces and thus is expected to perturb equilibria. Whether or not this happens is would depend on the size of the preferences and the kinetics of interconversion vs surface adhesion, features that likely need empirical measurement and may vary from one RNA to another. Therefore, you should ensure this possibility is clear, as empirical measurements have not been done.* AFM images capture conformations at the moment that molecules are deposited on a mica surface and we see no change to those particles after imagining them several times, indicating the conformations of those molecules stay the same on the spatial scale discernable by AFM and HORNET. Thus, there is no equilibrium and interconversion to perturb. I would like to stress that the spatial uncertainty of the HORNET method is on ~ 5 Å, at which local structural fluctuations and bond vibrational motions on the *fs*-*ns* timescale could occur but are beyond the resolution limit of the HORNET method and the conformational dynamics problems that HORNET addresses. We replaced the word “accurate” with “low-resolution” in the last sentence of Discussion to avoid confusion. About “*empirical measurement*”, we are not aware of any practical “empirical measurement” one can make to address this “perturb equilibrium” when no such an equilibrium exists on the mica surface.

Finally, the statement, “*it has not been determined whether the AFM approach (and data pipeline) can detect a moderately broad ensemble and changes across such an ensemble. Consequently, there should be a clear statement of not only the capabilities but also the potential limitations, and also of what has been directly tested and what is postulated.*” We have addressed his related comments in Rebuttal (*lines 20-31 and 41-53 in p10; lines 38-49 in p12; lines 5-16 in p14*). In addition, we have cited our prior publications **Redacted**, which directly addresses his question. We are disappointed the contents of those materials have not been read or understood by this reviewer.

Referee #3:

I would like to express my gratitude to the authors for their insightful and detailed response. I appreciate, as highlighted in the authors' reply, that the principal objective of this research lies not in pioneering new machine learning algorithms but rather in the adept application of these techniques to address a specific scientific enquiry. The method of combining atomic force microscopy (AFM) images with the machine learning approach, HORNET, to investigate RNA conformational diversity is indeed fascinating. Perhaps there was a misinterpretation in my earlier remarks; my intention was not to suggest that the authors employ more complex models with additional parameters. Instead, I aimed to highlight the existence of alternative effective methods currently available in the field (for instance, algorithms to identify optimal parameters within machine learning's parameter space, models based on pre-training neural networks, and attention-based frameworks in deep learning). Of course, it is not my desire to advocate for a switch to a different model, especially given that the applicability of HORNET in this context has been convincingly demonstrated by the authors.

I am delighted to acknowledge the authors' efforts in demonstrating the general applicability and accuracy of HORNET. Their detailed explanation regarding the various RNA types used for benchmarking, coupled with the rigorous testing protocols, significantly bolsters the credibility and potential impact of their approach. The successful identification of eight novel RNA structures is particularly laudable and highlights the importance of this research in the field of structural biology.

The response to the concerns regarding the software is also well-received. The authors' dedication to refining their software package, ensuring it is thoroughly documented and user-friendly, represents a commendable step towards broadening the accessibility of their research.

In conclusion, the authors have thoughtfully addressed the initial concerns raised. The improvements in the software, together with the methodological rigour evident in their study, are commendable. I have no further comments to add at this stage.

Referee #3 (remarks on code availability):

Based on my review of the code and its functionalities, I can confirm that it is well-structured and accessible for community use. The inclusion of a README file, which offers comprehensive instructions for installation and execution, significantly enhances its usability. Additionally, the presence of example files serves as a practical guide for users, aiding them in understanding and utilizing the code effectively.

We thank the reviewer for providing insightful and constructive suggestions/comments.